# Northern hemispheric atmospheric ethane trends (2006-2016) with reference to methane and propane

Mengze Li [1,2], Andrea Pozzer [1], Jos Lelieveld [1,3], Jonathan Williams [1,3]

1 Max Planck Institute for Chemistry, Hahn-Meitner-Weg 1, 55128 Mainz, Germany

2 Now at: Department of Climate and Space Sciences and Engineering, University of Michigan, Ann Arbor, USA

3 Climate and Atmosphere Research Center, The Cyprus Institute, 1645, Nicosia, Cyprus

*Correspondence to*: Jonathan Williams (jonathan.williams@mpic.de); Andrea Pozzer (andrea.pozzer@mpic.de); Mengze Li (mengze.li@mpic.de)

**Abstract.** Methane, ethane, and propane are among the most abundant hydrocarbons in the atmosphere. These compounds have many emission sources in common and are all primarily removed through OH oxidation. Their mixing ratios and long-term trends in the upper troposphere and stratosphere are rarely reported due to the paucity of measurements. In this study, we present long-term (2006-2016) northern hemispheric ethane, propane, and methane data from airborne observation in the Upper Troposphere - Lower Stratosphere (UTLS) region from IAGOS-CARIBIC project, combined with atmospheric model (EMAC) simulations for ethane at the same times and locations. The model simulations, and methane and propane observations provide additional information for understanding northern hemispheric ethane trends and emissions, which is the major focus of this study. The model uses the Copernicus emission inventory CAMS-GLOB and distinguishes 13 ethane emission sectors (natural and anthropogenic): BIO (biogenic emission), BIB (biomass burning), AWB (agricultural waste burning), ENE (power generation), FEF (fugitives), IND (industrial processes), RES (residential energy use), SHP (ships), SLV (solvents), SWD (solid waste and wastewater), TNR (off-road transportation), TRO (road transportation), and AIR (aviation). The results from the model simulations were compared with observational data and further optimized. The Northern Hemispheric (NH) upper tropospheric and stratospheric ethane trends were $0.33 \pm 0.27$ (mean ± one standard deviation) %/yr and $-3.6 \pm 0.3$%/yr, respectively, in 2006-2016. The global ethane emission for this decade was estimated to be 19.3 Tg/yr. Trends of methane and propane, and of the 13 model sectors provided more insights on the variation of ethane trends. FEF, RES, TRO, SWD, and BIB are the top five contributing sectors to the observed ethane trends. An ethane plume for NH upper troposphere and stratosphere in 2010-2011 was identified to be due to fossil fuel-related emissions, likely from oil and gas exploitation. The discrepancy between model results and observations suggests that the current inventories have underestimated ethane emission and must be improved and higher temporal-spatial resolution data of ethane are needed. This dataset is of value to future global ethane budget estimates and the optimization of current ethane inventories. The data are publicly accessible at https://doi.org/10.5281/zenodo.6301729 (Li et al., 2021).

# 1. Introduction

Ethane ($C_2H_6$) is among the most abundant non-methane hydrocarbons (NMHC) present in the atmosphere. Major sources of ethane to the atmosphere are via natural gas and oil production (~62%), biofuel combustion (20%), and biomass burning (18%). Interestingly, 84% of its total emissions are from the Northern Hemisphere (NH) (Xiao et al., 2008). Oxidation by hydroxyl (OH) radicals is the major atmospheric loss process for tropospheric ethane, while in the stratosphere the reaction with chlorine (Cl) radicals provides an additional loss process (Li et al., 2018). Due to the seasonal variation of ethane emissions and the photochemically generated OH radicals, ethane has a clear annual cycle in mole fractions, showing higher levels in winter. Its global lifetime is circa three months, with a minimum in summer (~2 months) and a maximum in winter (~10 months) (Xiao et al., 2008; Helmig et al., 2016; Li et al., 2018). Ethane oxidation forms acetaldehyde, which in turn contributes to the formation of PAN (peroxyacetyl nitrate) or peracetic acid depending on the levels of $NO_x$ (Millet et al., 2010). PAN acts as a reservoir species of nitrogen oxides ($NO_x$) and can strongly affect tropospheric ozone distributions by transporting $NO_x$ from the point of emission to remote locations. Furthermore, PAN is known to be a secondary pollutant like ozone with negative impacts on regional air quality and human health (Rudolph, 1995; González Abad et al., 2011; Fischer et al., 2014; Monks et al., 2018; Kort et al., 2016; Tzompa-Sosa et al., 2017; Dalsøren et al., 2018; Pozzer et al., 2020).

Several recent studies have estimated global ethane budgets using a combination of observations and model simulations. Xiao et al. (2008) estimated a global ethane source of 13.0 Tg/yr based on methane emissions for the 1990s. This study included information on sectoral and geographical ethane emissions, although the inventory might partially be outdated, at least for North America, due to the changes in oil and gas extraction since 2004 (Tzompa-Sosa et al., 2017). Simpson et al. (2012) reported a total 21% decrease in global ethane emissions from 14.3 to 11.3 Tg/yr from 1984 to 2010, likely due to the decline in fugitive emissions from fossil fuel extraction and use. Monks et al. (2018) estimated the global ethane emission in 2008 to be 15.4 ± 2.3 Tg/yr. Hausmann et al. (2016) calculated the contribution from oil and natural gas to the total ethane emission increase of 1-11 Tg/yr over 2007-2014. Franco et al. (2016) reported a global ethane emission of 18.2 Tg/yr for 2014 and that North American anthropogenic ethane emissions increased by 75% over 2008-2014. Helmig et al. (2016) calculated a growth rate of 0.42 (±0.19) Tg/yr of NH ethane emission

between mid-2009 and mid-2014, and Pozzer et al. (2020) estimated a 2.1 Tg/yr increase of global
anthropogenic ethane from 13.2 to 15.3 Tg/yr over the same period.

Despite the general agreement in global emission estimates, multiple studies have pointed out that the current inventories used in atmospheric chemistry models underestimate ethane emissions by up to a factor of 2-3 (Tzompa-Sosa et al., 2017; Angot et al., 2021; Monks et al., 2018; Dalsøren et al., 2018; Franco et al., 2016; Pétron et al., 2014; Tilmes et al., 2016; Emmons et al., 2015).
Dalsøren et al. (2018) concluded that the major source of uncertainty in these inventories comes from the assumed speciation of NMVOCs (non-methane volatile organic compounds) and disaggregation of carbon emissions into individual species based on little available data. Therefore, to determine the global ethane trends in terms of mole fractions and emissions with greater certainty, long-term global ethane datasets from observations and model simulations with minimal influences
from local sources (e.g. observations at higher altitudes) are required (Angot et al., 2021; Gardiner et al., 2008).

Previous studies attempting to understand the distribution, emissions, lifetime, and atmospheric trends of ethane have tended to be from surface sites, either from a regionally focused intensive field measurement campaign (e.g. Kort et al. (2016)) or from networks of remote sampling
stations (e.g. Franco et al. (2015), Helmig et al. (2016)). The advantage of surface sites is that they are easily accessed and maintained, however, such measurements inevitably reflect the local or regional situation, and changes in emissions immediately upwind of a measurement location can affect the results, masking any underlying long-term global trend. In addition, most ethane measurement sites are located in developed countries, such as North America and Europe, while
ethane observations in the rest of the world are sparse. This too hinders the assessment of global ethane trends, for while one country's emission may be declining another's could be increasing rapidly. For the aforementioned reasons, it is advantageous to assess the global long-term ethane trend from the upper troposphere and even the stratosphere where emissions can be expected to be well mixed by atmospheric circulations. In particular, the trend of ethane in the more isolated
and remote stratosphere is of interest when assessing long-term changes.

In this study, we use airborne observations covering the Northern Hemisphere (NH), including over regions where ground measurements are not set up or not possible. We present long-term northern hemispheric and geographically delineated (North America, Asia, Europe) ethane trends

in the upper troposphere and stratosphere for the decade 2006-2016 derived using airborne measurements and global model simulations. In addition, the trends of methane and propane collected from the same observations are examined to better understand the observed variation of NH ethane trends, as they have common sources and sinks in the atmosphere. All the data used in this study are publicly available at https://doi.org/10.5281/zenodo.6301729. These data can be used for further analysis on global and regional trends, emissions and lifetime of methane, ethane, and propane, their contributions to climate change, stratosphere-troposphere exchange, and improvement of current inventories and atmospheric models.

## 2. Materials and Methods

### 2.1 IAGOS-CARIBC observation

The IAGOS-CARIBIC project (In-service Aircraft for a Global Observing System-Civil Aircraft for the Regular Investigation of the atmosphere Based on an Instrument Container) is an aircraft-based scientific project with the aim of monitoring long-term global atmospheric physics and chemistry (Brenninkmeijer et al., 2007). The flight altitudes are at ~10 km, which is in the Upper Troposphere-Lower Stratosphere (UTLS) region. A custom-built whole air sampler collects pressurized air samples during each flight, and these samples are subsequently measured in the laboratory with Gas Chromatography (GC) coupled with three detectors: GC-ECD and GC-FID for greenhouse gas measurements (methane, carbon dioxide, nitrous oxide, and sulphur hexafluoride) (Schuck et al., 2009), and GC-FID and GC-AED for volatile organic compound measurements, including ethane and propane (Baker et al., 2010; Karu et al., 2021). The precision of ethane and propane data used in this study is 0.2% and 0.8%, respectively (Baker et al., 2010), and of methane 0.17% (Schuck et al., 2009). Details regarding operational and analytical procedures, calibration scales, and quality assurance are well documented in the cited references, and summarized as follows.

Each IAGOS-CARIBIC flight normally consists of four flight sequences with a total number of 116 air samples collected by whole air samplers (flasks). The inlet and outlet of each flask are connected by multi-position valves which can be automatically switched with programming. A

pumping system and pressure sensors are connected to the inlet valves to guarantee the final pressure in each flask to be around 4.5 bar. The outlet valves are connected to ambient air. Prior to pressurization, each flask is flushed with ambient air for 10 times (about 5-10 min). The average filling (sampling) time of each flask is about 45s (range 0.5-1.5 min) depending on the flight altitude, resulting a spatial resolution of 7-21km.

Methane, ethane, and propane were measured with a HP 6890 GC with a polymer Porapak Q 3/4" column (10 ft, 100/120 mesh) installed in a single oven. Nitrogen ($N_2$, purity 99.999%) was used as carrier gas at a constant flow rate of 50ml/min. The GC was operated at oven temperature of 220°C with flow rates of synthetic air of 250ml/min and hydrogen of 80ml/min. Water vapor in samples was removed by passing through a drying tube at the start of the analysis. The calibration standards and reference gas cylinders were ordered from NOAA (for methane), and the National Physical Laboratory (for ethane and propane) which are certified against World Meteorological Organization (WMO) Global Atmosphere Watch (GAW) program scale, and they are regularly renewed within every three years which warrants the stability of calibration gases. Three injections of calibration standards were made in between samples of each flight sequence in order to maintain the quality of measurements and reduce uncertainty.

The upper tropospheric and stratospheric air samples were differentiated by using potential vorticity (hereafter PV, unit PVU). Northern hemispheric air samples with PV larger than 2 PVU were identified as stratospheric samples, otherwise as upper tropospheric samples. Figure 1 shows the geographical distributions of upper tropospheric and stratospheric samples, and spatial segregation. It is noted that the region designated must not correspond to the source region, only the geographical location of the data points.

## 2.2 EMAC global model

The ECHAM/MESSy Atmospheric Chemistry (EMAC) model is a numerical chemistry and climate simulation system that includes sub-models describing tropospheric and middle atmosphere processes and their interaction with oceans, land, and human influences (Jöckel et al., 2010). It uses the second version of the Modular Earth Submodel System (MESSy2) to link multi-institutional computer codes. The core atmospheric model is the 5th generation European Centre Hamburg general circulation model (ECHAM5, Roeckner et al. (2006)). For the present study, we

applied EMAC (ECHAM5 version 5.3.02, MESSy version 2.55.0) in the T63L47MA-resolution, i.e. with a spherical truncation of T63 (corresponding to a quadratic Gaussian grid of approx. 1.8 by 1.8 degrees in latitude and longitude) with 47 vertical hybrid pressure levels up to 0.01 hPa (~80 km). The model has been weakly nudged towards the ERA5 reanalysis data of the ECMWF (Hersbach et al., 2020). The chemical mechanism comprises methane, alkanes, and alkenes up to $C_4$, ozone, odd nitrogen, some selected non-methane hydrocarbons (NMHCs), heterogeneous reactions, etc. In total, 310 reactions of 155 species are included in the model. The photolysis rates are calculated following Sander et al. (2014). No chlorine chemistry is included in the model. To account for realistic emissions, the CAMS-GLOB-ANT v4.2 emission inventory data is used for model simulations (Granier et al., 2019; Guevara et al., 2020). In this study, we have included 13 emission sectors (shown in Table 1) which are BIO (biogenic emission), BIB (biomass burning), AWB (agricultural waste burning), ENE (power generation), FEF (fugitives), IND (industrial processes), RES (residential energy use), SHP (ships), SLV (solvents), SWD (solid waste and waste water), TNR (off-road transportation), TRO (road transportation), and AIR (aviation). It is noted that AIR, BIB and BIO were combined as one sector to reduce the uncertainty. AIR is not shown in Table 1 as its contribution is negligible. It has been shown by multiple studies that the ethane emissions due to fossil fuel combustion are strongly underestimated in the emissions database (Guevara et al., 2021; Pozzer et al., 2020; Helmig et al., 2016). In this work, we therefore increased the anthropogenic emissions of ethane of a factor of 2.47 to match (for the year 2010) the total amount suggested by Pozzer et al. (2020) although the value used in this study (~11.8 Tg/yr) slightly underestimates the measured mole fraction as shown in Pozzer et al. (2020) (13.2 Tg/yr). We further optimized modeled ethane mole fractions for each emission sector (referred to as "opt" in the later figures and "optimized" in the later texts). The model optimization is done by increasing the emissions of each input emission sector by 45%. We found that the root mean squared error (RMSE) between the modeled and observational ethane mole fractions for the whole dataset was at a minimum after a 45% increase in the input emissions. The input ethane emissions from natural and anthropogenic sources are presented in Table 1, together with a description for each sector and optimized sectoral emissions (will be discussed in the *Results and Discussion* section).

In this study, two types of ethane trends were presented with the model simulation: (1) constant meteorology and constant emission (hereafter called climatology), sampled at the IAGOS-

CARIBIC sampling location with S4D algorithm (sampling in 4 dimensions) described in Jöckel et al. (2010). Any trends (or changes) detected in this simulation would be caused by differences in sample location and timing. (2) real meteorological conditions from ECMWF and the adjusted emissions described above, sampled at the IAGOS-CARIBIC sampling location with S4D algorithm (Jöckel et al., 2010).

## 2.3 Trend analysis

The trend and seasonality analysis algorithm ("Prophet") used in this study has been described in detail elsewhere (Taylor and Letham, 2018). The "Prophet" algorithm has been shown to perform well with non-continuous time-series datasets (Li et al., 2022), as is the case for aircraft data. The trend analysis model has four components: trend (non-periodic changes), seasonality (periodic changes), holiday effects, and error (idiosyncratic changes). In this study, effects of holidays are not included. We used a linear model with change points for the trend component, and the trend function consists of growth rate, adjustments of growth rate, and offset parameter. The flexibility of trend (e.g. overfitting or underfitting) can be adjusted by the parameter "changepoint_prior_scale". A change point represents the moments where the data shifts directions. The value of the parameter "changepoint_prior_scale" represents the strength of change points, more change points will be automatically detected when the value of this parameter increases. Seasonality is estimated by Fourier series (Harvey and Shephard, 1993). The uncertainty interval was set to be 95%. The code of trend analysis in Python for this study can be found in the Supplementary Material. Figure S1 shows the ethane trend and seasonality at Iceland estimated by "Prophet" algorithm. Compared with the trend and seasonality estimated by the NOAA algorithm (www.esrl.noaa.gov/gmd/ccgg/mbl/crvfit/crvfit.html) using the same dataset in Figure 1(b) of Helmig et al. (2016), the seasonality of ethane is well captured by both algorithms and the results match well with each other. The uncertainty from the trend analysis is estimated by applying ten fitting levels on the trend (i.e. "changepoint_prior_scale" = 0.1, 0.2, 0.3, …, 0.9, 1.0). The difference between the most underfitting to most overfitting is taken as the uncertainty and the average value of the ten fitting levels is used to represent the underlying long-term trend.

In this study, we also calculated simple linear trends (hereafter as linear trend to distinguish with the trends derived from "Prophet" algorithm) within a time period as follows:

Linear trend $= (c_{End}-c_{Start})/(t_{End}-t_{Start})$   (1)

where $t_{End}$ and $t_{Start}$ represent the end and start date and time of the target time period, $c_{End}$ or $c_{Start}$ is the mole fraction of trace gases (ethane, methane or propane) at the end or start date and time.

## 3. Results and Discussion

### 3.1 Literature perspective of global ethane trends

Many studies have reported ethane trend analysis based on either ground-based sampling or FTIR (Fourier Transform Infrared Spectrometer) measurements. A summary of these studies is shown in Table 2. In the troposphere (Table 2(a)), the trends of $C_2H_6$ partial column at four European sites (Jungfraujoch, Zugspitze, Harestua and Kiruna) during 1996-2006 were between about -1.09 to -2.11%/yr (Angelbratt et al., 2011). Simpson et al. (2012) concluded a strong global ethane decline of 21% over 26 years (1984-2010), with a stronger decline occurring from 1984 to 1999 (-7.2 ± 1.7 ppt/yr) than from 2000 to 2010 (-1.9 ± 1.3 ppt/yr). Franco et al. (2015) showed the ethane trend at Jungfraujoch to be -0.92%/yr during 1994-2008, followed by a strong positive trend of 4.9%/yr during 2009-2014, which may be related to the emissions from shale gas exploitation in North America. Helmig et al. (2016) calculated a mean ethane growth rate of 2.9-4.7%/yr from 2009 to 2014 at 32 NH ground measurement sites and concluded that North American oil and gas development was the primary source of the increasing emission of ethane. Franco et al. (2016) compared the ethane total column change at six sites across NH for the period of 2003-2008 and 2009-2014, and also revealed a sharp increase of 3-5%/yr during 2009-2014 compared with 2003-2008, which was associated with oil and gas industry emission. They also specifically estimated a 1.2 Tg/yr increase of anthropogenic ethane emission from North America between 2008-2014. Hausmann et al. (2016) presented a positive ethane trend of ca. 4.6%/yr at Zugspitze (47° N) and a negative trend of ca. -2.5%/yr at Lauder (45° S) for 2007-2014, and inferred an ethane increase from oil and gas emission of 1-11 Tg/yr for 2007-2014. Angot et al. (2021) showed an increase in ethane trend of ca. 5.6%/yr at GEOSummit (73°N) for 2010-2014, followed by a temporary pause of ethane growth in 2015-2018. Sun et al. (2021) presented a negative ethane trend of -2.6 ± 1.3%/yr over 2015-2020 in a densely populated eastern Chinese city Hefei. In this study, we

estimated an increasing NH upper tropospheric ethane trend of 0.33 ± 0.27%/yr (mean ± 1SD) between February 2006 and February 2016 (relative to February 2006, thereafter same).

In contrast to tropospheric ethane trends, trends in the stratosphere have been far less investigated. Gardiner et al. (2008) (Table 2 (b)) presented annual trend in stratospheric ethane column (relative to year 2000) at six sites and these varied from 0.43 to -3.31%/yr until the year 2005. Franco et al. (2015) reported ethane trends at 8-16 km measured at Jungfraujoch of -1.75 ± 1.30%/yr for 2004-2008 and 9.4 ± 3.2%/yr for 2009-2013, indicating an ~11% sharp increase since 2009. Helmig et al. (2016) showed that the UTLS column ethane (8-21km) measured at Jungfraujoch was decreasing at -1.0 ± 0.2%/yr (1995-2009) and started a sharp increase at rate of 6.0 ± 1.1%/yr from 2009 until 2015, while the difference in trend growth rate between the two time periods is smaller for the mid-tropospheric column (3.6-8 km): -0.8 ± 0.3%/yr (1995-2009) and 4.2 ± 1.0%/yr (2009-2015). In this study, we derived a NH lowermost stratospheric ethane decreasing trend of -3.6 ± 0.3%/yr for the period February 2006 – February 2016.

It is noted that our aircraft samples have significantly different spatial distributions compared with the studies summarized above, any comparison should be made in a careful manner. When comparing surface and airborne datasets from multiple locations to assess global atmospheric changes, it will become increasingly important to ensure comparability of data quality, a process that has begun through the grounding of a World Calibration Center for VOCs, although this dataset predates this initiative.

## 3.2 Overview of IAGOS-CARIBC observations

In total 6,607 Northern Hemispheric samples were collected during Feb 2006-Feb 2016. 51% of them (3,365 samples) are identified as upper tropospheric samples (PV<2 PVU), the rest 49% (3,242) samples are stratospheric samples (Figure 1). All samples are categorized into four groups based on their sampling locations: North America (NAM), Asia (ASI), Europe (EUR), and Rest of the world (ROW) (Figure 1, Table S1). Temporal and spatial distributions of sample numbers are shown in Figure S2.

The observed upper tropospheric ethane mole fraction shows clear seasonality (Figure 2) driven by the atmospheric hydroxyl radical (OH) cycle and emissions. Upper tropospheric NAM and EUR ethane mole fractions increase from October/November peaking in April, decrease from April until October. This is consistent with the FTIR observation (Franco et al., 2015). Upper tropospheric ASI ethane peaks in June, two months later than NAM and EUR, and has two smaller peaks in October and February. Ethane mole fraction shows a stronger and different seasonality in EUR compared to the other regions. One possible explanation for this is a weaker influence by the in-mixing of stratospheric air over EUR. In contrast, the stratospheric ethane mole fractions do not show strong seasonality, except that NAM has a seasonal trend with 3-month later shift compared to the upper tropospheric NAM trend, and stratospheric ASI ethane shows the same timing peak in June with upper tropospheric ASI ethane which potentially indicates the intrusion of tropospheric air masses into the stratosphere due to Asian summer monsoon (Xiong et al., 2009; Park et al., 2007). There is little seasonality evident in the ethane mole fractions in the stratosphere. Since stratospheric aircraft measurement campaigns are generally of short duration (several weeks), a direct comparison to previous data is not possible, however, vertical column data obtained by ground based FTIR for 8-21km reported by Helmig et al. (2016) also showed no clear seasonal variation.

## 3.3 Tropospheric trends

### 3.3.1   Upper tropospheric observation vs. model simulation

The upper tropospheric ethane trends (Figure 3) and corresponding uncertainties (Figure S3 (a)) from the observations, the model and model optimizations (section 2.2), the top 5 contributing model sectors, and the climatology are shown in Figures 3, S3 and section 2.2, correspondingly.

As the air samples were not collected in exactly the same positions (e.g. altitude, latitude, longitude), the observed trends of trace gases could be potentially influenced by biases between the sampling locations. In order to assess whether a sampling location bias is associated with the derived trend, the measured trends were compared to results from a global model (EMAC) where the modeled data were extracted at the nearest grid of latitude, longitude, altitude, and time to the original measurement. Figure 3 (a) (grey line) shows the upper tropospheric ethane trend from the

EMAC simulation with constant meteorology and constant year-to-year emission with seasonal cycle (climatology). Thus if a trend is indicated from the model data, then it is expected to be associated with the sampling location rather than a real underlying trend. Although small variations of the ethane trend are observed due to the sampling location, these are negligible compared to the trend derived from the observations, implying that the different spatio-temporal sampling locations

did not influence the estimated trends.

We then focus on the ethane trends in the whole NH upper troposphere, and in addition, three regions: NAM, EUR, and ASI, whose emissions are estimated to be the dominant sources of global ethane emissions, accounting for 58-63% in 2008 (Monks et al., 2018). A clear increasing trend in ethane between Feb 2006-May 2010 of 19.2%/yr (±4.8, 1SD) relative to Feb 2006 and a decreasing

trend in May 2010-Feb 2016 of 7.5%/yr (±1.1) relative to May 2010 were observed for the upper troposphere (Figure 3 (a)). Such trend patterns are observed for all three regions of interest (NAM, ASI, EUR in Figure 3 (b)(c)(d)). Interestingly they are the inverse of the trends observed at the surface stations: a decreasing trend before 2009 and a sharp increase in 2009-2014 (Simpson et al., 2012; Franco et al., 2015; Franco et al., 2016; Helmig et al., 2016). To understand the driving

factors behind the observed trends, we simulated the ethane mole fractions with the atmospheric model (EMAC) for the IAGOS-CARIBIC samples (see section 2.2).

The trends from the model simulations and the optimized model results (increasing the input model emissions by 45%) are shown in Figure 3 as red and blue lines. The initial model results underestimate ethane mole fractions by about 45%, whereas the model estimation is closer to

observation for methane with the same model and observation dataset (Zimmermann et al., 2020). The model incorporates all known emissions via emission inventories so any deviations between model and measurements can be interpreted as indicators of hitherto unknown emissions or sinks, atmospheric processes, or errors in emission inventories. The optimized model results match reasonably well with the measured NH upper tropospheric trend (Figure 3 (a)). However, this is

not the case for the regional scales. A significant discrepancy between model and observation for NAM and ASI appears in 2010-2011 (Figure 3 (b)(c)). As the model includes fixed emissions or emissions with prescribed changes, such an abrupt increase in the ethane trend for NAM and ASI in 2010-2011 is presumably due to a short-term additional source that generated a large-scale ethane plume. The model simulates an inverse trend compared to the observed trend for EUR

(Figure 3 (d)), although the CAMS-GLOB-ANT dataset has already included emission inventories for some major European cities (Guevara et al., 2021).

The top 5 contributing model sectors for ethane source trends are FEF (fugitives), RES (residential energy use), TRO (road transportation), SWD (solid waste and waste water), and BIB (biomass burning), and their optimized trends are shown in Figure 3. Interestingly the FEF opt contribution
is comparable to RES opt, which highlights the importance of fugitive emissions to the global ethane budget as has been previously noted by Helmig et al. (2016). The pronounced peak in 2010-2011 for the modeled NH upper tropospheric ethane is related to the increase in FEF, RES, and SWD, and the decreasing trend in 2011-2013 can be explained by the decrease in FEF, RES, and BIB (Figure 3 (a)). SWD and TRO contributed most to the trends in NAM, ASI, and EUR, while
FEF, BIB, and RES have similar contribution (Figure 3 (b)(c)(d)). We note that TRO, SWD, and other sectors listed in Table 1 are modeled results.

Figure S4 shows the modeled sectoral contribution to regional and global ethane trends. The width of flow is proportional to the quantity of sectoral contribution. Our model results estimated the average contribution of biogenic (BIO), biomass burning (BIB), and anthropogenic sources (sum
of all other sectors) to the NH upper tropospheric ethane in 2006-2016 are 9%, 16%, and 75%, respectively. This matches the estimated ~4%, 18%, and 78%, respectively, from Helmig et al. (2016). The contribution of the top anthropogenic sources to upper tropospheric ethane are TRO (28.7%), SWD (21.7%), FEF (14.0%), RES (6.0%), AWB (1.7%), and ENE (1.1%). Detailed relative contributions of each sector are shown in Table S2. The contribution of TRO from this
study is more than that of ~10% estimated by Warneke et al. (2012); Peischl et al. (2013); Wunch et al. (2016).

### 3.3.2   Model geographical sector contribution

Four geographical sectors, i.e. ASI, NAM, EUR, and ROW were included to investigate the origin
of the ethane emissions (Figure 4, Figure S5). Geographical sectors refer to the regions where the emissions came from, whereas "geographical regions" (Table S1) refer to the locations where the aircraft samples were collected. Ethane emission from ASI dominates the trends for the whole NH upper troposphere, NAM, ASI, and EUR, contributing 35%-60%, 40%-60%, 60%-70%, and 37%-

47%, respectively for 2006-2016. Ethane emissions from ROW contributes 15%-40% to the overall ethane trends in the upper troposphere. Emissions from EUR and NAM are the least contributors with each only 5%-25% contribution to ethane trends. Large contributions of ethane emissions from ASI to other regions indicated that our air samples collected at ~10 km were originated from a large spatial scale, and thus the observed ethane trends should not be interpreted as local emissions.

### 3.3.3 Upper tropospheric ethane, methane, and propane trend comparison

Methane and propane share emission sources with ethane, including fossil fuel extraction, transport, and use, especially related to oil and natural gas (Helmig et al., 2016; Dalsøren et al., 2018; Bourtsoukidis et al., 2020; Zimmermann et al., 2020). Further, these three compounds share the same major sink in the atmosphere: oxidation by OH radical.

In 2006-2016, NH upper tropospheric ethane has a total change of 18.1 (mean) [min, max: -27.2, 29.4] ppt from observation, that corresponds to a linear trend (described in section 2.3) of 1.8 [-2.7, 2.9] ppt/yr, and 0.33 [-0.45, 0.55]%/yr relative to 2006. The observed NH upper tropospheric methane increases in total 63.2 [62.7, 63.6] ppb, corresponding to a linear trend of 6.3 [6.3, 6.4] ppb/yr (3.5 [3.5, 3.6] ‰/yr relative to 2006). In the same period, the observed NH upper tropospheric propane increases in total 7.0 [-7.3, 11.1] ppt, representing a linear trend of 0.70 [-0.73, 1.11] ppt/yr (1.02 [-0.82, 1.72]%/yr relative to 2006). Zhang et al. (2011) presented a ~3 ‰/yr increase of upper tropospheric methane at 206 hPa over China from 2006-2008 using satellite observations, which matches the methane trend from our study.

For the whole NH upper troposphere, ethane and propane have similar trends in 2006-2016 (i.e. a rise and then a fall), whilst the observed methane trend follows an increase throughout that period (Figure 5). A common peak of all three compounds appears in 2010-2011, which possibly indicates an abrupt increase in oil and gas emissions. This peak is also observed for ASI, EUR, ROW and NAM (not for NAM methane) (Figures S6, S7, S8, S9), suggesting regional and global increase in fossil fuel emissions. The contribution of OH radical variation to the peak in 2010-2011 is expected to be small as several previous studies have shown the atmospheric OH concentration did not change significantly in that period (Rigby et al., 2017; Li et al., 2018; Ipcc, 2013; Montzka et al.,

2011). NAM ethane and propane trends from the middle of 2014 to 2016 show a clear decline, probably due to a slowdown in U.S. natural gas emissions (Angot et al., 2021).


### 3.3.4    Ethane emission estimation

Observations of surface ethane mixing ratios at two ground stations (Mauna Loa (MLO), and Hohenpeissenberg (HPB)) were compared with model simulations using the optimized emissions from this study (Figure S10). It is noted that the good agreement between two ground station

observations and model simulations does not grant the accuracy of our model, further model results for ground level ethane should be studied in the future.

The global ethane emission was estimated to be 19.3 Tg/yr for February 2006 to February 2016, with biogenic emissions 0.8 Tg/yr, biomass burning 1.5 Tg/yr, and anthropogenic emissions 17.1 Tg/yr (Table 1). This estimate matches well with the estimated ethane emissions from other studies,

e.g. 18.2 Tg/yr for 2014 from Franco et al. (2016) and somewhat higher than the 15.3 Tg/yr (anthropogenic emission) for 2014 from Pozzer et al. (2020).

## 3.4 Stratospheric trends

### 3.4.1    Observation vs. model simulation

While ground based stations will be affected by upwind sources, the stratospheric samples offer a remote and averaged global perspective. Stratospheric ethane trends, estimated with all the IAGOS-CARIBIC samples taken in the NH lowermost stratosphere with PV larger than 2 PVU during 2006-2016, along with modeled stratospheric trends, are shown in Figure 6 (corresponding uncertainties in Figure S3 (b)). The variation of the stratospheric climatology (Figure 6 (a))

indicates the sampling location bias for the observed stratospheric ethane trend. It varies more than the tropospheric one (Figure 3), but it is again a minor contribution, so that location biased trends can be discounted. The observed stratospheric ethane over the whole NH shows a general trend of -3.6 (±0.3)%/yr in 2006-2016, with two exceptional peaks in 2010 and 2013. The peak in 2010 is not seen at regional levels (NAM, ASI, EUR, Figure 6 (b)(c)(d)), which suggests global upward

transport of the upper tropospheric ethane (peaking in 2010-2011) into the stratosphere and the important contribution from ROW. The second peak in 2013 can be due to the regional emission transport into the lowermost stratosphere as such a peak is observed simultaneously over NAM and ASI. In general, the optimized model trend matches well with the observed NH stratospheric trend in 2006-2013 (Figure 6 (a)). A noticeable discrepancy between the optimized model simulation

and observation appears since 2013. In the stratosphere, the OH radical concentration on average decreases by a factor of 10 compared with tropospheric OH levels, whereas Cl radicals are more abundant and therefore plays a greater relative role in ethane oxidation (Li et al., 2018). The loss of ethane in the stratosphere by reaction with Cl radicals is about 40 times more than that by OH radicals (reaction rate of ethane with Cl is about 400 times faster than with OH at 250K (Atkinson

et al., 2001), and stratospheric OH is about ten times more abundant than stratospheric Cl (Li et al., 2018)), whereas the ethane loss in the troposphere by Cl is negligible compared with by OH due to the small amounts of tropospheric Cl (OH:Cl around 10,000) (Lelieveld et al., 1999; Gromov et al., 2018)). The chlorine chemistry is not included in our model but the abundance of chlorine in the stratosphere is a significant loss factor for ethane, thus part of the observed discrepancy can

come from the missing chlorine chemistry in the model. After 2013, the model prediction for ASI was far from observation (Figure 6(c)), but this was not the case for other regions. Previous studies have shown that the global and Asian emissions of some chlorinated trace gases (e.g. CFC-11, $CHCl_3$) were increasing during 2012-2016 (Rigby et al., 2019; Fang et al., 2019; Montzka et al., 2021), and strong chlorine chemistry was associated with Asian outflow in the UTLS region in

2013 (Baker et al., 2016).This could be an explanation for the larger discrepancy between model and observation since 2013 in ASI.

The top 5 contributing model sectors for stratospheric ethane trends, at global and regional scales, are TRO (~28%), SWD (~24%), BIB (~15%), FEF (~13%), and RES (7%) (Figure 3, Table S2), their optimized trends are shown in Figure 6.

Model geographical sector contributions for the stratospheric ethane trends are shown in Figure 7 and Figure S11. Similar to the upper troposphere, ASI ethane emissions contribute the most to the global and regional stratospheric ethane trends (~50%). We attribute this to the Asian Monsoon transport of air pollutants from the troposphere to the stratosphere, which is supported by other studies (Lelieveld et al., 2018; Randel et al., 2010; Park et al., 2009; Lelieveld et al., 2002; Bian et

al., 2020).  Ethane emissions from ROW contributes 20-25%, and EUR and NAM 10-20% each.

### 3.4.2 Stratospheric ethane, methane, and propane trend comparison

Figure 8 shows the observed stratospheric trends of ethane, methane, and propane in 2006-2016. The observed NH stratospheric ethane has a total change of -191.3 [-221.2, -166.7] ppt corresponding to a linear trend of -19.1 [-22.1, -16.7] ppt/yr, and -3.6 [-4.15, 3.20] %/yr relative to 2006. The observed methane in the NH stratosphere increases in total 36.9 [34.5, 38.0] ppb, that represents a linear trend of 3.7 [3.45, 3.80] ppb/yr (2.1 [2.0, 2.2] ‰/yr) relative to 2006. In the same period, the observed NH stratospheric propane declined in total 52.2 [51.3, 55.7] ppt, that corresponds to a linear trend of -5.2 [-5.6, -5.1] ppt/yr (-5.6 [-6.1, -5.5] %/yr) relative to 2006. Rohs et al. (2006) derived an increase in stratospheric methane (~30km) of ~5 ‰/yr using balloon-born observations for 1978-2003, and Rinsland et al. (2009) presented a larger increase (~8 ‰/yr) for the lower stratosphere in 1985-2008. The regional trends of ethane, propane, and methane at NAM, ASI, EUR and ROW are shown in Figures S12-S15.

Similar to the upper tropospheric trends, ethane and propane shared similar trends in the NH stratosphere, NAM, and EUR. The 2010-2011 peak observed in the upper troposphere also appears in the stratosphere, indicating a strong influence of troposphere-stratosphere exchange. It is noted that the observed stratospheric trends on regional scales represent a mixture of local emission and global atmospheric transport.

## 3.5 Limitations and implications

Despite the usefulness, uniqueness and high quality of our datasets, several limitations of our study should be noted. (a) representativeness of the presented trends. Although our flight sampling is frequent and covers a large area of the NH, the spatial and temporal distributions of our samples are not even. This may cause the trends being influenced by specific regions where more samples were collected. (b) chlorine chemistry is missing in the EMAC model. Chlorine radicals are much more abundant in the stratosphere than the surface, thus the change in chlorine plays a great role in the observed trends. (c) our samples were collected in the UTLS region and can be influenced by atmospheric transport (e.g. troposphere-stratosphere exchange), surface sources, and chemical destruction processes. Therefore, the trends represent the net effects of these factors making the interpretation on a single factor difficult. (d) PV choice of identifying

upper tropospheric and stratospheric samples. In this study, we used PV=2 to define the tropopause, whereas other approaches exist. It is shown that on large space and time scales in the extratropics, the WMO tropopause corresponds rather well to a surface of constant potential vorticity (PV), although there exist systematic differences on smaller scales (Stohl et al., 2003; Wirth, 2000). (e) trend analysis tool "Prophet". One needs some experience with the algorithm to
choose and tune some parameters to get the best results for individual datasets, i.e. settings for our dataset may not be suitable for other datasets. (f) model optimization. Our EMAC model and input values for sectorial emissions have been examined and optimized in many previous studies, therefore, in this study we simply increased each emission sector by 45% to match the observations. The aim of model simulations is to better understand the contributions from each
emission sector, rather than improving the performance of model and emission inventories. (g) interpretation of results. This article is designed as a data description article to provide high quality and useful dataset for scientific use. There are many interesting features in the presented trends to be explored, however, it is beyond the aim of this study.

**Implications.** (a) observations of ethane, methane and propane were often restricted at regional
scale or short-duration. We have presented a long-term (10 years) airborne observations of ethane, methane and propane in the UTLS region at northern hemispheric scale. This dataset is unique and can be used to examine long-term troposphere-stratosphere exchange, chemical and dynamical changes in the UTLS region, and improve model performance. To the best of our knowledge, such long-term aircraft observations are only available from IAGOS-CARIBIC
project (our study) and CONTRAIL project (Machida et al., 2008; Sawa et al., 2015).(b) The "Prophet" algorithm is an open source software, and suitable for non-continuous time-series datasets. Unlike the commonly used linear fit approach for trend analysis in other studies, the "Prophet" algorithm is robust to missing data and the influence from outliers is minimized. It better captures the inter-annual variability and is not influenced by the time period of choice. (c)
other analysis approaches such as machine learning techniques can be used on our dataset to enlarge the spatial and temporal distributions. Combining our dataset with space-borne observations will provide a better view of global distributions and trends of trace gases.

# 3 Data availability

The NOAA ethane ground station data can be downloaded from NOAA website (https://gml.noaa.gov/). The IAGOS-CARIBIC observational data of ethane, methane, and propane in the period February 2006 – February 2016, and optimized ethane mixing ratios in sectors from EMAC model simulation for the same IAGOS-CARIBIC samples and time period, can be accessed at https://doi.org/10.5281/zenodo.6301729 (Li et al., 2021). Co-authorship may be appropriate if

the IAGOS-CARIBIC data are essential for a result or conclusion of a publication.

# 4 Conclusions

In this study, we present upper tropospheric and lower stratospheric ethane trends from airborne observations and atmospheric modeling over the period 2006-2016. The model performance was

optimized by scaling to the observational data. We identified ethane sectoral sources to which observed average trends over ten years (2006-2016) and three continents (North America, Europe, and Asia) could be attributed from observation and modeling. Trends of ethane, propane, and methane from observation were compared to identify ethane emission sources. The major findings are summarized as follows:

- The global ethane emission budget for February 2006 to February 2016 was estimated to be 19.3 Tg/yr. In the Northern Hemisphere, the upper tropospheric ethane had an increasing trend of $0.33 \pm 0.27\%$/yr and the stratospheric ethane had a decreasing trend of $-3.6 \pm 0.3\%$/yr for 2006-2016. The current inventory from CAMS-GLOB-ANT v4.2 underestimates ethane emission by roughly a factor of three.

- The top five contributing model sectors for upper tropospheric and stratospheric ethane trends are FEF (fugitives), RES (residential energy use), TRO (road transportation), SWD (solid waste and waste water), and BIB (biomass burning). Emissions from Asia dominate the observed ethane trends for both upper troposphere and lower stratosphere.

- A sharp increase in the observed upper tropospheric and stratospheric ethane at global

and regional scales in 2010-2011 was caused by fossil fuel related emissions, likely from oil associated and natural gas sources. In contrast to methane, the global ethane trends cannot be well simulated by advanced atmospheric chemistry modeling, which suggests

the need of accurate and frequent observations of global ethane and the improvement of emission inventories.


**Author contribution**

M.L. and J.W. developed the idea of this study. M.L. wrote the first draft of the manuscript. A.P. run the model simulations. All authors contributed to discussing and revising the manuscript.

**Competing interests**

The authors declare no conflict of interests.

**Acknowledgment**

We are thankful to Sourangsu Chowdhury for preparing the model emission input data into

different regions, and Nils Noll for providing biomass burning emission budget for ethane. We thank Tobias Sattler for contributing to the initial idea of this study. We thank NOAA for sharing ground station data of ethane. We thank Python, Esri and Figdraw for providing statistical and plotting tools. We thank the editor Nellie Elguindi and three anonymous reviewers.

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

# Figures and Tables


Table 1. Sectoral description and ethane emissions estimated from this study for Feb 2006-Feb 20016.

| Sector | Description | Emission from inventory (Tg/yr) | Estimated Emission (optimized)(Tg/yr) |
|---|---|---|---|
| BIO | Biogenic emission | 0.54 | 0.78 |
| BIB | Biomass burning | 1.01 | 1.46 |
| (a) Anthropogenic by sector | | | |
| AWB | Agricultural waste burning | 0.08 | 0.12 |
| ENE | Power generation (power and heat plants, refineries, others) | 0.04 | 0.06 |
| FEF | Fugitives | 5.28 | 7.65 |
| IND | Industrial processes | 0.90 | 1.30 |
| RES | Residential energy use | 3.32 | 4.82 |
| SHP | Ships | 0.02 | 0.03 |
| SLV | Solvents | 0.00 | 0.00 |
| SWD | Solid waste and waste water | 1.01 | 1.47 |
| TNR | Off-road transportation | 0.01 | 0.02 |
| TRO | Road transportation | 1.10 | 1.59 |
| | | | |
| (b) Anthropogenic by geographical sector | | | |
| ASI | Emission from Asia | 5.16 | 7.48 |
| EUR | Emission from Europe | 1.60 | 2.32 |
| NAM | Emission from North America | 1.01 | 1.46 |
| ROW | Emission from rest of the world | 3.99 | 5.79 |
| | | | |
| Total source | | 13.30 | 19.28 |

Table 2. Summary of studies reporting ethane trends in the (a) troposphere and (b) stratosphere.

| Trends (%/year) | Time period | References |
|---|---|---|
| **(a) Tropospheric trends** | | |
| -1.09 ~ -2.11 | 1996-2006 | Angelbratt et al. (2011) |
| (four European sites) | | |
| -0.81 (global) | 1986-2010 | Simpson et al. (2012) |
| -0.92 (Jungfraujoch, 47° N) | 1994-2008 | Franco et al. (2015) |
| 4.9 (Jungfraujoch, 47° N) | 2009-2014 | Franco et al. (2015) |
| 2.9-4.7 (32 ground sites) | 2009-2014 | Helmig et al. (2016) |
| 3-5 (six sites) | 2009-2014 | Franco et al. (2016) |
| | compared with 2003-2008 | |
| ca. 4.6 (Zugspitze, 47° N) | 2007-2014 | Hausmann et al. (2016) |
| ca. -2.5 (Lauder, 45° S) | 2007-2014 | Hausmann et al. (2016) |
| ca. 5.6 (GEOSummit, 73° N) | 01.2010-12.2014 | Angot et al. (2021) |
| -2.6 ± 1.34 (Hefei, 32° N) | 2015-2020 | Sun et al. (2021) |
| -0.47 ~ -1.30 (mean: -1.13) | 2007-2014 | This study |
| 0.33 ± 0.27 | 02.2006-02.2016 | This study |
| (Northern Hemispheric upper troposphere) | | |
| **(b) Stratospheric trends** | | |
| -3.31 ~ 0.43 | 2000-2005 | Gardiner et al. (2008) |
| (stratospheric column) | | |
| -1.75 ± 1.30 | 2004-2008 | Franco et al. (2015) |
| (8-16km above Jungfraujoch) | | |
| -1.0 ± 0.2 | 1995-2009 | Helmig et al. (2016) |
| (8-21km above Jungfraujoch) | | |
| 9.4 ± 3.2 | 2009-2013 | Franco et al. (2015) |
| (8-16km above Jungfraujoch) | | |
| 6.0 ± 1.1 | 2009-2015 | Helmig et al. (2016) |
| (8-21km above Jungfraujoch) | | |
| -3.6 ± 0.3 | 02.2006-02.2016 | This study |
| (Northern Hemispheric lowermost stratosphere) | | |


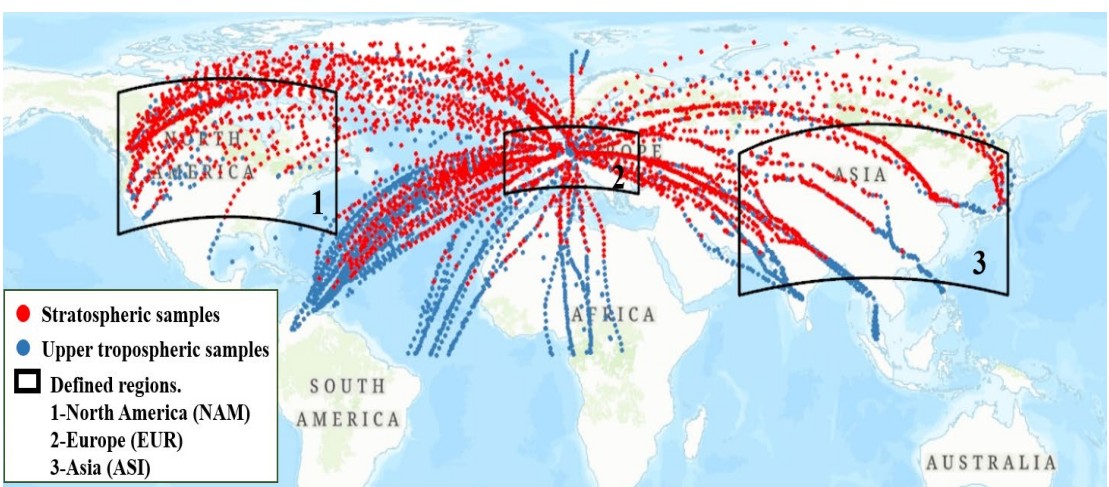

Figure 1. Geographical locations of aircraft samples (distinguished as upper tropospheric samples and stratospheric samples) and spatial segregation. Samples collected outside the black boxes are defined as ROW samples.

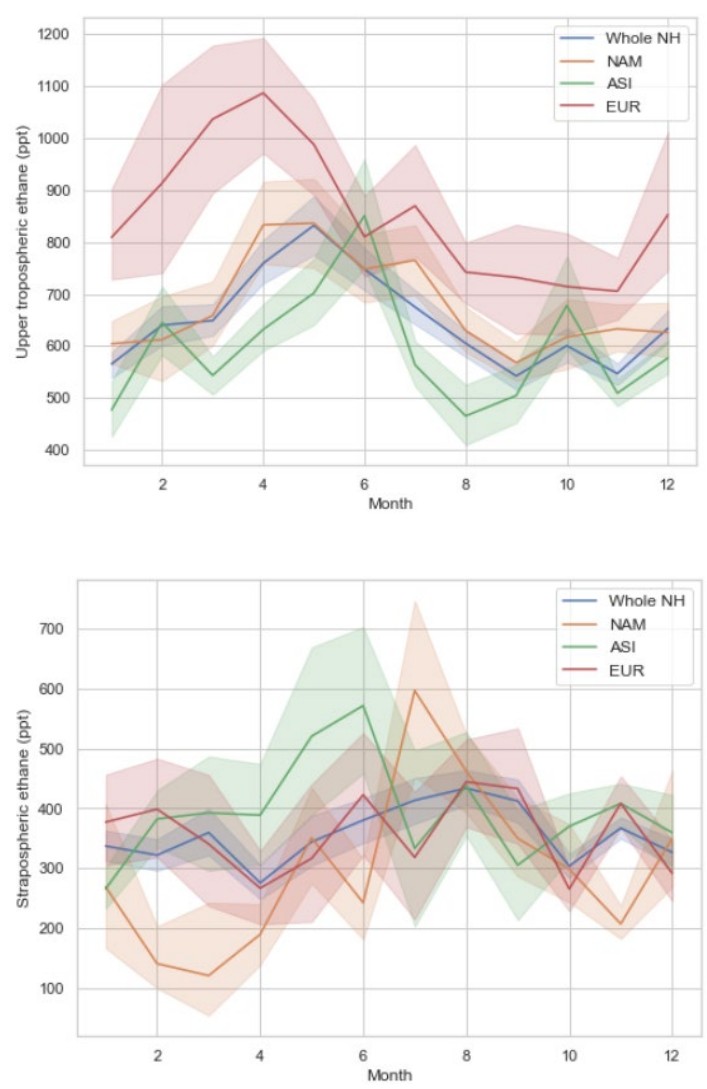

Figure 2. Seasonality of upper tropospheric and stratospheric ethane mole fractions over the whole
Northern Hemisphere (Whole NH), North America (NAM), Asia (ASI), and Europe (EUR).


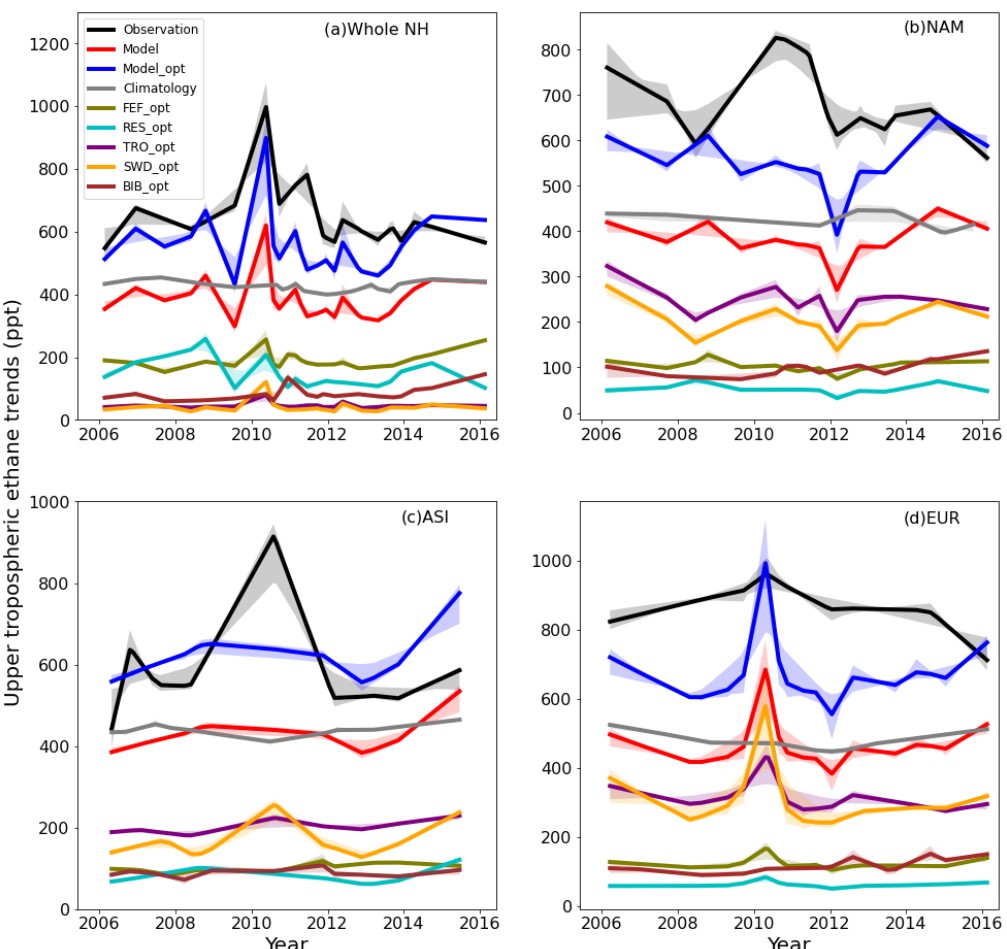

Figure 3. Upper tropospheric ethane trends from observations, the model (Model: sum of all sectoral emissions) and model optimization (Model_opt: sum of all optimized sectoral emissions; sector abbreviation ends with "opt": individual optimized sectoral emission), and climatology for (a) the whole NH; (b) North America; (c) Asia; and (d) Europe. Light shadows indicate trend

analysis uncertainty.


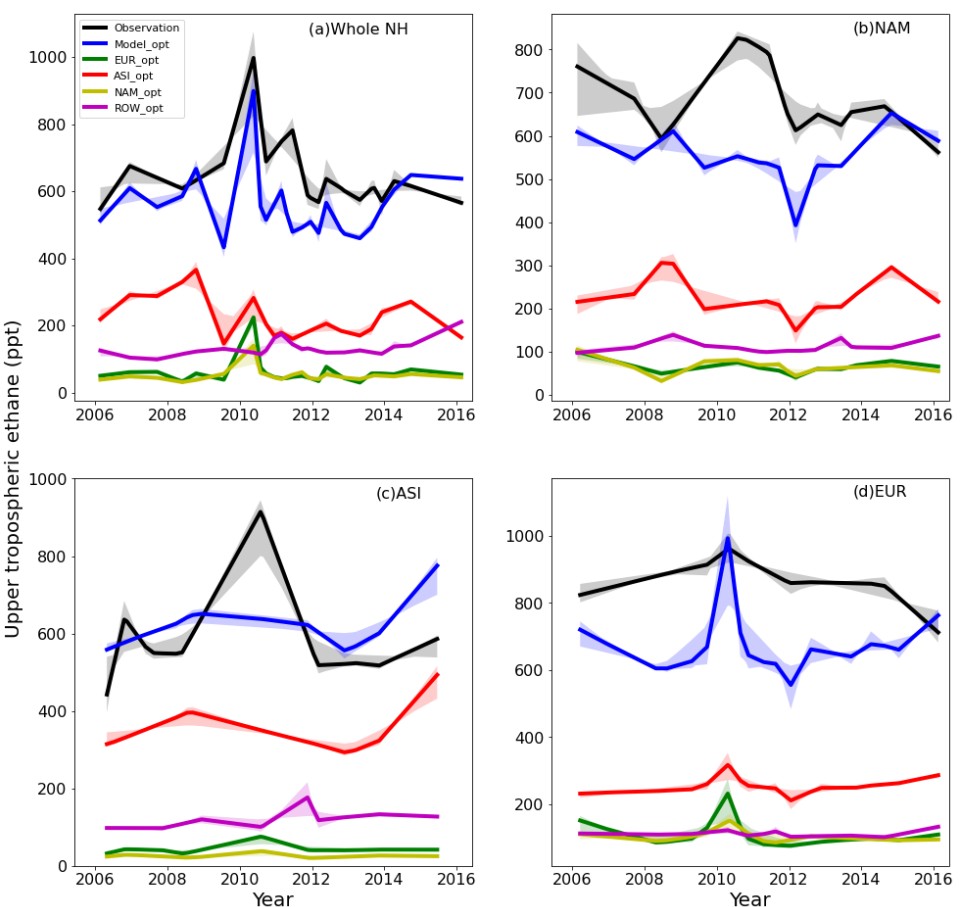

Figure 4. Observed trends and modeled optimized ("opt") geographical sector contribution (emissions originated from EUR, ASI, NAM, and ROW) to NH upper tropospheric ethane trends for (a) the whole NH; (b) North America; (c) Asia; and (d) Europe. Light shadows indicate trend

analysis uncertainty. "Model_opt" indicates the sum of all optimized emission sectors.

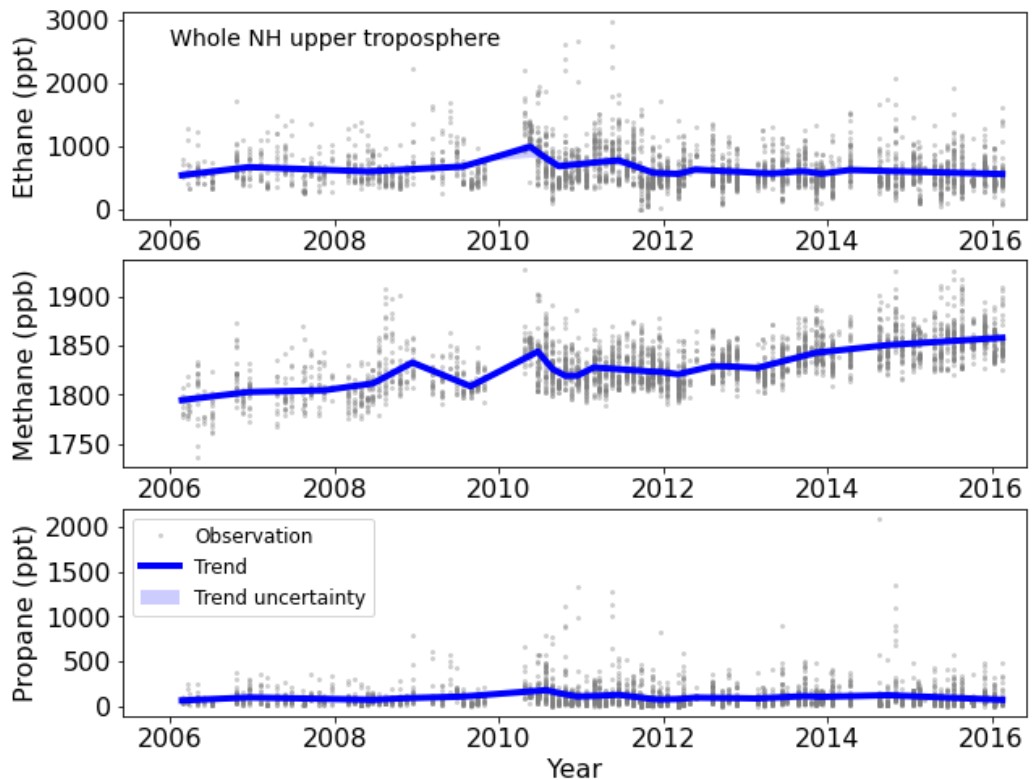


Figure 5. The observed ethane, methane and propane mole fractions (gray dots) and trends (blue lines) for the whole NH upper troposphere. Light shadows indicate trend analysis uncertainty.


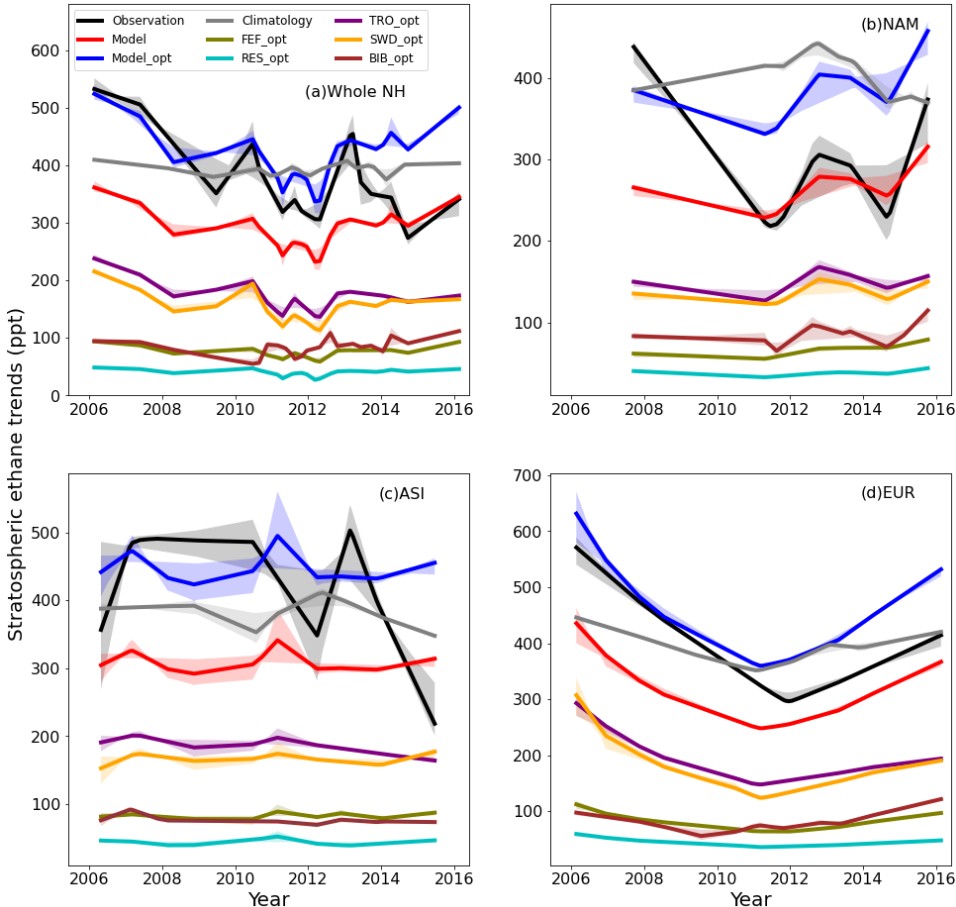

Figure 6. Stratospheric ethane trends from observation, model (Model: sum of all sectoral emissions) and model optimization (Model_opt: sum of all optimized sectoral emissions; sector abbreviation ends with "opt": individual optimized sectoral emission),  and climatology for (a) the whole NH stratosphere; (b) North America; (c) Asia; and (d) Europe. Light shadows indicate trend analysis uncertainty.

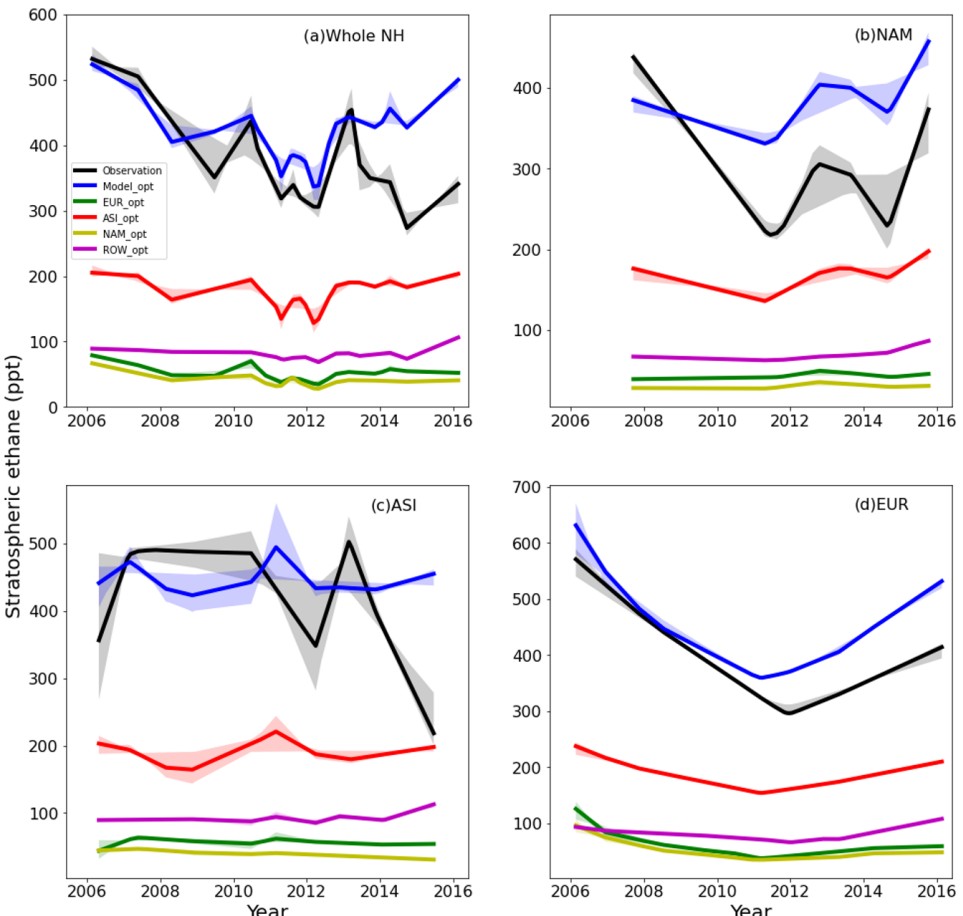

Figure 7. Observed trends and modeled optimized ("opt") geographical sector contribution
(emissions originated from EUR, ASI, NAM, and ROW) to stratospheric ethane trends for (a) the
whole NH stratosphere; (b) North America; (c) Asia; and (d) Europe. Light shadows indicate trend
analysis uncertainty. "Model_opt" indicates the sum of all optimized emission sectors.

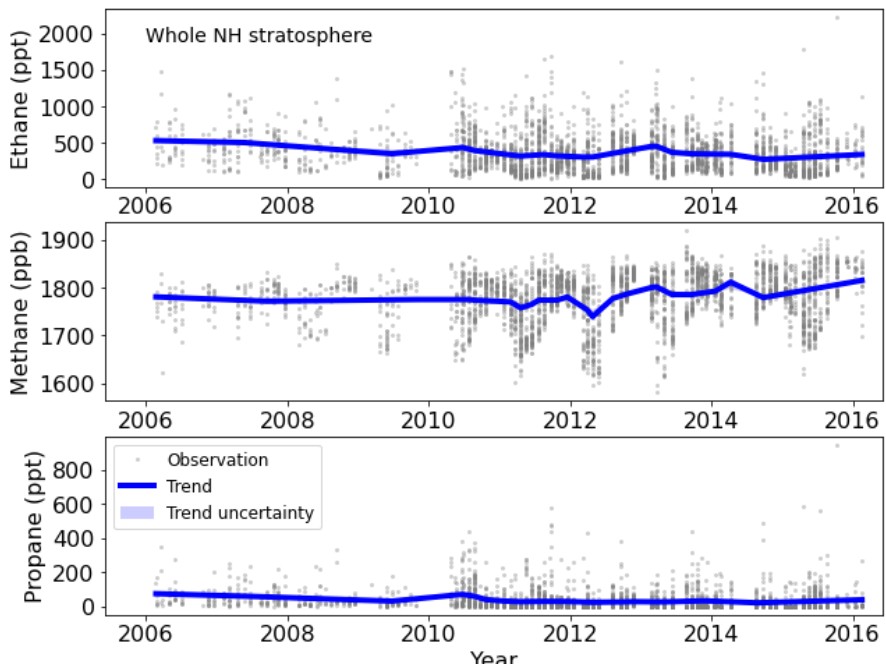

Figure 8. The observed ethane, methane and propane mole fractions (gray dots) and trends (blue

lines) for the whole NH stratosphere. Light shadows indicate trend analysis uncertainty.