# Peer review of "Northern hemispheric atmospheric ethane trends (2006-2016) with reference to methane and propane"

_Earth System Science Data, 2021_

## Author Comment (AC1)

**GENERAL COMMENTS**

In this study, Mengze et al. present an analysis of observed (airborne) vs modeled ethane, propane and methane trends for the period 2006-2016. They estimate a global emission of ethane of 19.28 Tg/yr. Their results show trends for upper tropospheric and stratospheric ethane, propane and methane.

The paper is generally well written and contributes to the scientific understanding of ethane, propane and methane trends globally. I recommend this paper for publication after major revisions.

My biggest concern is that authors jump into conclusions too fast in some sections. There is also a lack of information regarding the model simulation setup. There are multiple places where it is hard to distinguish if the authors are referring to modeled results or observations, making the reading a bit confusing. Also, almost all figure captions (in the manuscript and the supplemental material) should be improved by adding information regarding the legends, and the type of information shown (see specific comments below). Lastly, even though this study shows results for propane and methane, these compounds are barely discussed in the text and their results are not even mentioned in the abstract. I wonder if somehow the title should be changed to clarify the readers that ethane is the main compound discussed in the paper and just a few propane and methane results will be shown.

We thank the reviewer for the helpful comments, suggestions and discussion. We have addressed all points raised in the revised manuscript. The original comments from the reviewer are marked as bold in black, and our replies are in blue.

More information regarding model setup has been now added, and the wording of the figure captions improved. Details are given below.

Indeed the paper is focused primarily on ethane rather than methane and propane but the latter two molecules are used as less and more reactive comparators respectively. To make this focus clearer we have added the following sentence in the Abstract to clarify this point: "The model simulations, and methane and propane observations provide additional information for understanding northern hemispheric ethane trends and emissions, which is the primary focus of

this study." We have also changed the title to "Northern hemispheric atmospheric ethane trends (2006-2016) with reference to methane and propane"

The revised manuscript will be uploaded soon.

**SPECIFIC COMMENTS**

**Section 3.2**

1. **There is no comment on the most significant feature of Figure 1, which is the much stronger and different seasonality of ethane for EUR compared to the rest of the regions. Why are concentrations so high for this region and why the peak shows up 1-2 months prior compared to NAM and whole NH?**

The reviewer has raised many interesting scientific questions and discussions. According to the journal's author guidelines "Any interpretation of data is outside the scope of regular articles." We therefore limited our interpretation. Nevertheless, in answer to the point raised we now add more information and discussion.

Figure 1 shows the ethane mole fraction in the upper troposphere which can be influenced by both surface air and stratospheric air in the vertical, and by adjacent regions through advective transport. The observed upper tropospheric ethane concentration therefore doesn't necessarily reflect perfectly the surface emissions. One possible explanation for higher ethane over EUR is that at similar flight altitudes (~10km), upper tropospheric air over EUR is influenced less by the in-mixing of stratospheric air than over NAM or ASI (e.g. Asian monsoon, etc.). Support for this explanation can be seen in the stratospheric ethane concentration, ASI has higher ethane concentrations than EUR, probably due to more stratosphere-troposphere exchange. As our dataset, which is limited in time and space, doesn't provide enough information to confirm this, we have not speculated further on this point.

To make mention of this point we now add the following text:

"Ethane mole fractions show a stronger and different seasonality in EUR compared to the other regions. One possible explanation for this is relatively less influence by the in-mixing of stratospheric air over EUR."

2. **How does stratospheric ethane lack of general seasonality from this study compares to others?**

To the best of our knowledge, there is no study on stratospheric ethane seasonality based on in-situ aircraft measurements with which we can compare. Normally aircraft campaigns sampling in the stratosphere are only for short time periods and therefore do not cover a whole year or multiple years. Other observations of stratospheric ethane are based on FTIR techniques which report the total ethane column. Such studies have reported a lack of seasonality in that stratospheric ethane (column). For instance, Helmig et al., 2016 (cited in our manuscript) showed UTLS ethane column (8-21km) above Jungfraujoch exhibited little evidence of seasonality (see their Fig 1c).

We now add the following text to the manuscript to capture this point.

"There is little seasonality evident in the ethane mole fractions in the stratosphere. Since stratospheric aircraft measurement campaigns are generally of short duration (several weeks), a direct comparison to previous data is not possible, however, vertical column data obtained by ground based FTIR for 8-21km reported by Helmig et al., 2016 also showed no clear seasonal variation."

Reference:

Helmig, D., Rossabi, S., Hueber, J., Tans, P., Montzka, S. A., Masarie, K., Thoning, K., Plass-Duelmer, C., Claude, A., Carpenter, L. J., Lewis, A. C., Punjabi, S., Reimann, S., Vollmer, M. K., Steinbrecher, R., Hannigan, J. W., Emmons, L. K., Mahieu, E., Franco, B., Smale, D., and Pozzer, A.: Reversal of global atmospheric ethane and propane trends largely due to US oil and natural gas production, Nature Geoscience, 9, 490-495, 2016.

3. **Why does NAM show such seasonality? Any insights of the reasons behind it?**

Given that the data shown in Figure 1 are the average of 10 years thus the trends can be assumed to be robust seasonal patterns. In the case of NAM upper troposphere, the high springtime values in the Northern Hemisphere closely match that of the whole Northern Hemisphere so we may

interpret this as the hemispheric optimum between high emissions in winter and highest removal in summer. For the stratospheric NAM there is a peak in the summer months. This may be caused by the injection of ethane into the stratosphere from intense biomass burning episodes that occur periodically in summer generating pyrocumulus clouds that can inject emissions into the relatively low-lying stratosphere.

**Lines 219-221.**

1. **Figure S3 does not indicate the region that the uncertainties correspond to (e.g. NAM, EUR). If it is for the whole globe, it needs to be stated. Again, captions and descriptions in the text should be improved.**

We now add the corresponding regions in the descriptions of Figure S3 and improved the captions and descriptions of other figures.

2. **".. and model optimizations (section 2.2)". I couldn't find the description of the model optimizations in section 2.2. Check or add this information clearly. Use the same terms/words, so that is clear and easy for the reader to find this information. This is extremely important to better understand Figure 2.**

We now add more description of model optimizations in section 2.2, the text reads as follows:

"We further optimized modeled ethane mole fractions for each emission sector (indicated as "opt" in the later figures and "optimized" in the later texts). The model optimization is done by increasing the emissions of each input emission sector by 45%. We found that the root mean squared error (RMSE) between the modeled and observational ethane mole fractions for the whole dataset was the minimum by 45% increase in the input emissions."

**Line 222-224. Map with location and time frames of sample collection is needed.**

We now include the spatial segregation plot as a map in the manuscript showing the delineation of NAM, EUR, ASI and tropospheric, stratospheric air samples (see below). While it is not possible to display the time variable in this plot we make the data file including altitude, latitude,

longitude and time available at https://doi.org/10.5281/zenodo.6301729 and refer to it in the text. We also add the caveat that the region designated must not correspond to the source region, only the geographical location of the data points.

[Figure]

Figure 1. Geographical locations of aircraft samples (distinguished as upper tropospheric samples and stratospheric samples) and spatial segregation.

**Lines 224-225. How were these observations selected? Which are those observations selected?**

This sentence is indeed confusing. We meant that observations were selected or filtered by measurement location and regions of interest (NAM, EUR, ASI, ROW). We used all the available observations, none was removed any prior to analysis. To avoid confusion, we now delete this sentence.

**Figure 2 – Panel a) I am surprised by seen that the FEF opt contribution is almost similar to RES opt and how little BIB opt contributes from 2006-2014. Also, more information of model setup is needed. Specifically a table with emission inventories used, global and regional (NAM, EUR, etc) totals per sector will be helpful.**

We used CAMS-GLOB-ANT v4.2 emission inventory data for each model sector, and then applied the same optimization factor (1.45) to each sector in order to match the observation. Fugitive emission (FEF) has been shown in previous studies to be one of the major sources of global ethane (e.g. Helmig et al., 2016). Parts of oil and gas emission were included in FEF in

the emission inventory that we used in this study, and other parts were included in ENE. In this study, we use model simulations to validate the observational trends, and assume that the order of contribution to each sector from this inventory is correct or close to reality and only upscaling is required. Note that the trends from this study are determined from aircraft observations at ~10km height, where air samples do not only reflect surface emissions, therefore, the sectoral contribution from this study may not match exactly with the studies using ground measurements. We now add the following text in section 3.3.1:

"Interestingly the FEF opt contribution is comparable to RES opt, which highlights the importance of fugitive emissions to the global ethane budget as has been previously noted by (Helmig et al., 2016)."

Table 1 shows the estimated ethane emissions (Tg/yr) from this study for each sector and geographical region. We now add a column showing ethane emissions from the inventories used.

**Line 267.  Add the word "modeled" in "… the modeled NH upper tropospheric ethane…".**

Done.

**Line 269. State that SWD and TRO are modeled results.**

We have stated in the beginning of that paragraph with "…top 5 contributing model sectors … are … TRO (road transportation), SWD (solid waste and waste water)…".

Nonetheless we now add the sentence "We note that TRO, SWD, and other sectors listed in Table 1 are modeled results" for greater clarity.

**Lines 269. How can contributions of SWD and TRO be so high in NAM? Can these contributions be explained? Do these modeled results make sense? Also, how can SWD and TRO have such high sectoral contributions, but low contributions when considering trends from the optimized models (Figure 2).**

SWD and TRO were the sectors with large contribution in CAMS-GLOB-ANT v4.2 emission inventory (a prior assumption), we input these emissions in our model, and thus high

contributions of SWD and TRO are expected. We note this in the following paragraph with "The contribution of TRO from this study is more than that of ~10% estimated by Peischl et al. (2013); Warneke et al. (2012); Wunch et al. (2016)." No independent estimation of SWD contribution was found in the literature.

Figure 2 shows the optimized sectoral contributions of SWD and TRO, but not their contributions before optimization. We only apply a correction factor (1.45) for optimizing each sector.

**Line 271. Add the word "modeled" in "Figure 3 shows the modeledsectoral…"**

Done.

**Figure 3. Even though this figure looks really nice, it does not provide clear information on the contribution of each sector and region. The size of Whole NH, NAM, EUR, and ASI on the right side is the same for each region. On the left side, the sectors have different sizes, but there is no axis or value assigned to each contribution and also, for each sector the slice corresponding to each region is always the same. My suggestion is to change this figure for another one that shows the contributions by region as stated in lines 271-280. The way it is right now, those contributions are not clear.**

To improve the clarity of this figure we now move it to the Supplement and furnish more explanatory text. The right side of the figure shows the fractional contributions of all sectors for each region, assuming 100% for each region. This is why the sizes are the same. With this figure we are trying to show the relative contributions among sectors for each region. Unfortunately, we cannot derive emissions of each sector for troposphere and stratosphere separately from our model.

**Lines 283-284:**

1. **"Five geographical sectors" ?? Do you mean geographical regions? Explain what do you mean by geographical sectors because mixing regions with emission sources is confusing.**

We now add a map that shows the geographical regions and the spatial distributions of aircraft samples. We use the term "geographical sectors" to differentiate "geographical regions". We

refer "geographical regions" as the locations where the aircraft samples were collected, "geographical sectors" as the regions where the emissions came from. An example: for a sample collected at NAM (geographical region), 50% of its emissions may come from ASI (geographical sector), 30% from NAM (geographical sector), 20% from EUR (geographical sector). We now add more explanation in section 3.3.2 and it reads:

"Four geographical sectors, i.e. ASI, NAM, EUR, and ROWwere included to investigate the origin of the ethane emissions (Figure 4, Figure S5). Geographical sectors refer to the regions where the emissions came from, whereas "geographical regions" (Table S1) refer to the locations where the aircraft samples were collected."

**Why can't BIB and BIO emissions be separated by regions? Are the emissions not gridded in the model? Clarification is needed.**

In principle we could separate them. However, by separating BIB and BIO more uncertainty will be added. In addition biogenic emissions are very small (almost negligible). For biomass burning a different regional division is needed, based on bio-types rather than continent/region, and therefore would have made the analysis even more difficult and uncertain. Furthermore biomass burning is only ~20% or less of the total emissions for ethane and therefore we can consider them less important compared to the anthropogenic emissions. It was not accurate in the manuscript that "AIR+BIB+BIO (as they cannot be separated into regions)", we now change it to "AIR+BIB+BIO (combined as one sector to reduce uncertainty)"

We add the following text in section 2.2:

"It is noted that AIR, BIB and BIO were combined as one sector to reduce the uncertainty."

**Figure 4: Is Model Opt the sum of all the optimized geographical regions? The authors need to clarify this.**

We add the following in Figure 2 legend where "Model_opt" first appeared in the manuscript:

"Model_opt is the sum of the optimized model contributions listed in the corresponding figure."

**Lines 285-286.**

1. **"… 30%~55%, 35%~50%, 50%~65%, and 30%~40%...", the symbol "~" must be changed for "-".**

Done.

2. **Are these percentages average values from 2006-2016? Specify.**

Done.

3. **"Ethane emissions from ASI dominates the trends…" What are the model estimated ethane emissions (in Tg/yr) for ASI compared to the rest of the regions? A table with regional emissions used in the model is needed.**

In Table 1 (b) we have listed the estimated ethane emissions from different regions from this study (ASI: 7.48 Tg/yr, EUR: 2.32 Tg/yr, NAM: 1.46 Tg/yr).

**Lines 288-289. Clearly state to which atmospheric layer correspond these trends. For example: "…contribution to ethane trends in the upper troposphere and/or stratosphere".**

Done.

**Lines 295-301. How does these results compare with other studies?**

We have listed the comparison with other studies for ethane in Table 2. We have also extended the comparison with previous studies.

**Section 3.3.3 has "methane" in its title, but it is barely discussed in the text.**

As noted in the previous responses, the main focus of this paper is the northern hemispheric budget of ethane, which is in part elucidated by reference to methane and propane data collected in parallel. The title has been changed to reflect this more clearly. Nonetheless we now provide more details in the Method section about collection and measurement of methane. In this section 3.3.3 as well as section 3.4.2, we presented methane trends from four regions (whole northern hemisphere, NAM, ASI, and EUR) in the upper troposphere and stratosphere using high-quality dataset. Another peer-reviewed article by Zimmermann, et al., 2020 has used the same aircraft methane dataset and discussed methane more in detail in combination with atmospheric model.

Thus we want to avoid publishing overlap with this article. We now add some literature comparison for methane in the text.

Reference:

Zimmermann, P. H., Brenninkmeijer, C. A. M., Pozzer, A., Jöckel, P., Winterstein, F., Zahn, A., Houweling, S., and Lelieveld, J.: Model simulations of atmospheric methane (1997–2016) and their evaluation using NOAA and AGAGE surface and IAGOS-CARIBIC aircraft observations, Atmos. Chem. Phys., 20, 5787–5809, https://doi.org/10.5194/acp-20-5787-2020, 2020.

**Section 3.3.4**

1. **This section should be named differently because it discusses the comparison of the model to ONLY two observations. It's current name gives the false idea that there was a model simulation using ground station data. I suggest calling it: "Model results compared to two ground stations".**

Done. We have now removed this section and moved the figures into Supplementary.

2. **The conclusion from the last sentence should be erased. It cannot be concluded just by comparing 2 stations that the model provides realistic values for ethane surface level. A thorough analysis of multiple surface stations must be done before jumping to that conclusion.**

Done. We have deleted this sentence.

3. **I suggest completely getting rid of this section and mention that the use of this model results for surface-level ethane should be studied in the future.**

We have combined this section with section 3.3.5 and noted in the manuscript that further model results for surface ethane is needed in the future.

**Section 3.3.5**

1. **Erase the term "budget" in the title and text because this implies sources and sinks of a certain compound and here, only global emission totals are presented.**

Done.

2. **Thus, modify the title of this section.**

Done.

**Lines 334-335. How does the authors conclude that the stratosphere has a minor contribution for observed trends? Authors need to explain clearly why location biased trends can be discounted, even if it is for the same reasons that were discounted for the upper troposphere.**

We didn't discuss or conclude that the stratosphere has a minor contribution for observed trends in lines 334-335. We concluded that the location bias has a minor contribution to the observed trends. This point is now addressed more clearly in the following text:

"The stratospheric climatology (Figure 6 (a)) indicates the contribution from sampling location for the observed stratospheric ethane trend. It varies more than the tropospheric one, but it is again a minor contribution, so that location biased trends can be discounted."

**Lines 335-336. Add "over the Whole NH" before "shows a general…" to specify which panel the authors are referring to.**

Done.

**Lines 336-339. Suggest the following change: "The peak in 2010 is not seen at regional levels (NAM, ASI, EUR, Figure 6 (b)(c)(d)), which suggestglobal upward transport of the upper tropospheric ethane emissions (peaking in 2010-2011) into the stratosphere." No analysis was made to clearly indicate that there was an upward transport from the upper troposphere, therefore, it cannot be stated as such.**

Done.

**Line 339. Suggest the following change: "The second peak in 2013 can be due to the regional…". Same reason as explained in the previous comment.**

Done.

**Section 3.4.2 Add (if any) references and comparisons to other studies.**

Done.

**Lines 396-397. Specify which "current inventory" the authors are referring to.**

Done.

**TECHNICAL CORRECTIONS**

**Figures 1, 2, 3, 4, 6, and 7: Explanation of legends and terms used in the figures are missing in the caption.**

We have added explanation of legends and terms in the figures.

**There are two figures with number 5 (Lines 640 and 645).**

We have changed the figure numbering.

**Caption Figure 3. Clearly state sectoral contributions are model results.**

Done.

**Caption of Figure 4 should be revised because not all results correspond to "Optimized geographical sector contribution" as stated in the first sentence (Observations are not included in the caption). Also, clearly state that the optimized geographical contributions are model results.**

Done.

**Suggest the use of a different color scheme for the EUR, NAM, etc regions in Figures 4 and 7 to avoid confusion model optimizations in figure 2 and figure 6.**

We tried to make the same category (e.g. model sector, observation) with same color, for both upper tropospheric and stratospheric ethane. We have used 8 colors in figures for the upper troposphere, finding different colors for the stratosphere is therefore difficult.

**Sections 3.3.3 and 3.4.2 have exactly the same name. Indications to "troposphere" or "stratosphere" should be added to avoid confusion. The same goes for their corresponding figures in the supplemental material.**

Done. We have changed the titles for sections 3.3.3 and 3.4.2.

---

## Author Comment (AC2)

This manuscript reports on measurement data for methane, ethane, and propane from the IAGOS-CARIBIC commercial aircraft sampling program. Data from the upper troposphere/lower stratosphere are compared with outputs from the EMAC model. The conclusion from the disagreement in this comparison is that there are discrepancies between inventory and actual emissions of ethane.

This paper follows up on a series of other recent studies that have investigated the burden and atmospheric ethane, for instance [Aydin et al., 2011; Simpson et al., 2012; Franco et al., 2015; Franco et al., 2016; Hausmann et al., 2016; Helmig et al., 2016; Dalsoren et al., 2017; Tzompa-Sosa et al., 2017; Dalsoren et al., 2018; Monks et al., 2018; Tzompaâ□□Sosa et al., 2019; Angot et al., 2021].

I see a plethora of issues with this manuscript that dampen my enthusiasm for this work.

Overall, the manuscript does not seem all that carefully proofread, having an unusual high number of punctuation and formal errors. The concept of deriving a linear trend determination for the 2006-2016 window seems ill directed, given that it has been demonstrated that there was a decline in atmospheric ethane to approximately 2009, and then an uptick of ethane after that. Ignoring this trend reversal and then averaging trends over these two contrasting periods of atmospheric ethane changes does not seem to make much sense.

I am skeptical of the quality of the observational data and that the authors did not address any of the very obvious discrepancy of their findings with the prior published literature (see listing above). Year-to-year variability and trends in the ethane mole fraction are ways higher than reports in any of these prior publications. This points towards technical inconsistencies in the sampling program, possible analyte losses in the analytical system, or calibration issues.

We thank the reviewer for the forthright and critical comments which are all addressed in the text below. The original comments from the reviewer are marked as bold in black, and our replies are marked in blue.

We agree that there are differences with other published articles and we have clearly indicated it in the manuscript, for example in section 3.1 we have tried to summarize how the observed ethane trend from our unique long-term airborne dataset compares to other studies. Some

differences are clear and the comparison is straightforward. This makes the work of interest, particularly so when the data are compared to our best-modeled version of comparable data. Our observational trends have been evaluated with the atmospheric chemistry model (EMAC), the agreement between observation and model simulations supports the consistency of our observational data, and there is no reason to question the quality. One significant difference between our study and previously published works is the sampling location. Our aircraft sampled air at the tropopause region, whereas most other studies used ground-level measurement. This makes our reported trends and yearly variability inherently less dependent on sources directly upwind as is the case for a surface sampling site. Normally to estimate a trend of surface measurements, one needs to make sure the time series data are continuous (e.g. weekly, monthly), when not, missing values need to be filled by various approaches (e.g. using the same value as the previous value). This is possible and the uncertainty is small for surface observations, as they are often regular (e.g. weekly for NOAA VOCs) and with little missing values. Whereas it is more challenging for aircraft measurements, because (1) the time gap between two flights was not always exactly the same, (2) the sampling locations are rarely or impossibly the same (latitude, longitude, altitude). From what we have seen in the previous studies, trend analysis was often done by applying a linear fit. Such linear fit approach has shortcomings, e.g. (1) the estimated trend is influenced by the time period of choice (for a same dataset, the trend of one season may look much different as the trend of one year), (2) it doesn't capture inter-annual variability, (3) it can be influenced by outliers, etc. To improve these shortcomings, we used the "Prophet" algorithm which is open-source software and suitable for non-continuous time series data. It is robust to missing data and the influence from outliers is minimized. It is a non-linear model with seasonal effects included. Thus we think this algorithm will work better with our data than a linear fit. Due to the flight frequency of our aircraft measurements which is often one or two intensive days of measurements in a month, the temporal distribution of our data is ca. monthly, instead of being evenly distributed through every day. The "Prophet" algorithm outputs a value for trend for each measurement, and no outputs if no measurement exists on a day. Also, in the case of our data, the sampling locations may be different even if samples are collected above the same regions (EUR, ASI, etc), which results in different sample trajectories and abrupt changes in trend may occur. That is why our trend results do not look smooth and have large year-to-year variability. We are aware that our

trend analysis still has shortcomings, but we believe it is a better and more suitable approach for our data in comparison with linear fit. We now add more information in the manuscript to clarify the advantages and limitations of our analysis method (section 3.5).

There are many possible reasons to explain the differences with other studies (will be discussed in detail in the later replies). We respectfully disagree with the reviewer's statement that "This points towards technical inconsistencies in the sampling program, possible analyte losses in the analytical system, or calibration issues." Our analytical procedures have been well documented in publications and have been consistently applied using the same measurement system for many years (e.g. Schuck et al., 2009 for methane measurement, Baker et al., 2010 for ethane and propane measurements). These papers are also referenced in the manuscript. Operational lab notes are maintained and any measurements with potential interference from analytical issues were excluded. Our data has been examined by various journals, scientists, and no concerns about the quality of our data was reported. Data from the same instruments and projects are published, see e.g. Thorenz et al., 2017, Li et al., 2018, and Baker et al., 2016. The modeled data based on an updated emission inventory for ethane shows similar trends which lends further support for the dataset quality.

While we accept it is a good scientific practice to question any analytical method when the results obtained differ from previously held expectations, however, in this case we are confident that the analytical method and quality controls applied therein make this dataset robust. To strengthen this point we have now added more details in the manuscript about instrumentation, analytical and operational procedures in section 2, and limitations of our study in section 3.5.

Nevertheless, we understand the point of the referee, and in this manuscript, we have described the data and have an initial interpretation with comparison with model results, although we would be grateful to see further interpretation/studies based on our data, we have raised this point in the limitations and implications section.

The revised manuscript will be uploaded soon.

References:

Schuck, T. J., Brenninkmeijer, C. A. M., Slemr, F., Xueref-Remy, I., and Zahn, A.: Greenhouse gas analysis of air samples collected onboard the CARIBIC passenger aircraft, Atmos. Meas. Tech., 2009.

Baker, A. K., Slemr, F., and Brenninkmeijer, C. A. M.: Analysis of non-methane hydrocarbons in air samples collected aboard the CARIBIC passenger aircraft, Atmos. Meas. Tech., 2010.

Thorenz, U. R.; Baker, A. K.; Leedham Elvidge, E. C.; Sauvage, C.; Riede, H.; Velthoven, P. F. J. van; Hermann, M.; Weigelt, A.; Oram, D. E.; Brenninkmeijer, C. A. M.; Zahn, A.; Williams, J., Investigating African trace gas sources, vertical transport, and oxidation using IAGOS-CARIBIC measurements between Germany and South Africa between 2009 and 2011, Atmospheric Environment, 2017.

Li, M., Karu, E., Brenninkmeijer, C. et al. Tropospheric OH and stratospheric OH and Cl concentrations determined from CH4, CH3Cl, and SF6 measurements, Nature Clim Atmos Sci, 2018.

Baker, A. K., C. Sauvage, U. R. Thorenz, P. van Velthoven, D. E. Oram, A. Zahn, C. A. M. Brenninkmeijer & J. Williams, Evidence for strong, widespread chlorine radical chemistry associated with pollution outflow from continental Asia, Scientific Reports, 2016.

**Unfortunately, this work does not provide sufficient insight into the analytical protocols and quality controls to evaluate this question. Has the sampling apparatus and analytical lab ever been audited by the World Calibration Centre for VOCs [WCC-VOC, 2021], which most labs with global monitoring programs have been subjected to during the past ten years?  If so, then the results from this audit should be presented and critically evaluated.**

Our measurement apparatus and the papers describing it predate the establishment of the "World Calibration Centre for VOCs" and as yet we have not been offered the opportunity to undergo an analytical audit.  If this airborne measurement program restarts, as is hoped, then we would gladly participate. In lieu we have now added more information in the manuscript about the analytical protocols and quality controls applied. The procedures described in the previous published papers (Schuck et al., 2009, Baker et al., 2010) have been applied throughout this dataset so that long term data consistency and quality are assured. Certainly, having a world

calibration center to ensure comparability between different measurement methods and sites is important to address atmospheric budgets on a global scale. We now make this point in the text to advertise this point as follows:

"When comparing surface and airborne datasets from multiple locations to assess global atmospheric changes it will become increasingly important to ensure comparability of data quality, a process that has begun through the grounding of a World Calibration Center for VOCs, although this dataset predates this initiative."

**The reason for the difference in surface versus upper tropospheric trends needs more discussion/justification. Obviously, this defies the findings from column FTIR observations, e.g. [Helmig et al., 2016] who showed that, while lower in magnitude, lower and upper tropospheric trends were approximately in sink.**

Helmig et al., 2016 showed the ethane total column from FTIR observations at two stations: Jungfraujoch (for column 8-21km, and 3.6-8km) and Lauder (total column). Some of their trends are different from the upper tropospheric trends from our study. There are several possible explanations for this:

(1) Trends at the two stations at Jungfraujoch and Lauder do not represent the trends of the whole atmosphere (neither surface nor upper troposphere nor stratosphere). This has been shown in Figure 3(a) of Helmig et al., 2016 that large difference in ethane growth rates at different ground stations is observed (from -50 to 50 $pmol\ mol^{-1}y^{-1}$).

(2) FTIR observes the total (or partial) column of ethane. Considering the short lifetime of ethane, the observed trend from FTIR will likely be weighted more with the trend at near surface regions, whereas the observational data in our study do better represent the upper troposphere and stratosphere.

(3) Our aircraft sampled air at high altitudes on a global scale. This dataset is extremely useful to understand trends in regions where hydrocarbon emission inventories are poorly known. It is noted that the observed trends from aircraft measurements can be influenced by surface emissions such as convected plumes, or changes in stratosphere-troposphere exchange, or

changes in chemical destruction processes, and these can be reasons for the large year-to-year variability and trends observed from our study.

We now add more discussion in the manuscript to address this issue, though we must limit ourselves to the stated scope of the journal which is "Any interpretation of data is outside the scope of regular articles".

**Specific comments:**

**Abstract: The source/program of the measurement data should be mentioned. The model that was used should be mentioned.**

Done.

**Line 14-16:  I am surprised that the oil and gas sector is not clearly listed as an emissions category given its significant contribution to ethane emissions (see the above listed literature that overwhelmingly focused on the oil and gas emissions sector)?**

We used anthropogenic emission sectors from CAMS-GLOB-ANT v4.2 ethane emission inventory for model simulation. Oil and gas are included in the FEF (fugitives) and ENE (power generation) sectors, however, the emission inventory does not combine the oil and gas as one emission category. To our knowledge, the oil and gas emission studies from above-listed literature were not retrieved from model and emission inventories, but from the ratio of methane and ethane from observations, e.g. Helmig et al., 2016, Aydin et al., 2011.

**19: Given the overall uncertainty in emissions, four significant figures for the global ethane emissions do not seem justified.**

We categorized our samples into four regions, based on the number of samples we have in order to make sense in statistics. We agree that zooming into smaller regions would help better to understand global emissions and reduce uncertainty.

**23: Instead of using the term 'discrepancy' it would be more meaningful to use a term that clearly identifies if inventory emissions are over- or under-estimated.**

Done.

**30, 32, 41, etc: There are many, many cases throughout the manuscript where a comma should be placed within a sentence.**

Commas have been added to sentences on lines 30, 32, 41, etc to ease reading. We have carefully rechecked and improved the spelling. The professional editorial team from the journal will further assist with proofreading before this manuscript is published.

**56: Please explain what the number in parenthesis stands for.**

Done.

**103: Are methane and CO2 indeed measured with an ECD?**

Methane and CO2 are measured with FID, now we have revised it in the text.

**105-109: Details on the calibration scale, source of calibration standard should be provided, and how and when the scale was verified against the World Meteorological Organization Global Atmospheric Watch program that is used by the international community for global observations of ethane.**

We have provided more information on the calibration scale, calibration standard and other quality controls and protocols in the manuscript. The text reads:

" Methane, ethane, and propane were measured with a HP 6890 GC coupled with a flame ionization detector (FID) with a polymer Porapak Q 3/4" column (10 ft, 100/120 mesh) installed in a single oven. Nitrogen (N2, purity 99.999%) was used as carrier gas at a constant flow rate of 50ml/min. The FID was operated at oven temperature of 220°C with flow rates of synthetic air of 250ml/min and hydrogen of 80ml/min. Water vapor in samples was removed by passing through a drying tube at the start of the analysis. The calibration standards were ordered from NOAA (for methane), and the National Physical Laboratory (for ethane and propane) which are certified against World Meteorological Organization (WMO) Global Atmosphere Watch (GAW) program scale, and they are regularly renewed within every three years which warrants the stability of calibration gases. Three injections of calibration standards were made in between samples of each flight sequence in order to maintain the quality of measurements and reduce uncertainty."

**139: Replace 'of' with 'by'.**

We don't find any "of" in line 139 or anywhere near it.

**156: Insert 'the' before NOAA.**

Done.

**174: …related to emissions from the ….**

Done.

**188: What is the reference year for the % trend determination?**

We used Feb 2006 as the reference, now added in the text.

**193/194: No need to put parentheses around the time intervals.**

Done.

**204: Please provide more explanation on the criteria that were used for differentiating upper tropospheric air from stratospheric air and associated uncertainties.**

The criteria was using PV values to differentiate samples. In the meteorological community, PV is preferred over the thermal definition of the tropopause (the lowest level at which the temperature lapse rate decreases to 2 K km$^{-1}$ or less). PV is conserved under adiabatic and frictionless conditions, and it can capture tropospheric folds. Other tropopause definitions such as one based on ozone (O3) or an O3 to CO ratio also show a sharp transition at the tropopause. All these definitions give broadly consistent delineation of the tropopause although the chemically defined tropopause tends to be somewhat lower than the thermal one. PV was preferred to ensure that tropopause data was always available even if a particularly chemical measurement was not available.

As noted in the review of Stohl et al., 2003, on large space and timescales, in the extratropics, the WMO tropopause corresponds rather well to a surface of constant potential vorticity (PV) (e.g., Reed and Danielsen, 1958; Danielsen, 1968; Hoerling et al., 1991), although there exist systematic differences on smaller scales (Wirth, 2000). We now refer the reader to this review and provide more details about how the differentiating was done in the text.

References:

Stohl, A., et al. (2003), Stratosphere-troposphere exchange: A review, and what we have learned from STACCATO, J. Geophys. Res., 108, 8516, doi:10.1029/2002JD002490, D12.

Reed, R. J., & Danielsen, E. F. (1958). Fronts in the vicinity of the tropopause. Archiv für Meteorologie, Geophysik und Bioklimatologie, Ser. A, Meteorologie und Geophysik, 11(1), 1-17.

Danielsen, E. F. (1968). Stratospheric-tropospheric exchange based on radioactivity, ozone and potential vorticity. Journal of Atmospheric Sciences, 25(3), 502-518.

Hoerling, M. P., Schaack, T. K., & Lenzen, A. J. (1991). Global objective tropopause analysis. Monthly Weather Review, 119(8), 1816-1831.

Wirth, V. (2000), Thermal versus dynamical tropopause in upper-tropospheric balanced flow anomalies. Q.J.R. Meteorol. Soc., 126: 299-317. https://doi.org/10.1002/qj.49712656215

**208, Figure 1 caption:  Ethane is presented in mole fraction units (ppb) and not as a concentration. Use of the term 'concentration' is therefore technically not correct.**

We have revised it.

**219:  Wording is not correct. Uncertainties are presented in Figure S3.**

We have revised the wording.

**246 - 262: A more convincing approach would be to first evaluate and optimize the modeled trends using available and much more abundant (and likely better quality) surface data (possibly high elevation mountaintop data, e.g. Jungfraujoch, Mauna Loa, Summit) or FTIR column records, and then move on towards evaluating the upper troposphere trends.**

We agree that this is a valid approach to evaluate model and emission inventories. However, we did not proceed in this way as our EMAC model has been evaluated for ethane using surface data, e.g. Pozzer et al., 2021. The ethane emission inventory used in our study has been also optimized with ground observations from our previous study (Pozzer et al.,2020).

References:

Pozzer, A., Reifenberg, S., Kumar, V., Franco, B., Taraborrelli, D., Gromov, S., Ehrhart, S., Jöckel, P., Sander, R., Fall, V., Rosanka, S., Karydis, V., Akritidis, D., Emmerichs, T., Crippa, M., Guizzardi, D., Kaiser, J. W., Clarisse, L., Kiendler-Scharr, A., Tost, H., and Tsimpidi, A.: Simulation of organics in the atmosphere: evaluation of EMACv2.54 with the Mainz Organic Mechanism (MOM) coupled to the ORACLE (v1.0) submodel, Geosci. Model Dev. Discuss., 2021.

Pozzer, A., Schultz, M. G., and Helmig, D.: Impact of U.S. Oil and Natural Gas Emission Increases on Surface Ozone Is Most Pronounced in the Central United States, Environmental Science & Technology, 54, 12423-12433, 2020.

**249:  It shouldn't really be a surprise that the model performs better for methane, given the much lower variability of methane in time and geographically, and given its 40 times longer lifetime?  Please avoid subjective terms such as 'better', and instead give a concrete quantitative evaluation.**

We have revised it. Now it reads:

"The initial model results underestimate ethane mole fraction by about 45%, whereas the model estimation is closer to observation for methane with the same model and observation dataset (Zimmermann et al., 2020)."

**259: The oil spillage doesn't really matter for ethane. It would make more sense to provide emissions/leakage/spill estimates for natural gas, including ethane.**

We have deleted this statement.

**262: Insert 'the' after although.**

Done.

**283: Please explain/spell out all abbreviations when they are used for the first time.**

The abbreviations for regions (ASI, EUR, etc.) are explained in the first paragraph of section 3.2 and Table S1. All abbreviations for emission sectors are now added in section 2.2 and explained in Table 1.

**285: Doesn't this result contradict the findings from other prior literature, e.g.[Franco et al., 2015; Franco et al., 2016; Hausmann et al., 2016; Helmig et al., 2016] who claim that North America is the main contributing region for the rise in global ethane post 2009?**

We don't think our findings contradict previous studies. Some previous studies have indeed shown that surface ethane emissions from North America have been increasing since 2009, contributing to the rise in global surface ethane. This reported increase was observed from surface measurements, whereas the contributions of ethane from each region reported in our study (e.g. Figure 4) were simulated by the atmospheric model and emission inventory which may not include such ethane rise in North America from bottom-up approach. Our findings stated in line 285 indicate the relative contribution of air emitted from each region to the observed upper tropospheric ethane trend. We find that about 50% of upper tropospheric ethane of the whole Northern Hemisphere was transported from Asia, which makes sense considering the transport of pollutants from Asia into the stratosphere via Asian Monsoon. An example to better understand this concept: for Figure 4(b), the black line indicates the upper tropospheric ethane trends derived from samples collected above NAM, the red line (from model simulations) indicates the ethane emissions originated from ASI which influenced the black line. In this case, we see that a large fraction of ethane from samples collected above NAM was influenced by ASI emissions, which indicates that our samples and observed trends are influenced by global transport, and do not represent local emissions, and thus should not be interpreted as local ethane trends or be compared with surface observations. We now add these points in the manuscript.

**295-301: As mentioned previously, I question that trend results over this time window are meaningful, given that this spans over a point in time when there was a reversal of the ethane trend.**

We have presented the trends over two time spans when there was a turning point in 2010-2011 in the fourth paragraph of section 3.3.1, i.e. trends between 2006-2010, and 2010-2016. In section 3.3.3, we provided an overall trend for 2006-2016, calculated as the difference of ethane mole fraction in 2016 and 2006, divided by 10 years: $(c_{(2016)} - c_{(2006)})/10yr$. We now add this in section 2.3 for better clarification. This approach minimizes the influence of any short-term change in ethane sources or sinks over that time period. As we cannot confirm what are the drivers of trend reversal (location bias with emission plume, or change in stratospheretroposphere exchange, etc.), we preferred to present the observed trends in both ways (one as an overall trend, the other one to separate the reversal trend).

**314-318:  Again, 'well' is a subjective qualifier.  Please provide a concrete quantitative evaluation.**

We have now revised it and moved this section to Supplementary as suggested by the other reviewer.

**321: Unreasonable number of significant figures.**

Done.

**334:  …is again a minor ….**

Done.

**338:  What do you mean by 'upper tropospheric ethane emissions'?  I am not aware of ethane emission sources in the upper troposphere?**

We have removed the word "emissions".

**353: Figure 6 shows time series data, not trends.**

Figure 6 shows the stratospheric ethane trends which were derived from observational data by a statistical model "Prophet" in Python (see section 2.3). The trends shown in Figure 6 are not the observed ethane mole fractions from aircraft sampling. Our trends are equivalent to the orange line in Fig.1(b) of Helmig et al., 2016. We now also add the time series of observed mole fractions in the figures in SI.

**378: Shouldn't the NOAA data source be included here as well?**

Now we add the NOAA data source.

**396: Please name the inventory.**

Done.

**402: 'Trend' is not a suitable term here.**

Done.

**411: ..revised …**

Done

**605:  The listed ethane sources do not add up to the total that is listed at the bottom?**

The sum of BIO, BIB and Table 1(a) is equal to the sum of BIO, BIB and Table 1(b), which is 19.28 Tg/yr. Note that Table 1(a) and (b) are only for anthropogenic emissions.

**Figure 2: To the very best of my knowledge, the ethane spike that is reported in 2010 has not been seen in any other data products, including FTIR column observations and surface measurements, including those taken at mountain sites. This makes me quite skeptical of these results. In my opinion, this requires a much more thorough evaluation/verification, including through careful consideration of available other data sources. Can this be explained by interannual differences in the fraction of stratosphere-troposphere exchange that the aircrafts encountered during their sampling? That would then raise the question or representative each year's aircraft data are?**

As replied to the previous comments, there are differences between our aircraft measurements and ground observations. Indeed aircraft observations at UTLS region can be influenced by the change in stratosphere-troposphere exchange and sampling locations. Ground-based trends are easier to interpret as they are dominated by the upwind activities. High altitude trends are more difficult due to wider source footprint, stochastic convection events, and stratospheric mixing. This is why here we have applied a global model for comparison. One may indeed question the representativeness of aircraft data as it is dependent on sampling location. To address this, we have examined the sampling location bias in the manuscript (method described in section 2.2) by simulating ethane mole fraction for each aircraft sampling location with constant meteorology and constant ethane emission as model input (so called "climatology").We find this bias has little influence (discussed in section 3.3.1 and 3.4.1). Compared to surface observations, aircraft observations provide unique information to express a more homogenized global trend which is inevitably a variable composite of multiple global sources. We have deposited our data including sampling date and time, location (longitude, latitude, altitude), pressure, together with observed and modeled trace gas mole fractions on a publicly accessible repository. This guarantees the

reuse of our data in future studies to examine/be aware of the representative of aircraft data. We also add a new section (3.5) to discuss the limitations of our study including the representativeness of aircraft data.

**Figure 3: The size of the four colored emission regions does not seem to agree with the data in Table 1.**

We realize that this figure can cause confusion, therefore, we moved it to the Supplement and added more explanatory text. The right side of the figure shows the contributions of all sectors for each region, assuming 100% for each region which is why the sizes of them are the same. We were trying to show the relative contributions among sectors for each region. Unfortunately, we cannot derive emissions of each sector for troposphere and stratosphere separately from our model.

**Figure 5: Graph shows time series data, not trend results. The 2011 propane spike is very significant, representing a 50% increase from prior years. This dramatic change is much larger than year-to-year variability seen in other data products. This seems unreasonable and makes me question the quality of the underlying data. This calls for much more documentation on the quality control procedures and a convincing demonstration of the long-term stability of the sampling and analysis program.**

Figure 5 shows stratospheric ethane trends which were derived from observational data by a statistical model "Prophet" in Python (see section 2.3). We have interpreted the spike in 2011 as a plume of oil and gas emission. We also see a peak in 2010-2011 for ROW trend as well (figure added in SI). Because about half of the samples were collected above ROW (SI), the whole NH trend may be largely influenced by ROW. If we exclude the spike from the trend, we see that the change between 2006 and 2016 is small and comparable with other studies. Observing a spike by aircraft does not mean that the aircraft observation is wrong or is incompatible with a global trend since the observed ethane trends match reasonably well with trends from atmospheric chemistry model simulations. As requested, we have documented our measurement procedure further. These have been consistently applied so that we are confident of the quality of our data.

**Fig 6/7: I would like to see information on how the data were treated, averaged, binned, and how uncertainty ranges were determined. These figures are the pivotal example for my**

**skepticism of the data quality and the knowledge/understanding of the authors of the atmospheric behavior of these NMHC. For the ASI panel, there is a more than 50% drop of ethane in the observations from 2013 to 2015. So, that's a reduction of ethane to less than half in two years. No other data set that is referred to in the above-cited literature shows year-to-year changes of that magnitude. A concentration drop like this is highly unreasonable, most likely impossible. The best explanation that I can think of is that there were sampling or measurement problems that the authors did not become aware of and investigate. Or that there were strong differences in the fraction of stratospheric versus tropospheric air that were sampled each year and that the differentiation wasn't accomplished properly?**

Regarding the drop of stratospheric ethane above ASI shown in Figure 6/7, it should be noted that there is a very large uncertainty associated. Such large uncertainty is seen only for ASI, very likely due to the smaller amount of sampling numbers. As replied to previous comments, whole NH trends can be influenced by the regions with more samples collected (we include this point in the discussion of limitations). Trend variability within only two years may be even more influenced by that. This is also the reason why we only split our data into four regions for the presentation and not into more. Secondly, Asia is a region with strong stratosphere-troposphere exchange, and as well as surface emissions. Thirdly, if we compare the model and observation of stratospheric ethane above ASI, we see a relatively good match before 2011, and for other regions like NAM and EUR, the modeled trends match well with observation thorough the entire time period, this indicates that our measurement is not wrong, so we exclude the possibility of measurement problems. The discrepancy between model and observation for ASI after 2014 may indicate changes in stratosphere-troposphere exchange, or changes in ethane sources or sinks, which are not captured by model or emission inventory.

**Figure 8: Similar reservations as already explained for Figure 6/7: I can not believe how propane would drop in the stratosphere by a factor of more than four from 2006 to 2011. This defies every other data set on atmospheric NMHCs that I have seen in the published literature.**

To our best of knowledge, there is no study reporting stratospheric propane trends which covers the time period as our study, therefore, we kindly ask the reviewer to provide the literature which shows different stratospheric propane trends. As replied to the previous comments, stratospheric trends are not only being influenced by surface emissions, but also atmospheric transport

(stratosphere-troposphere exchange), and chemical destructions. For example, chlorine radicals are abundant in the stratosphere and are important oxidants for stratospheric propane, ethane and methane, whereas chlorine concentration on the surface is much smaller (~100 times lower than in the stratosphere) (Gromov et al., 2018). We now add this point in the manuscript, and we think comparing surface trends with stratospheric trends needs to be done in a careful manner.

As for the surface propane and ethane, Warneke et al., 2012 summarized a series of studies and showed that surface propane mole fractions have been decreasing at the rate of -4.6%/year in the LA basin (1960-2010), and at -3%/year in London (1998-2008), whereas ethane had even higher decreasing rate over the same time period and locations. Although surface trends of ethane or propane should not be directly compared with stratospheric trends, the fast drop of surface ethane and propane in some regions may play a role in the observed stratospheric trends.

Reference:

Gromov, S., Brenninkmeijer, C. A. M., and Jöckel, P.: A very limited role of tropospheric chlorine as a sink of the greenhouse gas methane, Atmos. Chem. Phys., 18, 9831–9843, https://doi.org/10.5194/acp-18-9831-2018, 2018.

Warneke, C., de Gouw, J. A., Holloway, J. S., Peischl, J., Ryerson, T. B., Atlas, E., Blake, D., Trainer, M., and Parrish, D. D. (2012), Multiyear trends in volatile organic compounds in Los Angeles, California: Five decades of decreasing emissions, J. Geophys. Res., 117, D00V17, doi:10.1029/2012JD017899.

**Literature cited**

**Angot, H., C. Davel, C. Wiedinmyer, G. Pétron, J. Chopra, J. Hueber, B. Blanchard, I. Bourgeois, I. Vimont, S. A. Montzka, B. R. Miller, J. W. Elkins, and D. Helmig (2021), Temporary pause in the growth of atmospheric ethane and propane in 2015–2018, Atmos. Chem. Phys. Discus., https://doi.org/10.5194/acp-2021-285.**

**Aydin, M., K. R. Verhulst, E. S. Saltzman, M. O. Battle, S. A. Montzka, D. R. Blake, Q. Tang, and M. J. Prather (2011), Recent decreases in fossil-fuel emissions of ethane and methane derived from firn air, Nature, 476, 198-201, doi:10.1038/nature10352.**

Dalsoren, S. B., G. MYhre, Ø. Hodnebrog, C. L. Myhre, A. Stohl, I. Pisso, S. Schwietzke, L. Hoeglund-Isaksson, D. Helmig, S. Reimann, S. Sauvage, N. Schmidbauer, K. A. Read, L. Carpenter, A. C. Lewis, S. Punjabi, and M. Wallasch (2017), Ethane and propane emissions substantially underestimated in community emission datasets, Nature Geoscience, In press.

Dalsoren, S. B., G. Myhre, O. Hodnebrog, C. L. Myhre, A. Stohl, I. Pisso, S. Schwietzke, L. Hoglund-Isaksson, D. Helmig, S. Reimann, S. Sauvage, N. Schmidbauer, K. A. Read, L. J. Carpenter, A. C. Lewis, S. Punjabi, and M. Wallasch (2018), Discrepancy between simulated and observed ethane and propane levels explained by underestimated fossil emissions, Nature Geoscience, 11, 178-186, doi:10.1038/s41561-018-0073-0.

Franco, B., W. Bader, G. C. Toon, C. Bray, A. Perrin, E. V. Fischer, K. Sudo, C. D. Boone, B. Bovy, B. Lejeune, C. Servais, and E. Mahieu (2015), Retrieval of ethane from ground-based FTIR solar spectra using improved spectroscopy: Recent burden increase above Jungfraujoch, Journal of Quantitative Spectroscopy & Radiative Transfer, 160, 36-49, doi:10.1016/j.jqsrt.2015.03.017.

Franco, B., E. Mahieu, L. K. Emmons, Z. A. Tzompa-Sosa, E. V. Fisher, K. Sudo, B. Bovy, S. Conway, D. Griffin, J. Hannigan, K. Strong, and K. A. Walker (2016), Evaluating ethane and methane emissions associated with the development of oil and natural gas extraction in North America, Environmental Research Letters, 11, 1-11.

Hausmann, P., R. Sussmann, and D. Smale (2016), Contribution of oil and natural gas production to renewed increase of atmospheric methane (2007–2014): top-down estimate from ethane and methane column observations, Atmos. Chem. Phys., 16, 3227-3244.

Helmig, D., S. Rossabi, J. Hueber, P. Tans, S. Montzka, K. Masarie, K. Thoning, C. Plass-Duelmer, A. Claude, L. Carpenter, A. C. Lewis, S. Punjabi, S. Reimann, M. Vollmer, R. Steinbrecher, J. Hannigan, L. Emmons, E. Mahieu, B. Franco, D. Smale, and A. Pozzer (2016), Reversal of global atmospheric ethane and propane trends largely due to US Oil and natural gas production, Nature Geoscience, 9, 490–495, doi:doi:10.1038/ngeo2721.

Monks, S., C. Wilson, L. Emmons, D. Helmig, J. Hannigan, D. Blake, and N. Blake (2018), Using an inverse model to reconcile differences in simulated and observed global ethane

concentrations and trends between 2008 and 2014, Journal of Geophysical Research, 11262-11282.

Simpson, I. J., M. P. S. Andersen, S. Meinardi, L. Bruhwiler, N. J. Blake, D. Helmig, F. S. Rowland, and D. R. Blake (2012), Long-term decline of global atmospheric ethane concentrations and implications for methane, Nature, 488, 490-494, doi:10.1038/nature11342.

Tzompa-Sosa, Z. A., E. Mahieu, B. Franco, C. A. Keller, A. J. Turner, D. Helmig, A. Fried, D. Richter, P. Weibring, J. Walega, T. I. Yacovitch, S. C. Herndon, D. R. Blake, F. Hase, J. W. Hannigan, S. Conway, K. Strong, M. Schneider, and E. V. Fischer (2017), Revisiting global fossil fuel and biofuel emissions of ethane, Journal of Geophysical Research-Atmospheres, 122, 2493-2512, doi:10.1002/2016jd025767.

Tzompaâ□□Sosa, Z. A., B. H. Henderson, C. A. Keller, K. Travis, E. Mahieu, B. Franco, M. Estes, D. Helmig, A. Fried, D. Richter, P. Weibring, J. Walega, D. R. Blake, J. W. Hannigan, I. Ortega, S. Conway, K. Strong, and E. V. Fischer (2019), Atmospheric implications of large C2â□□C5 alkane emissions from the U.S. oil and gas Industry, Journal of Geophysical Research, 124, 1148-1169.

WCC-VOC (2021), World Calibration Centre for Volatile Organic Compounds, http://www.imk-ifu.kit.edu/wcc-voc/, Karlsruhe Intitute of Technology.

---

## Author Comment (AC3)

**General comments**

Li et al. present long-term atmospheric ethane, propane, and methane data obtained from airborne observation in the UTLS region in the Northern Hemisphere through IAGOC-CARIBIC project for the period 2006–2016. The authors also present simulation outputs from the EMAC model to interpret the contributions of different source regions or different source sectors to the observed ethane trend.

The manuscript is generally well written. The airborne observational ethane, propane, and methane data in the UTLS region is unique and useful because there is only a limited platform that can measure with high precision such trace gases regularly in this region - perhaps only IAGOC-CARIBIC (this project) and CONTRAIL project (e.g., Machida et al. 2008; Sawa et al. 2015). The observational data will certainly be reused by many researchers in this field, e.g., validate the emission inventories (as the authors did), interpret the atmospheric transport and chemical processes in the UTLS region, through the comparison with model simulations.

However, the current manuscript is lacking the information on methods (e.g., model simulation setup, trend analysis, measurement uncertainty), making the readers a bit confused and difficult to interpret the presented results (and dataset). For example, I couldn't find how the emission inventories are optimized from prior emission inventories to match the observation data. Under the lack of such important information, it is difficult to judge these model outputs are worth publication or not, despite the uniqueness and potential usefulness of the observational data itself. Also as for the measurements, the authors just cited the previous (~10 years ago) publications for ethane and propane measurements (Baker et al. 2010) and for methane measurements (Schuck et al. 2009), but I would recommend the author to describe information on the data quality control over the whole analysis period (2006–2016). For example, how the long-term stability of the standard gases has been maintained? Such information should be necessary for the readers and data users when using the data and interpreting the long-term trend of these trace gases over the decades, especially for ethane and

**propane, whose measurement uncertainties are comparable with the discussed observed trends.**

**Considering the above, I would recommend this paper for publication after major revisions.**

We thank the reviewer for the helpful comments, suggestions and discussion. We appreciate that the reviewer finds our data unique, precise and useful for future studies. We have addressed all points raised in the revised manuscript. The original comments from the reviewer are marked as bold in black, and our replies are in blue.

We have added more information on model setup and optimization in section 2.2, measurement details (data collection, quality control, instrumentations, measurement procedures, uncertainties, etc.) in section 2.1, and trend analysis in section 2.3. We have also included a section to address the limitations and implications of our study.

The revised manuscript will be uploaded soon.

**Specific comments**

**Title: I would recommend that the authors reconsider the title because this study present almost only the ethane trends and the data is obtained in the Upper Troposphere – Lower Stratosphere (UTLS) in the northern hemisphere, not in the global.**

We now change the title to "Northern hemispheric atmospheric ethane trends (2006-2016) with reference to methane and propane". We also add the following texts in the Abstract to clarify this point:

"The model simulations, and methane and propane observations provide additional information for understanding northern hemispheric ethane trends and emissions, which is the primary focus of this study."

**L10 "global": Same as the comment in the title.**

Done.

**L66: Please clarify "the global ethane trend" and "long-term global ethane datasets". Emissions or atmospheric mole fractions? Also, what is "local influences"?**

We now revise the text as follows:

"Therefore, to determine the global ethane trends in terms of mole fractions and emissions with greater certainty, long-term global ethane datasets from observations and model simulations with minimal influences from local sources (e.g. observations at higher altitudes) are required"

**L84 "global": Same as the comment in the title.**

Done.

**L105-108: Please describe the calibration scales used in this study. Just "please see the references" is inconvenient for the readers who expect to use the dataset for this study. Also please mention the long-term stability of calibration gasses, as well as long-term reproducibility of the sample measurements.**

We now add more information in section 2.1 (see text below) regarding data collection, analytical protocols, measurement and calibration procedures, and quality control, etc.

"Methane, ethane, and propane were measured with a HP 6890 GC coupled with a flame ionization detector (FID) with a polymer Porapak Q 3/4" column (10 ft, 100/120 mesh) installed in a single oven. Nitrogen ($N_2$, purity 99.999%) was used as carrier gas at a constant flow rate of 50ml/min. The FID was operated at oven temperature of 220°C with flow rates of synthetic air of 250ml/min and hydrogen of 80ml/min. Water vapor in samples was removed by passing through a drying tube at the start of the analysis. The calibration standards were ordered from NOAA (for methane), and the National Physical Laboratory (for ethane and propane) which are certified against World Meteorological Organization (WMO) Global Atmosphere Watch (GAW) program scale, and they are regularly renewed within every three years which warrants the stability of calibration gases. Three injections of calibration standards were made in between samples of each flight sequence in order to maintain the quality of measurements and reduce uncertainty. "

**L124: Cl is important for the modeling of ethane, methane, and propane. The authors need to discuss the issue more in the section 3 (e.g., L345-350 and L374-375).**

Chlorine chemistry is more important for the stratosphere than for the troposphere. We add the following text to address it:

"The loss of ethane in the stratosphere by reaction with Cl radicals is about 40 times more than that by OH radicals (reaction rate of ethane with Cl is about 400 times faster than with OH at 250K (Atkinson et al., 2001), and stratospheric OH is about ten times more abundant than stratospheric Cl (Li et al., 2018)), whereas the ethane loss in the troposphere by Cl is negligible compared with by OH due to the small amounts of tropospheric Cl (OH:Cl around 10,000) (Gromov et al., 2018; Lelieveld et al., 1999)."

**L129-135: I think this part is very important for readers to know what the model optimization setup is. However, I don't think it is well written at this stage.**

We now add more information on model optimization in section 2.2. Now the section 2.2 reads:

"The ECHAM/MESSy Atmospheric Chemistry (EMAC) model is a numerical chemistry and climate simulation system that includes sub-models describing tropospheric and middle atmosphere processes and their interaction with oceans, land, and human influences (Jöckel et al., 2010). It uses the second version of the Modular Earth Submodel System (MESSy2) to link multi-institutional computer codes. The core atmospheric model is the 5th generation European Centre Hamburg general circulation model (ECHAM5, Roeckner et al. (2006)). For the present study, we applied EMAC (ECHAM5 version 5.3.02, MESSy version 2.55.0) in the T63L47MA-resolution, i.e. with a spherical truncation of T63 (corresponding to a quadratic Gaussian grid of approx. 1.8 by 1.8 degrees in latitude and longitude) with 47 vertical hybrid pressure levels up to 0.01 hPa (~80 km). The model has been weakly nudged towards the ERA5 reanalysis data of the ECMWF (Hersbach et al., 2020). The chemical mechanism comprises methane, alkanes, and alkenes up to C4, ozone, odd nitrogen, some selected non-methane hydrocarbons (NMHCs), heterogeneous reactions, etc. In total, 310 reactions of 155 species are included in the model. The photolysis rates are calculated following Sander et al. (2014). No chlorine chemistry is included in the model. To account for realistic emissions, the CAMS-GLOB-ANT v4.2 emission inventory data is used for model simulations (Granier et al., 2019; Guevara et al., 2020). In this study, we have included 13 emission sectors (shown in Table 1) which are BIO (biogenic emission), BIB (biomass burning), AWB (agricultural waste burning), ENE (power generation), FEF (fugitives), IND (industrial processes), RES (residential energy use), SHP (ships), SLV (solvents), SWD (solid waste and waste water), TNR (off-road transportation), TRO (road transportation), and AIR (aviation). It is noted that AIR, BIB and BIO were combined as one

sector to reduce the uncertainty. AIR is not shown in Table 1 as its contribution is negligible. It has been shown by multiple studies that the ethane emissions due to fossil fuel combustion are strongly underestimated in the emissions database (Guevara et al., 2021; Pozzer et al., 2020; Helmig et al., 2016). In this work, we therefore increased the anthropogenic emissions of ethane of a factor of 2.47 to match (for the year 2010) the total amount suggested by Pozzer et al. (2020) although the value used in this study (~11.8 Tg/yr) slightly underestimates the measured mole fraction as shown in Pozzer et al. (2020) (13.2 Tg/yr). We further optimized modeled ethane mole fractions for each emission sector (referred to as "opt" in the later figures and "optimized" in the later texts). The model optimization is done by increasing the emissions of each input emission sector by 45%. We found that the root mean squared error (RMSE) between the modeled and observational ethane mole fractions for the whole dataset was at a minimum after a 45% increase in the input emissions. The input ethane emissions from natural and anthropogenic sources are presented in Table 1, together with description for each sector and optimized sectoral emissions (will be discussed in the Results and Discussion section).

In this study, two types of ethane trends were presented with the model simulation: (1) constant meteorology and constant emission (hereafter called climatology), sampled at the IAGOS-CARIBIC sampling location with S4D algorithm (sampling in 4 dimensions) described in Jöckel et al. (2010). Any trends (or changes) detected in this simulation would be caused by differences in sample location and timing. (2) real meteorological conditions from ECMWF and the adjusted emissions described above, sampled at the IAGOS-CARIBIC sampling location with S4D algorithm (Jöckel et al., 2010). "

**L129-132: Did you increase not the fossil fuel combustion only, but the total anthropogenic emissions? If so, what is the justification of this? Then did you increase the total anthropogenic emissions to match which Tg/yr in Pozzer et al. (2020) (13.2 or 15.3 Tg/yr, as in Section 1)?**

We increased all the emission sectors including anthropogenic, biogenic and biomass burning emissions by 45%. As our model and emission input have been examined many times before, we only did this simple optimization. And the aim of our study is not to improve model prediction, but to use model to explain the changes in observed trends. We now add more explanation in section 2 to clarify this.

**L132-135: How to estimate the "45%"? Without the information of the optimization method, it is difficult to interpret the model results hereafter.**

We found that the root mean square error (RMSE) is minimum between model and observations when the model results are increased by 45%. We use this as the method for model optimization. We now add the information in section 2.2.

**L144 "Prophet": As long as I see the main figures (e.g. Fig. 2, 4, 5), I wonder how this algorithm can capture the discussed long-term trend over the decade. The results show noisy inter-annual variations, not averaged smoothed trend. Maybe the author also need to show the raw data obtained in this study, as done in Fig. S1.**

Our aircraft sampling is different to surface measurement in terms of temporal and spatial distributions of samples. Normally to estimate a trend of surface measurements, one needs to make sure the time series data are continuous (e.g. weekly, monthly), when not, missing values need to be filled by various approaches (e.g. using the same value as the previous value). This is possible and the uncertainty is small for surface observations, as they are often regular (e.g. weekly for NOAA VOCs) and with little missing values. Whereas it is more challenging for aircraft measurements, because (1) the time gap between two flights was not always exactly the same, (2) the sampling locations are rarely or impossibly the same (latitude, longitude, altitude). From what we have seen in the previous studies, trend analysis was often done by applying a simple linear fit. Such linear fit approach has shortcomings, e.g. (1) the estimated trend is influenced by the time period of choice (for a same dataset, the trend of one season may look much different as the trend of one year), (2) it doesn't capture inter-annual variability, (3) it can be influenced by outliers, etc. To improve these shortcomings, we used the "Prophet" algorithm which is an open source software and suitable for non-continuous time series data. It is robust to missing data and the influence from outliers is minimized. It is a non-linear model with seasonal effects included. Thus we think this algorithm will work better with our data than a linear fit. Due to the flight frequency of our aircraft measurements which is often one or two intensive days of measurements in a month, the temporal distribution of our data is ca. monthly, instead of being evenly distributed through every day. The "Prophet" algorithm outputs a value for trend for each measurement, and no outputs if no measurement exists on a day. Also, in the case of our data, the sampling locations may be different even if samples are collected above the same regions (EUR, ASI, etc), which results in different sample trajectories and abrupt changes in

trend may occur. That is why our trend results do not look smooth. We are aware that our trend analysis still has shortcomings, but we believe it is a better and more suitable approach for our data in comparison with linear fit. We now add more information in the manuscript to clarify the advantages and limitations of our trend analysis method (section 3.5). We also add ethane mole fractions from aircraft observations in figures in Supplementary Information (SI) together with the trends from Prophet. These figures clearly show that Prophet captures the trends well.

**L150-162 "changepoint_prior_scale": Please explain the definition of the parameter; otherwise, the readers cannot understand what the uncertainty interval represents.**

We now clarify it in the section 2.3.

**L155-158: If so, please show the comparison of fitting between NOAA algorithm and Prophet algorithm in the supplement Fig. S1?**

For Iceland surface ethane observation data from NOAA, both algorithms estimated the trend of ~900 ppt for Jan 2006, NOAA algorithm estimated the trend for Jan 2007 of ~1400 ppt, and Prophet algorithm estimated it to be 1350~1500 ppt. For the years 2008-2015, both algorithms estimated the trends to be around 1400 ppt. As we don't have the results from NOAA algorithm from Helmig et al., 2016 and including the figure from Helmig et al., 2016 may potentially bring copyright issues, we only describe the agreement between two algorithms in the manuscript, and provide the reference of Helmig et al., 2016 for readers who are interested in looking into more details.

**L165-205: I wonder why the estimated ethane trend of this study is so different from those of previous studies (same as in L237-245, L364-371). Calculating the trend for another periods would be useful to directly compare the results (e.g., 2009-2014), but I also wonder how the "Prophet" can robustly estimate the trend, due to the large inter-annual variations (as commented in L144). Also, the authors need to explain how they separate the air mass origin between troposphere and stratosphere before presenting the result (e.g., in Method). If the criteria of troposphere and stratosphere is set to be "PV = 2", as in L204, please describe it earlier.**

Previous studies provided ethane trends based on surface observations which are largely influenced by local ethane sources and sinks. Our aircraft sampled air at high altitude (~10km) on a global scale. This dataset is extremely useful to understand trends in regions where

hydrocarbon emission inventories are poorly known. It is noted that the observed trends from aircraft measurements can be influenced by surface emissions such as convected plumes, or changes in stratosphere-troposphere exchange. Thus it is expected to see different ethane trends compared with other surface observations.

We now calculate the ethane trend for another period (2007-2014) to directly compare with other studies, and the values are added in Table 2.

We now add more explanation on the criteria of the tropospheric and stratospheric air samples in section 2.1.

**L207: According to Fig. S2, the sample number obtained in Rest of the world (ROW) is dominant more than half of all data in the stratosphere. Then I wonder where the air samples are collected in the ROW and wonder if the current air mass classification is valid for interpreting the trend?**

Our aircraft flight paths are from Germany in Europe to the other continents/regions defined in our study (ASI, NAM, ROW). ROW samples are mostly collected above the oceans (see newly added Figure 1 in the revised manuscript and below) and have broad spatial distributions. These samples have diverse trajectories and thus cannot be easily classified. Indeed ROW is dominant for stratospheric samples and also for upper tropospheric samples in the year 2010 and 2011 when we observed a spike in the whole NH upper tropospheric ethane trend. We now add figures for ethane, methane and propane trends at ROW (troposphere and stratosphere) in SI. We don't observe a spike at ROW for 2010-2011. We suspect that the spike for the whole NH was due to the uneven temporal distributions of the origin of samples. There was no ASI samples in 2009, thus the 2009 trend was influenced by ROW, NAM and EUR. In 2010 and 2011, there were more ASI samples and that was the time period when ASI trend reached peak, thus the whole NH trend shows a spike at that time. This reveals the limitations of our trend analysis, we add a new section to discuss the limitations of our study.

[Figure]

**L213-215: Is the seasonality in ASI not statistically significant? It is interestingly showing the same peak timing with UT in June. Is it potentially indicating the intrusion of tropospheric air masses into stratosphere due to Asia summer monsoon (e.g., Park et al. 2007; Xiong et al. 2009).**

This is a good point. We modify the text in section 3.2 and it reads now:

"In contrast, the stratospheric ethane mole fractions do not show strong seasonality, except that NAM has a seasonal trend with 3-month later shift compared to the upper tropospheric NAM trend, and stratospheric ASI ethane shows the same timing peak in June with upper tropospheric ASI ethane which potentially indicates the intrusion of tropospheric air masses into the stratosphere due to Asian summer monsoon (Xiong et al., 2009; Park et al., 2007)."

**L228-234: It seems saying about the discrepancy of the inter-annual variations between observation and model climatology, not that of the whole trend for 2006–2016. Also, the observational trends estimated by Prophet show very large inter-annual variations and thus it seems difficult to compare the long-term trends by this analysis.**

The climatology simulation was performed to test the importance of the sampling location on the trend estimation. The climatology, by definition in its construction, does not present trends nor interannual variability. Therefore, any trend observable by the sampling of the climatological data along the flight tracks is purely due to the aircraft position and not from changes in atmospheric composition. Therefore, the fact that no noticeable trends are present confirms that the frequency and location of the sampling did not influence the estimation of the trends, while those are due to changes in the atmospheric composition (by e.g. changes in emissions, in

meteorology and hence transport, or in chemical transformation). We now add the climatology trends for each region in the figures.

Regarding Prophet, please see our replies to a previous comment (L144).

**L264-270: I would suggest the authors describe the source sub-category information in the method section. And then explain which sub-category emissions are optimized.**

We have added descriptions of each emission sector in the method section and explained how the optimization was done.

**L272-274: Same as the general comment and specific comment (L129-135 and L264-270). I wonder how the author optimized the emission inventories.**

We now explain more about the model optimization in the section 2.2.

**L283-284 "AIR+BIB+BIO": It is said they cannot be separated into regions. Then how are they included into the model simulation?**

These sectors were kept global, i.e. they represent the global emissions. In principle, we could separate them. However, more uncertainty will be added by separating BIB and BIO. In addition, biogenic emissions are very small (almost negligible). For biomass burning (BIB), a more detailed analysis could be performed, although the regions should be based on fire type/burned biomes (e.g. savanna, boreal forest, and others), rather than geographical area, and therefore we preferred to avoid and focus on the geographical area. For AIR, it is difficult to assign singularly to a region. It was not accurate in the manuscript that "AIR+BIB+BIO (as they cannot be separated into regions)", we now change it to "AIR+BIB+BIO (combined as one sector to reduce uncertainty)". We now add more clarification in section 2.2.

**L295-301 & L364-371: How the authors calculate the long-term trend and uncertainties? Please describe it somewhere.**

We now describe it in section 2.3 as follow:

"In this study, we also calculated simple linear trends (hereafter as linear trend to distinguish with the trends derived from "Prophet" algorithm) within a time period as follows:

Linear trend $= (c_{End} - c_{Start})/(t_{End} - t_{Start})$   (1)

where t$_{End}$ and t$_{Start}$ represent the end and start date and time of the target time period, $c_{End}$ or $c_{Start}$ is the mole fraction of trace gases (ethane, methane or propane) at the end or start date and time. ”

**L304-307: But as presented in L264-270 and Fig. 2b-d and, the pronounced peak in 2010-2011 for NAM, ASI, and EUR are mainly explained by SWD (solid waste and waste water) and TRO (road transportation). Are these results consistent with the sentence here? As seen in Fig.2a, the peaks in FEF and RES are actually seen in 2010-2011. Are these happened in ROW then?**

The sectoral emissions (e.g. FEF, SWD, TRO, etc) were optimized based on the input emission inventory. We see that the optimized modeled emissions (sum of all sectoral emissions) captured the similar ethane trends as from observations, whereas not for the peak in 2010-2011. We now include trends at ROW (in SI) to better understand this. We see a peak in 2010-2011 for ROW trend as well. Because about half of the samples were collected above ROW, the whole NH trend may be largely influenced by ROW, thus its trend and sectoral emission signature might be different from these at regional scale (NAM, EUR, ASI). We point this out in section 3.5 of limitations.

**L316-317: Which figure are you referring to?**

It refers to figure 5 which shows the observations and model simulations for ethane at surface stations. Now this figure has been moved to the Supplementary.

**L321-L333: Again how to estimate the optimised emissions? Just by comparing with the observation with model output with "try and error" method? Needs more explanation. Also I wonder how meaningful to present the emission amount under 2 decimal places. As long as I understand, the presented optimization method is very rough and seems not to be able to estimate within such small amount of emissions.**

We now explain how the model optimization was done in the manuscript, and reduce the decimal of emission amount.

**L329-330: But, please be noted that the atmospheric ethane, methane, and propane in the UTLS region can be largely influenced by atmospheric transport and chemical destruction processes.**

We now add this in section 3.5 to discuss limitations of our analysis. Section 3.5 reads:

"Despite the usefulness, uniqueness and high quality of our datasets, several limitations of our study should be noted. (a) representativeness of the presented trends. Although our flight sampling is frequent and covers a large area of the NH, the spatial and temporal distributions of our samples are not even. This may cause the trends being influenced by a specific regions where more samples were collected. (b) chlorine chemistry is missing in the EMAC model. Chlorine radicals are much more abundant in the stratosphere than the surface, thus the change in chlorine plays a great role in the observed trends. (c) our samples were collected in the UTLS region and can be influenced by atmospheric transport (e.g. troposphere-stratosphere exchange), surface sources, and chemical destruction processes. Therefore, the trends represent the net effects of these factors making the interpretation on a single factor difficult. (d) PV choice of identifying upper tropospheric and stratospheric samples. In this study, we used PV=2 to define the tropopause, whereas other approaches exist. It is shown that on large space and time scales in the extratropics, the WMO tropopause corresponds rather well to a surface of constant potential vorticity (PV), although there exist systematic differences on smaller scales (Stohl et al., 2003; Wirth, 2000). (e) trend analysis tool "Prophet". One needs some experience with the algorithm to choose and tune some parameters to get the best results for individual datasets, i.e. settings for our dataset may not be suitable for other datasets. (f) model optimization. Our EMAC model and input values for sectoral emissions have been examined and optimized in many previous studies, therefore, in this study we simply increased each emission sector by 45% to match the observations. The aim of model simulations is to better understand the contributions from each emission sector, rather than improving the performance of model and emission inventories. (g) interpretation of results. This article is designed as a data description article to provide high quality and useful dataset for scientific use. There are many interesting features in the presented trends to be explored, however, it is beyond the aim of this study.

**Implications.** (a) observations of ethane, methane and propane were often restricted at regional scale or short-duration. We presented a long-term (10 years) airborne observations of ethane, methane and propane in the UTLS region at northern hemispheric scale. This dataset is unique and can be used to examine long-term troposphere-stratosphere exchange, chemical and dynamical changes in the UTLS region, and improve model performance. To our best knowledge, such long-term aircraft observations are only available from IAGOS-CARIBIC project (our study) and CONTRAIL project (Machida et al., 2008; Sawa et al., 2015).(b) The

"Prophet" algorithm is an open source software, and suitable for non-continuous time-series datasets. Unlike the commonly used linear fit approach for trend analysis in other studies, the "Prophet" algorithm is robust to missing data and the influence from outliers is minimized. It better captures the inter-annual variability and is not influenced by the time period of choice. (c) other analysis approaches such as machine learning techniques can be used on our dataset to enlarge the spatial and temporal distributions. Combining our dataset with space-borne observations will provide a better view of global distributions and trends of trace gases."

**L336-337: It is not clear why the peaks in 2010 not seen at regional levels even though it was caused by global upward transport of the upper tropospheric ethane emissions. Maybe the contribution of ROW is important? Also see the comment in L207.**

We now add ROW trends in SI. We do see a peak in 2010 at ROW, which may suggest that the contribution of ROW is important. We now add this point in the manuscript.

**L345-350: Any possible explanation that the Cl in the stratosphere has increased after 2013? As mentioned here, the discrepancy between model and observation since 2013 is mainly seen only in ASI. Does it mean the Cl destruction process is only influential in the stratosphere in ASI?**

The reviewer has raised interesting scientific questions and discussions. According to the journal's author guidelines "Any interpretation of data is outside the scope of regular articles." We therefore limited our interpretation. Nevertheless, in answer to the point raised we now add more information and discussion. The discrepancy between model and observations since 2013 is seen in ASI but not NAM or EUR, we suspect that it is caused by regional signature in ASI. It has been shown that the global and Asian emissions of some chlorinated compounds (e.g. CFC-11, CHCl3) were increasing in 2012-2016 (Fang et al., 2019; Rigby et al., 2019), and strong chlorine chemistry associated with Asian outflow in the UTLS region in 2013 (Baker et al., 2016). Thus the Cl destruction of ethane in ASI may explain the discrepancy between model and observation. We now add more texts in the manuscript to clarify these points:

"Previous studies have shown that the global and Asian emissions of some chlorinated trace gases (e.g. CFC-11, CHCl3) were increasing during 2012-2016 (Rigby et al., 2019; Fang et al., 2019; Montzka et al., 2021), and a strong chlorine chemistry was associated with Asian outflow

in the UTLS region in 2013 (Baker et al., 2016).This could be an explanation for the larger discrepancy between model and observation since 2013 in ASI."

References:

Baker, A. K., Sauvage, C., Thorenz, U. R., van Velthoven, P., Oram, D. E., Zahn, A., Brenninkmeijer, C. A., and Williams, J.: Evidence for strong, widespread chlorine radical chemistry associated with pollution outflow from continental Asia, Scientific reports, 6, 1-9, 2016.

Fang, X., Park, S., Saito, T., Tunnicliffe, R., Ganesan, A. L., Rigby, M., Li, S., Yokouchi, Y., Fraser, P. J., and Harth, C. M.: Rapid increase in ozone-depleting chloroform emissions from China, Nature Geoscience, 12, 89-93, 2019.

Rigby, M., Park, S., Saito, T., Western, L., Redington, A., Fang, X., Henne, S., Manning, A., Prinn, R., and Dutton, G.: Increase in CFC-11 emissions from eastern China based on atmospheric observations, Nature, 569, 546-550, 2019.

Montzka et al. (2021). A decline in global CFC-11 emissions during 2018− 2019. Nature, 590(7846), 428-432. https://doi.org/10.1038/s41586-021-03260-5

**L373-374: But why it is not seen in methane? Needs more explanation.**

Different from ethane and propane, methane has more complex emission sources. These sources can originate from different regions, which makes the interpretation more difficult. As replied to previous comments, the whole NH trends can be largely influenced by ROW, thus we suspect that maybe some unique sources of methane at ROW were changing differently as at other regions.

**L374-375: Any possibility of ethane destruction by Cl increased after 2013? For example, the emissions CFCs from eastern mainland China has increased after around 2013 (Rigby et al. 2019; Montzke et al. 2021), which may have affected the increase of Cl production in the stratosphere and may have caused the larger destruction of ethane in ASI?**

As replied to a previous comment, we add the following text in the manuscript:

"Previous studies have shown that the global and Asian emissions of some chlorinated trace gases (e.g. CFC-11, CHCl3) were increasing during 2012-2016 (Rigby et al., 2019; Fang et al., 2019; Montzka et al., 2021), and a strong chlorine chemistry was associated with Asian outflow in the UTLS region in 2013 (Baker et al., 2016).This could be an explanation for the larger discrepancy between model and observation since 2013 in ASI."

**L379 Data availability: Please include the PV values, as well as other observed meteorological information, into the dataset for each air sample. These are important for readers to know how the air masses are classified into troposphere or stratosphere. Also please add the measurement information in more detail (e.g., calibration scales, precision, measurement number, standard deviation (if exists), and reference lists of the measurements). For the simplicity and usability, I would even suggest that the tropospheric and stratospheric data sheet can be combined into one sheet with timeseries, just by adding a flag column like "trop" or "strat".**

We now change the dataset as the reviewer suggested.

The new link for the dataset: https://zenodo.org/record/6301729

**Figure 2: Gray lines are not plotted in Fig 2b–d**

We now add gray lines for all the subplots.

**Figure 3: I'm not sure how this figure could be useful for readers. It is said the vertical scale of each source sector represents the source contributions, but the contributions to each geographical region shouldn't be the same.**

We realize that this figure can cause confusion, therefore, we moved it to the Supplement and add more explanatory text. The right side of the figure shows the contributions of all sectors for each region, assuming 100% for each region which is why the sizes of them are the same. We were trying to show the relative contributions among sectors for each region. Unfortunately, we cannot derive emissions of each sector for troposphere and stratosphere separately from our model.

**Figure 7: First I wondered why the authors need to show the "AIR + BIB + BIO_opt" here, with geographical sector contributions. After reading the sentence L283-284, I found it is said "AIR+BIB+BIO" cannot be separated into regions. But, then I wonder how they are included into the model simulation.**

We now remove the "AIR + BIB + BIO_opt" from those figures.

**Technical corrections**

**L19: As already commented, I don't think the two decimal place is meaningful under the current rough model optimization method.**

Done.

**L56: Tg/yr2 ?**

We present the ethane emission growth rate which in unit of Tg/yr.

**L394: Same as L19.**

Done.

**Figure 5: There are two Figure 5!**

Done.

**References**

**Baker, A. K., Slemr, F., & Brenninkmeijer, C. A. M. (2010). Analysis of non-methane hydrocarbons in air samples collected aboard the CARIBIC passenger aircraft. Atmospheric Measurement Techniques, 3(1), 311-321. https://doi.org/10.5194/amt-3-311-2010**

**Schuck, T. J., Brenninkmeijer, C. A. M., Slemr, F., Xueref-Remy, I., & Zahn, A. (2009). Greenhouse gas analysis of air samples collected onboard the CARIBIC passenger aircraft. Atmospheric Measurement Techniques, 2(2), 449-464. https://doi.org/10.5194/amt-2-449-2009**

**Machida et al. (2008). Worldwide measurements of atmospheric CO2 and other trace gas species using commercial airlines. Journal of Atmospheric and Oceanic Technology, 25(10), 1744-1754. https://doi.org/10.1175/2008JTECHA1082.1**

Sawa et al. (2015). Seasonal changes of CO2, CH4, N2O, and SF6 in the upper troposphere/lower stratosphere over the Eurasian continent observed by commercial airliner. Geophysical Research Letters, 42(6), 2001-2008. https://doi.org/10.1002/2014GL062734

Pozzer, A., Schultz, M. G., & Helmig, D. (2020). Impact of US Oil and natural gas emission increases on surface ozone is most pronounced in the Central United States. Environmental science & technology, 54(19), 12423-12433. https://doi.org/10.1021/acs.est.9b06983

Park et al. (2007). Transport above the Asian summer monsoon anticyclone inferred from Aura Microwave Limb Sounder tracers. Journal of Geophysical Research: Atmospheres, 112(D16). https://doi.org/10.1029/2006JD008294

Xiong et al. (2009). Methane plume over south Asia during the monsoon season: satellite observation and model simulation. Atmospheric Chemistry and Physics, 9(3), 783-794. https://doi.org/10.5194/acp-9-783-2009

Rigby et al. (2019). Increase in CFC-11 emissions from eastern China based on atmospheric observations. Nature, 569(7757), 546-550. https://doi.org/10.1038/s41586-019-1193-4

Montzka et al. (2021). A decline in global CFC-11 emissions during 2018− 2019. Nature, 590(7846), 428-432.

---

## Referee Report (RR1)

**Review comments on "Northern hemispheric atmospheric ethane trends (2006-2016) with reference to methane and propane"**

Author(s): Mengze Li et al.

MS No.: essd-2021-246

MS type: Data description paper

**General comments**

Li et al. present long-term ethane mole fraction data obtained from airborne observation in the UTLS region in the Northern Hemisphere through IAGOC-CARIBIC project for the period 2006–2016. The authors also present simultaneously measured propane and methane mole fraction data as well as output of simulated atmospheric ethane using the EMAC model to understand the contributions of different source regions/sectors to the observed ethane variations.

After the revision, the manuscript and dataset has been generally improved, but at the same time it has also revealed significant issues for publication. Before going into the details, I would like to emphasize again that the observational ethane, propane, and methane data obtained from this project is very unique and useful for the scientific community, because there exist only limited airborne observation platforms that can collect the air samples in the UTLS region and can measure ethane, propane, and methane with high quality over the 10 years. I think the data can used by many researchers in this field after the issues as below have been resolved.

1) The current manuscript is mainly focusing on the comparison of the trend components derived by Prophet algorithm between observations and model outputs, which even precedes the presentation of original observational data. I think this has hidden the overall features of dataset as well as data quality and is not well suitable for the data description paper. I would like to ask the authors to pay more attention about the observation data itself (i.e., rigorous data classification in terms of altitudes, latitudes, longitudes, PV, and etc, also see my general comment 2). Now, the authors have newly added the timeseries of raw measured data together with their estimated trend by the Prophet (Fig 5 and Fig 8 and other figures in the supplement). The figures are very useful and helpful to understand the overall structure of dataset – for example, how the sampling frequency is for 2006-2016 in

this project, how the variance of the raw data is, and how the trend is properly estimated. For the future revision, I would like to suggest that the figures are presented before presenting the comparison with model nor respective components by Prophet, as long as the authors purpose is the publication of their observational dataset. I think to guarantee the quality of the dataset it depends on whether the QC of the observation data is done properly and the presentation of observation data itself is reasonable, rather than the comparison with the model output.

2) The current data classification in terms vertical axis is based on whether it is lower or higher than 2 PVU only. I found the data regarded as upper troposphere contains even much lower altitudes than expected (946 m at minimum). The change of such spatiotemporal sampling density has significantly affected the estimated trends and conclusion. For example, one of the main conclusions of this paper (and dataset) is that a sharp increase of atmospheric ethane was observed in the upper troposphere in 2010-2011, the cause of which is estimated to be fossil fuel related sources. However, according to the Fig 5 and Fig 8, it is found there is no observation data around half month before and after 2010, and then two flight data just after the sampling program restarted in 2010 showed overall higher mole fractions with relatively small variance, which looks affecting the rapid change of trend estimated by Prophet (Fig 5). I found most of these flight data were mainly sampled at altitudes lower than 8000 m (please see the attached Figure A). I would say such an increase in 2010 is not trend in the upper troposphere nor caused by the fossil emission increase, but is just the reflection of the vertical profile in the trace gases (see Figure B). Similar points can be said to the stratospheric data because it contains different strength of stratospheric data (from 2 PVU to ~12 PVU, see Figure C-D). Please reconsider how the data classification is reasonable to be presented as the data description paper.

3) Under the temporal sparseness of data in several years and in several regions, the analysis using Prophet would have underestimated the uncertainty in estimated trend. I don't think the uncertainty can be evaluated just by changing the fitting parameters "changepoint_prior_scale" (as authors did), because the estimated trend cannot usually avoid the overfitting when there is a large temporal data gap. If the authors would still keen to estimate the trend from Prophet, I would like to request to do some bootstrap tests, for example, jackknife method, to evaluate the robustness of estimated trend. Such tests should be especially important when estimating the trends for each geographical region, due to the sparseness of their

data compared with whole NH dataset (i.e., ASI in the upper troposphere and NAM, ASI, and EUR in the lower stratosphere). Any discussions and interpretations on the seasonality and trend (and comparison with model) should be made after such cautious statistic test is done.

4) I think the procedure of model optimization is not so unique nor well organized, which makes me wonder if the model output is useful for understanding observation data and is worth publication, contrary to the significance of observation data. If the authors main purpose is to publish the observational data, I would suggest they focus more on the presentation of observational data itself and statistical analysis after rigorous data classification and may remove the model output from the publication of data. Otherwise, if their focus is both the observation data and model output, I would suggest they develop and sophisticate their model optimization procedure for the future use of model data.

Considering the above, I would recommend this paper for publication after major revisions.

**Specific comments**

Overall: There are many confusions between "mole fraction" and "trend". After reading through the manuscript several times, I can imagine the authors somewhere refer to the trend component derived from Prophet as trend (e.g., Fig 3-8 and related sentences), but it is not always easy for readers to follow because we generally expect the ethane trend as in Table 2 or its related sentences (the unit, %/year or ppt/year). Also there still exists multiple places where it is hard to distinguish if the authors are referring to modeled results or observations.

L25-L26: Now I wonder if the statement of emission estimate can be removed because the aim of this study using model is not to validate the emission inventories and also because the emission optimization methodology is not well sophisticated.

L28 "An ethane plume": I would recommend replacing it by "An sharp increase of ethane". "An ethane plume" sounds like more short-term intrusion of polluted air masses but here it seems saying about inter-annual variations. However, as I

commented, I think it is also reflected by changes in sampling altitudes (please see general comment 2).

L29-31 "and higher temporal-spatial resolution data of ethane are needed.": I would suggest deleting it here since there is no discussion about the issues of spatiotemporal resolutions of emission inventories in the main text.

L52-L76: As long as I understand, the author is reviewing the study of emissions estimates from top-down approach in this paragraph, and discussing the issue of discrepancy between top-down and bottom-up estimates in the next paragraph. I would suggest clarifying it since there are several confusions as below.

L54-L56: The sentence can be deleted since this paragraph is the summary of top-down approach (as long as I understand); thus the discussion on inventory is just confusing.

L66: Insert "based on top-down approach" after "global emission estimates".

L76-L80: This sentence is not clear. Please rephrase.

L108-L109: By inspecting the dataset, I found 946 m sampling altitude data exists as the minimum altitude. I would strongly recommend the author presents the information on the frequency of sampling altitudes and reconsider the data classification.

L109-L114: I would suggest the author explains the measured trace gases by each three detectors separately or just explains the measurements of ethane, propane, and ethane, respectively. Under the current writing, the reader cannot still find the correspondence between each detector and measured trace gas.

L135-L137 "Three injections of calibration standards": What does it mean? Need clarification.

L164-L165 "to reduce the uncertainty": What is the "uncertainty" here? Why combining AIR, BIB, and BIO into one sector can reduce the "uncertainty"?

L166-L176: Actually, this part is still not clear and confusing to me.

L168-L172: What is "the total amount"? The global total anthropogenic ethane emission? Despite the optimization, it is said the value used in this study is ~11.8 Tg/yr whereas the value used in Pozzer et al. (2020) is 13.2 Tg/yr? Are these global total ethane emissions including natural sources; thus they are not consistent even after the adjustment?

L173-L174 "We further optimized …": Do you mean the 45% increase is done from CAMS-GLOB-ANT v4.2 inventory itself or from that after multiplied by 2.47 as mentioned above? In the current manuscript, it sounds like the authors attempted to optimize the global total ethane emissions in the latter (two-step) method. If the optimization method is the former, I would suggest deleting or much reduce the sentence in L168-L172 to avoid confusion and clarity.

L174-L176: What is "whole dataset"? Do you mean many iterations were performed by changing the scaling factors to determine the minimum of RMSE? Please clarify more. The authors didn't present the raw output of model simulations, but I think at least it is desirable to present the correlation plot between observations and model outputs after the optimization (maybe in the supplement).

L205-L208: Please see my general comment 3. I wonder if this approach is reasonable when estimating the trend uncertainty in this case whose data is sparse in time.

L210-L213: What is "start" and "end" representing here? For example, in L366-L368, how the start and end of the data were determined? Is it also applied to the output derived from Prophet or applied to the raw observational data? Is it strongly dependent on the measurement data at the $t_{End}$ and $t_{start}$.

L216-L251: The two paragraphs are just listing of information and hard to read. Please summarize and be more concise. The descriptions of emission estimates are also mixed in L232-L234 and L234-L235. Since the authors intended to summarize atmospheric ethane trend here, the sentence of emissions can be removed for the readability. The headline can also be changed as "Literature perspective of global atmospheric ethane trends" in this case.

L272-L273: Any possibility that the sampling altitudes in EUR are lower than the other regions? As I commented in L108-L109, the information of sampling altitudes for each region is also necessary to be presented (and please reconsider the data classification when needed).

L285-L359: Please see my general comment 1. I think the comparison of estimated ethane trends between observations and model (Figure 3-4) should be followed by the presentation of raw observational data itself (Figure 5). This is important for potential data user because they use the raw data not the estimated trend derived by Prophet.

L309-L311: Delete?

L323-L324: As commented in L28, the annual increase is not short-term.

L329-L331: This sentence is not clear to me. Please rephrase it.

L333-L334: FEF looks rather increasing over 2011-2013 in Figure 3a.

L335-L336 "We note…": This sentence is just suddenly appearing here. Delete?

L337-L346: This may not be an important finding from this study and can be removed. The author didn't optimize each emission sector but just roughly scaled up by 45% from CAMS-GLOB-ANT v4.2?.

L366-L374: Please see the comment in L210-L213. Also how to estimate the max and min?

L377-L380: As commented, the high mole fractions in the first two flights in 2010 would have been due to the lower sampling altitudes. I think it is an overinterpretation to relate it to the increase of fossil emissions in 2010-2011 at this stage. After data classification with altitudes, it would be useful to present some scatter plots of ethane-methane or ethane-propane when the ethane mole fractions are high levels in order to relate with fossil sources.

L389-L391: I wonder why the authors write this here. As long as I understand, here the authors are attempting to present not the accuracy of their model, but the validity of their optimized global total ethane emission?

L407-L431: Please see my general comment 3. Please reconsider investigating the uncertainty before going into the detailed discussion even if the authors still wish to keep these parts.

L463-L464: I think this is an important issue on the data classification, not limitations. Also, as I commented, the issue of temporal data gap should also be critical when estimating the trend.

L472 "the WMO tropopause": Replace it by "the WMO thermal tropopause".

L474-L476 & L481-L483: I don't think the authors need to add these sentences.

L501-L504: Please add the information of the calibration scale, instrument, measurement precision, the number of measurements, and standard deviations into the dataset.

L510-L512: Please clarify more.

L515-L516: I think the remarks on emission estimates need to be minimized because the current optimization method is very rough and the authors are also saying the aim of model simulations is to better understand the contributions from each emission sector, not improving the emission inventories.

L518-L519 "a factor of three": As long as I read through the paper, the author has increased the global ethane emissions by 45% from CAMS-GLOB-ANT v4.2 to match their atmospheric ethane observations, not by a factor of three.

L526-L527: There is no presentation on the simulation results of methane in this study. It sounds very sudden.

Table 1: Please specify "inventory" as CAMS-GLOB-ANT v4.2. Also the row of "Biogenic emission" and "Biomass burning" also needs a headline. Please add the explanation that the sum of BIO + BB and (a) or (b) is equal to total sources.

Table 2 "~": What does it means? The uncertainties presented here represent the same manners? Also please unify the usage of parentheses in the first column; otherwise add the new column to describe the locations of measurements?

Figure 5: The trends in Propane are difficult to distinguish (also in Figure 8, S6-S9). As commented, these time series plots are very important and preferred to come first rather than comparisons with model outputs. Also useful to show the sum of all components (i.e. best fit curve) derived from Prophets.

Figure S3: Not clear the meaning of "trend analysis uncertainty". What is the meaning of percentage here? The uncertainty range is generally smaller for the period 2006–2009 when the number of data is small, potentially due to the over-fitting to the sparse data?

Figure S4: I don't think this figure is necessary even in the supplement. The contributions of each emission sector to the atmospheric simulated ethane trends are already shown in Figure 3 and Figure 6. This figure looks rather confusing.

Table S2, Figure S5, Figure S11: Isn't it the sectoral contribution to "simulated ethane mole fractions" not "ethane trends"? If it is saying the values of trend component derived from Prophet, please clarify so.

Figure S6-S9, S12-S15: Same as the comments in Figure 5.

Figure S7 and S13: It looks the observational data number is very sparse and difficult to estimate the robust trend? Need further statistical test to investigate the effect of time data gap (e.g., bootstrap method) if the authors still wish to present their trends.

Figure S12-S14: There are vertical lines in the left sides. Need removed.

**Technical corrections**

L52: Insert "atmospheric" before "observations".

L253 "It is noted …above, any…": Conjunction is required.

L253-L257 "…of data quality, a process": Also need conjunction.

L273-L278: Very long sentence. Please separate it into two sentences.

L459-L497: I doubt the writing style in this section is following a proper manner. There are headlines starting from alphabet in parenthesis - some are complete sentences and some are not but all headlines start with lower cases.

L499: 4 Data availability.

L507: 5 Conclusions

Figure 2: Insert "observed" before "over the whole …".

[Figure]

Figure A: The observed ethane, methane, propane mole fractions for the whole NH upper troposphere (same as Figure 5 in the authors' manuscript), together with sampling altitudes.

[Figure]

Figure B: Same as Figure B, but for sampling altitudes over 8000m.

[Figure]

Figure C: The observed ethane, methane, propane mole fractions for the whole NH lower stratosphere (same as Figure 8 in the authors' manuscript), together with PV.

[Figure]

Figure D: Same as Figure C, but for PV over 6 PVU.

---

## Referee Report (RR2)

**GENERAL COMMENTS**

After carefully revising all the responses to the comments made to reviewers, I see that the authors have done a thorough revision of the previously submitted manuscript. Information regarding the model setup and important descriptions in the methodology were added. Also, numerous changes to the discussion section were made. The revised manuscript was greatly improved by the new information, clarifications, and changes to the title.

That being said, I have significant concerns regarding the model setup as described in the revised manuscript and concluded that their modeling approach has serious flaws, which are then reflected in the results. The authors used to run their model an ethane emission inventory (Table 1) that does not reflect the current scientific knowledge of 1) the sectoral distribution (percent-wise), 2) the magnitude of the emissions by sector and region, and 3) the geographical distribution of emissions for the represented period (2006-2016). I will further explain these:

1 and 2) Some emission sectors and their associated emission estimates are unrealistic (e.i. SWD-solid waste and wastewater, TNR, TRO). Just as the author responded to line 269 in response to reviewer comments, "No independent estimation of SWD contribution was found in the literature". And even so, authors included that sector without justification with emissions equal to other well-known sources such as biomass burning-BIB.

2 and 3) The geographical distribution of anthropogenic emissions presented in Table 1 is not based on any study, and no justification is offered for such values. Asian emissions are too high compared to North America and Europe (Xiao et al., 2008, Tzompa-Sosa et al., 2017). Furthermore, emissions from the ROW-Rest-of-the-World (excluding Asia, Europe, and North America) are estimated to contribute about ~17% to global emissions, whereas Mengze et al. report a 30% contribution, again without providing any justification.

The lack of attention to these characteristics in the model input ethane emissions ultimately diminishes the quality of the results and conclusions therein. Currently, the model simulations constitute a fundamental part of the manuscript since they were used to determine the upper-tropospheric and low stratospheric trends. It is no surprise that the model did a poor job determining regional trends (Lines 327-328 from the red-line manuscript essd-2021-246-ATC2.pdf). It is important to note that the global ethane emission in the optimized model simulation (19.3 Tg/yr) is similar to other studies (Franco et al. 2016), and this could explain why the model did a better job at determining the NH upper tropospheric trend (lines 326-327).

Overall, I can see a significant scientific value in the measurement dataset they present and the linear trends derived from the observations only. I suggest either removing all the sections and conclusions related to the model results or re-run the model with science-based emissions of ethane, methane, and propane. Additionally, it would be interesting to run the model using the trend derived from observations and see how similar or different the model is compared to observations. Major changes need to be made to this manuscript before acceptance.

Line 324. "The model incorporates all known emissions via emission inventories so any..."

Suggest changing the wording to "The model incorporates emissions from various emission inventories described in table XX". In that table, provide a list of all the names (with references) of the emission inventories used for each relevant compound (i.e., ethane, methane, propane). Also, include emission totals by region and sector for methane and propane since both compounds are mentioned in the paper (similar to Table 1).

Authors need to avoid subjective adjectives such as "well" (line 327), "good" (line 409) and instead provide values for every comparison made.

Suggest adding "in the upper-troposphere and lower-stratosphere" in the title.

Line 160 & line 490. Does the model include chemistry for other halogens, such as Br? Also, suggest adding a reference to Sherwen et al. [2016] that concludes that Cl may be an important $C_2H_6$ sink that can decrease the simulated global burden of $C_2H_6$ by about ~20%. Also, add other studies that estimated the impact of the lack of other halogens like Br.

Line 175. It seems that there is confusion between the mole fraction term and emissions. The emissions were optimized to match the observed mole fractions. Thus, "We further optimized ethane mole fractions for each emission section" is technically incorrect.

Line 179. What is the value of this minimum RMSE?

Lines 209-210. If the Prophet algorithm is designed for non-continuous datasets, why do results match with the NOAA algorithm even though it is designed for continuous datasets? Is this an expected result, or what does having similar values mean?

The entire Section 3.1 should be moved to the introduction. Also, the first paragraph of section 3.2 should be moved to the introduction. The time of collection and location of the sampling should not be in the discussion but in the introduction and methodology section.

Lines 277-278. Are there any studies that talk about a weaker mixing of stratospheric air over EUR? If so, add references.

Lines 304-307. Can authors provide values for trends due to sampling location vs. trends derived from observations to support this conclusion?

Lines 338-371. As suggested in the general comments, model sectors and model geographical sectors (section 3.3.2) contribution analyses should be deleted because sources, totals, geographic distributions, and trends do not reflect current scientific knowledge.

Lines 390-392.  Having peaks in all regions does suggest regional and global increases in emissions, but not an increase in fossil fuel emissions. Thus, remove "fossil fuel" from this sentence.

Lines 412-414. This result should show the estimated emission in 2006 and the subsequent increase due to the increasing trend derived from observations. Otherwise, state the exact year at which those estimated emissions by sector correspond.

Line 541. Remove the word "budget". Also, see the previous comment on how to present this trend value. Possibly write it as "averaged" emission for the same period.

Line 545. The underestimation is only ~30% based on the optimized model emissions (13.3 Tg vs. 19.3 Tg). When re-writing this sentence, make it clear which emissions this comparison refers to.

Lines 551-552. "…was caused by fossil fuel-related emissions, likely from oil associated and natural gas sources". The conclusion of the 2010-2011 peak coming from oil and gas emissions is stated in the text (line 390) only as a possible explanation and is not based on any of the main results (see Fig 3) and discussions made in the text (lines 330, 390, 430, 478). Even figure 3 (lower panel with model optimized sectoral emissions) does not reflect this statement, as other sectors such as SWD, RES, and TRO show the same increase in 2010-2011. Therefore, this conclusion should be removed from here and from the abstract since the authors have no strong results to reach such a conclusion in the manuscript.

Lines 552-553. "The global ethane trends cannot be well simulated by advanced atmospheric chemistry modeling ". This is a strong claim that is not supported in the text. If there are clear examples of studies that have had struggles modeling ethane trends, they should be cited here and in the text so that this conclusion is consistent with the main document.

**TECHNICAL CORRECTIONS**

Line 174- Move "(13.2 Tg/yr)" to the end of line 172.

Line 187- Add upper case in "Real..."

Line 321. Model estimation or optimized model results?

Add the word "modeled" every time the text refers to regional trends. For example, in line 475, it should say: "The modeled regional trends..."

---

## Referee Report (RR3)

**GENERAL COMMENTS**

After the second revision, the authors followed the suggestions made by reviewers and eliminated the model simulation previously included. The new manuscript focuses on the observational dataset and the trends derived from it.

This new manuscript is well written and the methods and results are presented in a clear manner. I recommend this paper for publication after minor revisions.

**SPECIFIC COMMENTS**

Line 90. Suggest changing the second part of this statement to: "…, including regions without ground measurements."

Line 146-150. If the term "whole NH" will be used, it is better to change the name of ROW to RNH "Rest of Northern Hemisphere" and make sure data from the Southern Hemisphere is not included. The way it is written right now gives the impression that some data in ROW corresponds to the Southern Hemisphere, and thus, combining the four regions into one does not make sense.

Line 154. Change "must" for "does".

Lines 194-201. These lines should be moved to the methods section (2.1).

Line 238. Specify what you mean by "low" (maybe add a cut-off number).

Section 3.3. Add information regarding other halogens like Bromine.

Line 285. Define "large". Provide a value for this difference.

Line 388. Since some values are negative, "growth rates" should be changed for "annual rates of change in atmospheric abundances of".

---

## Referee Report (RR4)

Manuscript Review for ESSD

Li et al.,: Northern hemispheric atmospheric ethane trends in the upper troposphere and lower stratosphere (2006-2016) with reference to methane and propane

This manuscript is a revision/resubmission of a previously evaluated submission to the same journal. The authors made several changes from the prior submission. Despite their efforts, in my opinion, some important shortcomings remain. Concerns by Reviewer #3 about the classification and filtering of air samples have not been fully resolved. I have also reviewed comments and revisions from the very first submission. Several of the previously articulated problems are still valid:

As pointed out in the prior reviews and comments, there appears to have been a minimum in the atmospheric mole fraction of ethane around the year 2009-2010, with declining levels prior, and increasing levels afterwards. These trends were seen consistently in monitoring records from the Northern Hemisphere. With this having been well established, it does not seem appropriate to conduct linear trend analysis across this minimum. Obviously, the trend results will then all depend on which starting and end time one chooses before and after the minimum. Despite this having been emphasized before, the authors adhere to this analysis and dedicate a full section to this method and discussion of the results.

The second problem I see, and this is a big one, is that to the best of my understanding and evaluation, it seems that the data that are used for attempting these trend analyses are ways to variable in its source and representation of a particular altitude (troposphere versus stratosphere) as well as latitude for justifying this analysis. The authors conducted filtering of their data based on polar vorticity (PV) and continental areas. I don't think this worked. The data that are displayed in Figures 2 and the derived trend results in Figure 3 and 5 defy the atmospheric behavior of methane, ethane, and propane that has been well established in other literature. Ethane is a relatively long-lived gas in the atmosphere, with its lifetime being on the scale of or longer than the mixing time scale in the Northern Hemisphere troposphere. Therefore, it does not seem plausible that upper troposphere ethane would behave so completely different than ethane in mid to lower tropospheric records. I have pulled out some data examples in the figures below, i.e. from Mauna Loa Observatory, HI, USA, Jungfraujoch, Switzerland, and Summit, Greenland. There are many more that can be easily reproduced from records in the World Data Center for Reactive Gases (EBAS/Nilu). I have intentionally chosen examples from high altitude monitoring stations as those are deemed to best describe the behavior in the lower free troposphere. The important take home messages are that seasonal cycles of ethane are very regular. Year-to-year changes are relatively small, just a few percent, if at all. These are all deemed high quality observations, reflecting the seasonal and year-to-year changes at a fixed site. However, there is a very substantial latitudinal gradient, with ethane dropping steeply from mid-latitudes to the tropics (see Figure 4).

The data presented in this paper look very different, showing a much higher variability and much higher year-to-year changes, which the authors interpret as atmospheric concentration trends and attempt to relate to surface emissions changes. I don't think this is appropriate and correct. I think it is much more likely that the variability seen in these data to a large extend reflects differences in the origin of the air masses that were sampled, both in altitude (tropospheric versus stratospheric contribution) and with latitude. What the authors interpret as trends likely are year-to-year changes in the fraction of collected samples having one sort of these influences versus another.

I don't think the paper is publishable as it is.  Since this journal focuses on presentation of data, a way to proceed may be to just publish the data by themselves without the trends analyses.

Another, totally different approach might be to focus instead on the analysis of ratios of the hydrocarbon over methane.  One could also utilize the propane/ethane ratio as an indicator of photochemical age and possibly derive a fractional contribution of stratospheric air on the samples and then compare that with the PV rating. Or maybe one could attempt filtering the data based on these ratios and attempt a trend analysis on the ratio-filtered data. But all of that may be well beyond the scope of an ESSD publication and be better suited for a regular Atmospheric Science journal.

[Figure]

Figure 1:  Ethane atmospheric mole fractions (red crosses) at Mauna Loa Observatory (3397 m asl, data and graph retrieved from https://www.nilu.no/).

[Figure]

Figure 2:  Ethane atmospheric mole fractions (red crosses) at Jungfraujoch (3463 m asl, data and graph retrieved from https://www.nilu.no/).

[Figure]

Figure 3:  Ethane atmospheric mole fractions from flask sampling (red dots) and in-situ monitoring (black dots) at Summit, Greenland (3200 m asl, graph reproduced from WMO Reactive Gases Bulletin (https://library.wmo.int/index.php?lvl=notice_display&id=19877#.Yr9JIXbMKUk ).

[Figure]

Figure 4:  Latitudinal gradient and seasonal cycle of atmospheric ethane (graph reproduced from WMO Reactive Gases Bulletin (https://library.wmo.int/index.php?lvl=notice_display&id=19877#.Yr9JIXbMKUk )).

---

## Author Response (AR2)

**Reviewer #1 (Referee-report #2)**

GENERAL COMMENTS

After carefully revising all the responses to the comments made to reviewers, I see that the authors have done a thorough revision of the previously submitted manuscript. Information regarding the model setup and important descriptions in the methodology were added. Also, numerous changes to the discussion section were made. The revised manuscript was greatly improved by the new information, clarifications, and changes to the title.

That being said, I have significant concerns regarding the model setup as described in the revised manuscript and concluded that their modeling approach has serious flaws, which are then reflected in the results. The authors used to run their model an ethane emission inventory (Table 1) that does not reflect the current scientific knowledge of 1) the sectoral distribution (percent-wise), 2) the magnitude of the emissions by sector and region, and 3) the geographical distribution of emissions for the represented period (2006-2016). I will further explain these:

1       and 2) Some emission sectors and their associated emission estimates are unrealistic (e.i. SWD-solid waste and wastewater, TNR, TRO). Just as the author responded to line 269 in response to reviewer comments, "No independent estimation of SWD contribution was found in the literature". And even so, authors included that sector without justification with emissions equal to other well-known sources such as biomass burning-BIB.

2       and 3) The geographical distribution of anthropogenic emissions presented in Table 1 is not based on any study, and no justification is offered for such values. Asian emissions are too high compared to North America and Europe (Xiao et al., 2008, Tzompa-Sosa et al., 2017).

Furthermore, emissions from the ROW-Rest-of-the-World (excluding Asia, Europe, and North America) are estimated to contribute about ~17% to global emissions, whereas Mengze et al. report a 30% contribution, again without providing any justification.

The lack of attention to these characteristics in the model input ethane emissions ultimately diminishes the quality of the results and conclusions therein. Currently, the model simulations constitute a fundamental part of the manuscript since they were used to determine the upper- tropospheric and low stratospheric trends. It is no surprise that

the model did a poor job determining regional trends (Lines 327-328 from the red-line manuscript essd-2021-246- ATC2.pdf). It is important to note that the global ethane emission in the optimized model simulation (19.3 Tg/yr) is similar to other studies (Franco et al. 2016), and this could explain why the model did a better job at determining the NH upper tropospheric trend (lines 326-327).

Overall, I can see a significant scientific value in the measurement dataset they present and the linear trends derived from the observations only. I suggest either removing all the sections and conclusions related to the model results or re-run the model with science-based emissions of ethane, methane, and propane. Additionally, it would be interesting to run the model using the trend derived from observations and see how similar or different the model is compared to observations. Major changes need to be made to this manuscript before acceptance.

We thank the reviewer for the second-round comments on our manuscript, we find the comments and suggestions very helpful.

Both reviewers #1 and #3 have suggested the removal of all model simulations. After careful consideration, we have decided to do this so as to focus on the observations only. We have therefore made major changes to the manuscript to describe and explore the observations in greater detail.

SPECIFIC COMMENTS (LINE NUMBERS CORRESPOND TO THE RED-LINE DOCUMENT essd-2021-246-ATC2.pdf)

Line 324. "The model incorporates all known emissions via emission inventories so any…"

Suggest changing the wording to "The model incorporates emissions from various emission inventories described in table XX". In that table, provide a list of all the names (with references) of the emission inventories used for each relevant compound (i.e., ethane, methane, propane). Also, include emission totals by region and sector for methane and propane since both compounds are mentioned in the paper (similar to Table 1).

This is now being removed from the manuscript because it is related to model setup and simulation.

Authors need to avoid subjective adjectives such as "well" (line 327), "good" (line 409) and instead provide values for every comparison made.

Done. We have carefully checked the manuscript to avoid subjective description.

Suggest adding "in the upper-troposphere and lower-stratosphere" in the title.

Done. Now the title reads: "Northern hemispheric atmospheric ethane trends in the upper troposphere and lower stratosphere (2006-2016) with reference to methane and propane"

Line 160 & line 490. Does the model include chemistry for other halogens, such as Br? Also, suggest adding a reference to Sherwen et al. [2016] that concludes that Cl may be an important C2H6 sink that can decrease the simulated global burden of C2H6 by about ~20%. Also, add other studies that estimated the impact of the lack of other halogens like Br.

This is now being removed from the manuscript because it is related to model setup and simulation.

Line 175. It seems that there is confusion between the mole fraction term and emissions. The emissions were optimized to match the observed mole fractions. Thus, "We further optimized ethane mole fractions for each emission section" is technically incorrect.

This is now being removed from the manuscript because it is related to model setup and simulation.

Line 179. What is the value of this minimum RMSE?

This is now being removed from the manuscript because it is related to model setup and simulation.

Lines 209-210. If the Prophet algorithm is designed for non-continuous datasets, why do results match with the NOAA algorithm even though it is designed for continuous datasets? Is this an expected result, or what does having similar values mean?

Because the Prophet algorithm is the only product that we know which works for non-continuous datasets, we cannot test and compare it directly with other algorithms using a non-continuous dataset. The NOAA algorithm has been examined many times in the literature and it has advantages in deriving trends for continuous datasets. We therefore used a continuous dataset to test the performance of Prophet versus NOAA algorithms. The agreement of results from both algorithms (as we expected) indicates that Prophet algorithm works for continuous datasets, and may indirectly prove its performance for non-continuous dataset. In another words, if Prophet algorithm didn't match with NOAA algorithm using a continuous dataset, it means that Prophet algorithm may not even be able to work for our non-continuous dataset. That's why we did this comparison.

The entire Section 3.1 should be moved to the introduction. Also, the first paragraph of section 3.2 should be moved to the introduction. The time of collection and location of the sampling should not be in the discussion but in the introduction and methodology section.

Done. We have moved the section 3.1 and the first paragraph of section 3.2 to the introduction. We have also removed the paragraphs in the introduction that are related to model and emission inventories.

Lines 277-278. Are there any studies that talk about a weaker mixing of stratospheric air over EUR? If so, add references.

We did a literature search but not find any studies reporting a weaker stratospheric mixing over EUR.

Lines 304-307. Can authors provide values for trends due to sampling location vs. trends derived from observations to support this conclusion?

This is now being removed from the manuscript because it is related to model setup and simulation.

Lines 338-371. As suggested in the general comments, model sectors and model geographical sectors (section 3.3.2) contribution analyses should be deleted because

sources, totals, geographic distributions, and trends do not reflect current scientific knowledge.

This is now being removed from the manuscript because it is related to model setup and simulation.

Lines 390-392. Having peaks in all regions does suggest regional and global increases in emissions, but not an increase in fossil fuel emissions. Thus, remove "fossil fuel" from this sentence.

This is now being removed from the manuscript because it is related to model setup and simulation.

Lines 412-414. This result should show the estimated emission in 2006 and the subsequent increase due to the increasing trend derived from observations. Otherwise, state the exact year at which those estimated emissions by sector correspond.

This is now being removed from the manuscript because it is related to model setup and simulation.

Line 541. Remove the word "budget". Also, see the previous comment on how to present this trend value. Possibly write it as "averaged" emission for the same period.

This is now being removed from the manuscript because it is related to model setup and simulation.

Line 545. The underestimation is only ~30% based on the optimized model emissions (13.3 Tg vs. 19.3 Tg). When re-writing this sentence, make it clear which emissions this comparison refers to.

This is now being removed from the manuscript because it is related to model setup and simulation.

Lines 551-552. "...was caused by fossil fuel-related emissions, likely from oil associated and natural gas sources". The conclusion of the 2010-2011 peak coming from oil and gas emissions is stated in the text (line 390) only as a possible explanation and is not based on any of the main results (see Fig 3) and discussions made in the text (lines 330, 390, 430, 478). Even figure 3 (lower panel with model optimized sectoral emissions) does not reflect this statement, as other sectors such as SWD, RES, and TRO show the same increase in 2010-2011. Therefore, this conclusion should be removed from here and from the abstract since the authors have no strong results to reach such a conclusion in the manuscript.

This is now being removed from the manuscript because it is related to model setup and simulation.

Lines 552-553. "The global ethane trends cannot be well simulated by advanced atmospheric chemistry modeling ". This is a strong claim that is not supported in the text. If there are clear examples of studies that have had struggles modeling ethane trends, they should be cited here and in the text so that this conclusion is consistent with the main document.

This is now being removed from the manuscript because it is related to model setup and simulation.

TECHNICAL CORRECTIONS

Line 174- Move "(13.2 Tg/yr)" to the end of line 172.

This is now being removed from the manuscript because it is related to model setup and simulation.

Line 187- Add upper case in "Real..."

This is now being removed from the manuscript because it is related to model setup and simulation.

Line 321. Model estimation or optimized model results?

This is now being removed from the manuscript because it is related to model setup and simulation.

Add the word "modeled" every time the text refers to regional trends. For example, in line 475, it should say: "The modeled regional trends..."

This is now being removed from the manuscript because it is related to model setup and simulation.

**Reviewer #3 (Referee-report #1)**

Review comments on "Northern hemispheric atmospheric ethane trends (2006- 2016) with reference to methane and propane"

Author(s): Mengze Li et al. MS No.: essd-2021-246

MS type: Data description paper

General comments

Li et al. present long-term ethane mole fraction data obtained from airborne observation in the UTLS region in the Northern Hemisphere through IAGOC-CARIBIC project for the period 2006–2016. The authors also present simultaneously measured propane and methane mole fraction data as well as output of simulated atmospheric ethane using the EMAC model to understand the contributions of different source regions/sectors to the observed ethane variations.

After the revision, the manuscript and dataset has been generally improved, but at the same time it has also revealed significant issues for publication. Before going into the details, I would like to emphasize again that the observational ethane, propane, and methane data obtained from this project is very unique and useful for the scientific community, because there exist only limited airborne observation platforms that can collect the air samples in the UTLS region and can measure ethane, propane, and methane with high quality over the 10 years. I think the data can used by many researchers in this field after the issues as below have been resolved.

The current manuscript is mainly focusing on the comparison of the trend components derived by Prophet algorithm between observations and model outputs, which even precedes the presentation of original observational data. I think this has hidden the overall features of dataset as well as data quality and is not well suitable for the data description paper. I would like to ask the authors to pay more attention about the observation data itself (i.e., rigorous data classification in terms of altitudes, latitudes, longitudes, PV, and etc, also see my general comment 2). Now, the authors have newly added the timeseries of raw measured data together with their estimated trend by the Prophet (Fig 5 and Fig 8

and other figures in the supplement). The figures are very useful and helpful to understand the overall structure of dataset – for example, how the sampling frequency is for 2006-2016 in this project, how the variance of the raw data is, and how the trend is properly estimated. For the future revision, I would like to suggest that the figures are presented before presenting the comparison with model nor respective components by Prophet, as long as the authors purpose is the publication of their observational dataset. I think to guarantee the quality of the dataset it depends on whether the QC of the observation data is done properly and the presentation of observation data itself is reasonable, rather than the comparison with the model output.

We appreciate a lot that the reviewer provides very helpful and constructive comments for second-round revision. We are also glad to see that the reviewer finds our data very unique and useful for the scientific community. In this second iteration, we have removed all the model setup and simulations and now focus entirely on the measurement data, as suggested by both reviewers. We have made major changes to the manuscript, added resampling methods to estimate trend uncertainty, and discussion on data classification, as suggested by the reviewer.

1)      The current data classification in terms vertical axis is based on whether it is lower or higher than 2 PVU only. I found the data regarded as upper troposphere contains even much lower altitudes than expected (946 m at minimum). The change of such spatiotemporal sampling density has significantly affected the estimated trends and conclusion. For example, one of the main conclusions of this paper (and dataset) is that a sharp increase of atmospheric ethane was observed in the upper troposphere in 2010-2011, the cause of which is estimated to be fossil fuel related sources. However, according to the Fig 5 and Fig 8, it is found there is no observation data around half month before and after 2010, and then two flight data just after the sampling program restarted in 2010 showed overall higher mole fractions with relatively small variance, which looks affecting the rapid change of trend estimated by Prophet (Fig 5). I found most of these flight data were mainly sampled at altitudes lower than 8000 m (please see the attached Figure A). I would say such an increase in 2010 is not trend in the upper troposphere nor caused by the fossil emission increase, but is just the reflection of the vertical profile in the trace gases (see Figure B). Similar points can be said to the stratospheric data because it contains

different strength of stratospheric data (from 2 PVU to ~12 PVU, see Figure C-D). Please reconsider how the data classification is reasonable to be presented as the data description paper.

We thank the reviewer for pointing this out. We have now (1) excluded samples collected below 8,000m from analysis, (2) added more figures and discussion on the data classification for the lower stratospheric trends, see section 3.3.4. As suggested by the reviewer, we now identify the tropospheric samples as altitude >8,000m & PV<2, and the lower stratospheric samples as PV>2. We also presented the trends of the lower part (2≤ PV<6 PVU) and upper part (PV≥6 PVU) of the lower stratosphere, to investigate the changes of trends within the lower stratosphere and the difference when using different data classifications.

2)    Under the temporal sparseness of data in several years and in several regions, the analysis using Prophet would have underestimated the uncertainty in estimated trend. I don't think the uncertainty can be evaluated just by changing the fitting parameters "changepoint_prior_scale" (as authors did), because the estimated trend cannot usually avoid the overfitting when there is a large temporal data gap. If the authors would still keen to estimate the trend from Prophet, I would like to request to do some bootstrap tests, for example, jackknife method, to evaluate the robustness of estimated trend. Such tests should be especially important when estimating the trends for each geographical region, due to the sparseness of their data compared with whole NH dataset (i.e., ASI in the upper troposphere and NAM, ASI, and EUR in the lower stratosphere). Any discussions and interpretations on the seasonality and trend (and comparison with model) should be made after such cautious statistic test is done.

This is a good point. We now use resampling methods to evaluate the trend uncertainty (section 2.2). We have 20 sub-regions (4 categories: upper troposphere, lower stratosphere, lower part of the lower stratosphere, upper part of the lower stratosphere; 5 regions of each category: whole NH, EUR, NAM, ASI, ROW), dataset of each region is resampled 20 times with each subset consisting 90% of the sample number. For each subset, we run 10 times "Prophet" algorithms using 10 levels of fitting scale, and use the average of 10 fittings as the non-linear trend for that subset. For example, there are 3,288 upper tropospheric samples, we run 10 fittings with Prophet to estimate the trend, then we

resample the dataset of 3,288 samples 20 times randomly, each subset from resampling consists 3,288*90%= 2,959 samples, we run 10 fittings with Prophet for each subset, and plot the trends of 20 subsets as the trend uncertainty for the original dataset which has 3,288 samples. That means, for each sub-region, we run 10 times level of fittings with Prophet for each of the 21 datasets (1 original + 20 subsets from resampling), that is 210 times run of Prophet. We have 20 sub-regions, so in total 4,200 run of Prophet was done and the results are shown in the figures and manuscript to evaluate the robustness of trend uncertainty.

3) I think the procedure of model optimization is not so unique nor well organized, which makes me wonder if the model output is useful for understanding observation data and is worth publication, contrary to the significance of observation data. If the authors main purpose is to publish the observational data, I would suggest they focus more on the presentation of observational data itself and statistical analysis after rigorous data classification and may remove the model output from the publication of data. Otherwise, if their focus is both the observation data and model output, I would suggest they develop and sophisticate their model optimization procedure for the future use of model data.

We have now removed all the model outputs and focus on the presentation of observational data.

Considering the above, I would recommend this paper for publication after major revisions.

Specific comments

Overall: There are many confusions between "mole fraction" and "trend". After reading through the manuscript several times, I can imagine the authors somewhere refer to the trend component derived from Prophet as trend (e.g., Fig 3-8 and related sentences), but it is not always easy for readers to follow because we generally expect the ethane trend as in Table 2 or its related sentences (the unit, %/year or ppt/year). Also there still exists multiple places where it is hard to distinguish if the authors are referring to modeled results or observations.

We have removed all the model outputs. We have also added clarification on different trends in section 2.2 Trend analyses. We now refer the trend from linear fitting on the entire time period as "linear trend", and the trends derived from Prophet algorithm as "non-linear trend" because Prophet outputs trends for each day when the day has observations, and such trends are non-linear.

L25-L26: Now I wonder if the statement of emission estimate can be removed because the aim of this study using model is not to validate the emission inventories and also because the emission optimization methodology is not well sophisticated.

We agree. This is now being removed from the manuscript because it is related to model setup and simulation.

L28 "An ethane plume": I would recommend replacing it by "An sharp increase of ethane". "An ethane plume" sounds like more short-term intrusion of polluted air masses but here it seems saying about inter-annual variations. However, as I commented, I think it is also reflected by changes in sampling altitudes (please see general comment 2).

We have removed this statement, and added figures and text in manuscript to reflect the changes in sampling altitudes.

L29-31 "and higher temporal-spatial resolution data of ethane are needed.": I would suggest deleting it here since there is no discussion about the issues of spatiotemporal resolutions of emission inventories in the main text.

Done, this has been deleted.

L52-L76: As long as I understand, the author is reviewing the study of emissions estimates from top-down approach in this paragraph, and discussing the issue of discrepancy between top-down and bottom-up estimates in the next paragraph. I would suggest clarifying it since there are several confusions as below.

This paragraph has been deleted as it is related to model and emission inventories.

L54-L56: The sentence can be deleted since this paragraph is the summary of top-down approach (as long as I understand); thus the discussion on inventory is just confusing.

Done.

L66: Insert "based on top-down approach" after "global emission estimates".

This paragraph has been deleted.

L76-L80: This sentence is not clear. Please rephrase.

Done. Now the revised sentence reads:

"Previous investigations of the distribution, emissions, lifetime, and atmospheric trends of ethane have been mostly based on surface-based measurements. These have been either from a regionally focused intensive field measurement campaign (e.g. Kort et al. (2016)) or from networks of remote sampling stations (e.g. Franco et al. (2015), Helmig et al. (2016))."

L108-L109: By inspecting the dataset, I found 946 m sampling altitude data exists as the minimum altitude. I would strongly recommend the author presents the information on the frequency of sampling altitudes and reconsider the data classification.

Done. We have shown the sampling altitudes of all the data in Figure 1, and classify tropospheric samples as altitude>8,000 m and PV<2 for further analysis, as suggested by the reviewer.

L109-L114: I would suggest the author explains the measured trace gases by each three detectors separately or just explains the measurements of ethane, propane, and ethane, respectively. Under the current writing, the reader cannot still find the correspondence between each detector and measured trace gas.

Done. We have now explained the measurement by each detector separately.

L135-L137 "Three injections of calibration standards": What does it mean? Need clarification.

Done. We have revised the text and it reads now:

"Three additional calibration standards samples were measured in between samples of each flight sequence in order to monitor the quality of measurements and reduce uncertainty."

L164-L165 "to reduce the uncertainty": What is the "uncertainty" here? Why combining AIR, BIB, and BIO into one sector can reduce the "uncertainty"?

This is now being removed from the manuscript because it is related to model setup and simulation.

L166-L176: Actually, this part is still not clear and confusing to me.

This is now being removed from the manuscript because it is related to model setup and simulation.

L168-L172: What is "the total amount"? The global total anthropogenic ethane emission? Despite the optimization, it is said the value used in this study is ~11.8 Tg/yr whereas the value used in Pozzer et al. (2020) is 13.2 Tg/yr? Are these global total ethane emissions including natural sources; thus they are not consistent even after the adjustment?

This is now being removed from the manuscript because it is related to model setup and simulation.

L173-L174 "We further optimized …": Do you mean the 45% increase is done from CAMS-GLOB-ANT v4.2 inventory itself or from that after multiplied by 2.47 as mentioned above? In the current manuscript, it sounds like the authors attempted to optimize the global total ethane emissions in the latter (two-step) method. If the optimization method is the former, I would suggest deleting or much reduce the sentence in L168-L172 to avoid confusion and clarity.

This is now being removed from the manuscript because it is related to model setup and simulation.

L174-L176: What is "whole dataset"? Do you mean many iterations were performed by changing the scaling factors to determine the minimum of RMSE? Please clarify more. The authors didn't present the raw output of model simulations, but I think at least it is desirable to present the correlation plot between observations and model outputs after the optimization (maybe in the supplement).

This is now being removed from the manuscript because it is related to model setup and simulation.

L205-L208: Please see my general comment 3. I wonder if this approach is reasonable when estimating the trend uncertainty in this case whose data is sparse in time.

We have added resampling methods to estimate trend uncertainty, as suggested by the reviewer, details please see our response to general comment 3.

L210-L213: What is "start" and "end" representing here? For example, in L366-L368, how the start and end of the data were determined? Is it also applied to the output derived from Prophet or applied to the raw observational data? Is it strongly dependent on the measurement data at the tEnd and tstart.

We have replaced this with linear fit, and explained more in detail in section 2.2.

L216-L251: The two paragraphs are just listing of information and hard to read. Please summarize and be more concise. The descriptions of emission estimates are also mixed in L232-L234 and L234-L235. Since the authors intended to summarize atmospheric ethane trend here, the sentence of emissions can be removed for the readability. The headline can also be changed as "Literature perspective of global atmospheric ethane trends" in this case.

We have now put these two paragraphs in the Introduction section as suggested by another reviewer. We have removed the parts regarding emission estimates as they are not relevant to the study. We have also summarized the main messages in the beginning of each paragraph so that it is easier for readers to follow.

L272-L273: Any possibility that the sampling altitudes in EUR are lower than the other regions? As I commented in L108-L109, the information of sampling altitudes for each region is also necessary to be presented (and please reconsider the data classification when needed).

We have added Figure 1 and Figure S3 to show the distribution of sampling altitudes, and added text in the manuscript to address this (section 3.2.2). Indeed, the sampling altitudes in EUR are lower than other regions.

L285-L359: Please see my general comment 1. I think the comparison of estimated ethane trends between observations and model (Figure 3-4) should be followed by the presentation of raw observational data itself (Figure 5). This is important for potential data user because they use the raw data not the estimated trend derived by Prophet.

Done. We have now removed all the model output.

L309-L311: Delete?

Done.

L323-L324: As commented in L28, the annual increase is not short-term.

This is now being removed from the manuscript because it is related to model setup and simulation.

L329-L331: This sentence is not clear to me. Please rephrase it.

This is now being removed from the manuscript because it is related to model setup and simulation.

L333-L334: FEF looks rather increasing over 2011-2013 in Figure 3a.

This is now being removed from the manuscript because it is related to model setup and simulation.

L335-L336 "We note…": This sentence is just suddenly appearing here. Delete?

This is now being removed from the manuscript because it is related to model setup and simulation.

L337-L346: This may not be an important finding from this study and can be removed. The author didn't optimize each emission sector but just roughly scaled up by 45% from CAMS-GLOB-ANT v4.2?.

This is now being removed from the manuscript because it is related to model setup and simulation.

L366-L374: Please see the comment in L210-L213. Also how to estimate the max and min?

Done. This has been removed and replaced by linear fit methods.

L377-L380: As commented, the high mole fractions in the first two flights in 2010 would have been due to the lower sampling altitudes. I think it is an overinterpretation to relate it to the increase of fossil emissions in 2010-2011 at this stage. After data classification with altitudes, it would be useful to present some scatter plots of ethane- methane or ethane-propane when the ethane mole fractions are high levels in order to relate with fossil sources.

We have removed the interpretation on fossil sources, and now the manuscript focuses on data itself.

L389-L391: I wonder why the authors write this here. As long as I understand, here the authors are attempting to present not the accuracy of their model, but the validity of their optimized global total ethane emission?

This is now being removed from the manuscript because it is related to model setup and simulation.

L407-L431: Please see my general comment 3. Please reconsider investigating the uncertainty before going into the detailed discussion even if the authors still wish to keep these parts.

Done, we have removed all the model simulation.

L463-L464: I think this is an important issue on the data classification, not limitations. Also, as I commented, the issue of temporal data gap should also be critical when estimating the trend.

We have added more discussion on data classification, and used resampling methods to examine the trend analysis uncertainty, please see our replies to general comment 2.

L472 "the WMO tropopause": Replace it by "the WMO thermal tropopause".

Done.

L474-L476 & L481-L483: I don't think the authors need to add these sentences.

Done. We have removed these sentences.

L501-L504: Please add the information of the calibration scale, instrument, measurement precision, the number of measurements, and standard deviations into the dataset.

Will be done when I upload the dataset.

L510-L512: Please clarify more.

This part has been removed because it is related to model simulation.

L515-L516: I think the remarks on emission estimates need to be minimized because the current optimization method is very rough and the authors are also saying the aim of model simulations is to better understand the contributions from each emission sector, not improving the emission inventories.

Done. We have removed all the model simulation.

L518-L519 "a factor of three": As long as I read through the paper, the author has increased the global ethane emissions by 45% from CAMS-GLOB-ANT v4.2 to match their atmospheric ethane observations, not by a factor of three.

This part has been removed because it is related to model simulation.

L526-L527: There is no presentation on the simulation results of methane in this study. It sounds very sudden.

This part has been removed.

Table 1: Please specify "inventory" as CAMS-GLOB-ANT v4.2. Also the row of "Biogenic emission" and "Biomass burning" also needs a headline. Please add the explanation that the sum of BIO + BB and (a) or (b) is equal to total sources.

This is now being removed from the manuscript because it is related to model setup and simulation.

Table 2 "~": What does it means? The uncertainties presented here represent the same manners? Also please unify the usage of parentheses in the first column; otherwise add the new column to describe the locations of measurements?

We have added explanation in the table legend: "Table 1. Summary of studies reporting ethane trends in the (a) troposphere and (b) stratosphere. Parentheses in first column indicates the locations of measurements."

Figure 5: The trends in Propane are difficult to distinguish (also in Figure 8, S6-S9). As commented, these time series plots are very important and preferred to come first rather than comparisons with model outputs. Also useful to show the sum of all components (i.e. best fit curve) derived from Prophets.

This figure has been now deleted because it is related to model simulation.

Figure S3: Not clear the meaning of "trend analysis uncertainty". What is the meaning of percentage here? The uncertainty range is generally smaller for the period 2006–2009 when the number of data is small, potentially due to the over-fitting to the sparse data?

*This figure has been now deleted because it is related to model simulation.*

Figure S4: I don't think this figure is necessary even in the supplement. The contributions of each emission sector to the atmospheric simulated ethane trends are already shown in Figure 3 and Figure 6. This figure looks rather confusing.

*This figure has been now deleted because it is related to model simulation.*

Table S2, Figure S5, Figure S11: Isn't it the sectoral contribution to "simulated ethane mole fractions" not "ethane trends"? If it is saying the values of trend component derived from Prophet, please clarify so.

*This figure has been now deleted because it is related to model simulation.*

Figure S6-S9, S12-S15: Same as the comments in Figure 5.

*This figure has been now deleted because it is related to model simulation.*

Figure S7 and S13: It looks the observational data number is very sparse and difficult to estimate the robust trend? Need further statistical test to investigate the effect of time data gap (e.g., bootstrap method) if the authors still wish to present their trends.

*As the reviewer suggested in previous comments, we have added resampling methods to evaluate the robustness of trend analysis and added text in the manuscript to raise the point of data gap.*

Figure S12-S14: There are vertical lines in the left sides. Need removed.

*This figure has been now deleted.*

**Technical corrections**

L52: Insert "atmospheric" before "observations".

This is now being removed from the manuscript because it is related to model setup and simulation.

L253 "It is noted …above, any…": Conjunction is required.

Done. We have revised the sentence, it now reads:

"It is noted that our aircraft samples have significantly different spatial distributions compared with the studies summarized in the Introduction section, therefore, any comparison should be made in a careful manner."

L253-L257 "…of data quality, a process": Also need conjunction.

Done. Now it reads:

"When comparing surface and airborne datasets from multiple locations to assess global atmospheric changes, as it will become increasingly important to ensure comparability of data quality. A process that has begun through the grounding of a World Calibration Center for VOCs, although this dataset predates this initiative."

L273-L278: Very long sentence. Please separate it into two sentences.

Done.

L459-L497: I doubt the writing style in this section is following a proper manner. There are headlines starting from alphabet in parenthesis - some are complete sentences and some are not but all headlines start with lower cases.

We will consult with the journal editorial team to see if the writing style is proper and how to revise.

L499: 4 Data availability.

Done.

L507: 5 Conclusions

Done.

Figure 2: Insert "observed" before "over the whole …".

This figure has been removed and replaced by other figures.

[Figure]

Figure A: The observed ethane, methane, propane mole fractions for the whole NH upper troposphere (same as Figure 5 in the authors' manuscript), together with sampling altitudes.

[Figure]

Figure B: Same as Figure B, but for sampling altitudes over 8000m.

[Figure]

Figure C: The observed ethane, methane, propane mole fractions for the whole NH lower stratosphere (same as Figure 8 in the authors' manuscript), together with PV.

[Figure]

Figure D: Same as Figure C, but for PV over 6 PVU.

---

## Author Response (AR3)

**referee-report-1**

REVISION LI ET AL.

GENERAL COMMENTS

After the second revision, the authors followed the suggestions made by reviewers and

eliminated the model simulation previously included. The new manuscript focuses on the observational dataset and the trends derived from it.

This new manuscript is well written and the methods and results are presented in a clear manner. I recommend this paper for publication after minor revisions.

We appreciate a lot that the reviewer provided lots of helpful comments and suggestions through all rounds of revisions, and the quality of our manuscript has been improved a lot and it now fits the journal much better.  We have now revised the manuscript according to the reviewer's comments.

**SPECIFIC COMMENTS**

Line 90. Suggest changing the second part of this statement to: "…, including regions without ground measurements."

Done.

Line 146-150. If the term "whole NH" will be used, it is better to change the name of ROW to RNH "Rest of Northern Hemisphere" and make sure data from the Southern Hemisphere is not included. The way it is written right now gives the impression that some data in ROW corresponds to the Southern Hemisphere, and thus, combining the four regions into one does not make sense.

Done. We have replaced ROW with RNH in text and figures.

Line 154. Change "must" for "does".

Done.

Lines 194-201. These lines should be moved to the methods section (2.1).

Done.

Line 238. Specify what you mean by "low" (maybe add a cut-off number).

Done.

Section 3.3. Add information regarding other halogens like Bromine.

Done. We have added more information and the text reads:

"In the stratosphere, the OH radical concentration on average decreases by a factor of 10 compared with tropospheric OH levels, whereas halogen (e.g. chlorine (Cl) and bromine (Br)) are more abundant and react faster with ethane, methane and propane, therefore they play a greater relative role in ethane, methane and propane oxidation (Li et al., 2018)."

Line 285. Define "large". Provide a value for this difference.

Done.

Line 388. Since some values are negative, "growth rates" should be changed for "annual rates of change in atmospheric abundances of".

Done.

**referee-report-2**

Manuscript Review for ESSD

Li et al.,: Northern hemispheric atmospheric ethane trends in the upper troposphere and lower stratosphere (2006-2016) with reference to methane and propane

This manuscript is a revision/resubmission of a previously evaluated submission to the same journal. The authors made several changes from the prior submission. Despite their efforts, in my opinion, some important shortcomings remain. Concerns by Reviewer #3 about the classification and filtering of air samples have not been fully resolved. I have also reviewed comments and revisions from the very first submission. Several of the previously articulated problems are still valid:

We thank the reviewer for the comments. We have addressed them and our replies are shown below.

As pointed out in the prior reviews and comments, there appears to have been a minimum in the atmospheric mole fraction of ethane around the year 2009-2010, with declining levels prior, and increasing levels afterwards. These trends were seen consistently in monitoring records from the Northern Hemisphere. With this having been well established, it does not seem appropriate to conduct linear trend analysis across this minimum. Obviously, the trend results will then all depend on which starting and end time one chooses before and after the minimum. Despite this having been emphasized before, the authors adhere to this analysis and dedicate a full section to this method and discussion of the results.

We understand the reviewer's point on separating the linear trend into different time periods, instead of across all time. This is seen in some literature, e.g. in Helmig et al. (2016) and Franco et al. (2015). As the reviewer commented, the selection of starting and end time will influence the trend results. This is true for either applying one linear trend across all time (the way we did in our study), or multiple time periods (as the reviewer suggested). Both Helmig et al. (2016) and Franco et al. (2015) have estimated ethane trends over Jungfraujoch, but the authors chose different starting and end time and their trends are different. Helmig et al. (2016) used May 2014 as the ethane turnover time, whereas Franco et al. (2015) used Jan 2014. The increasing

ethane trends after turnover in 2014 was estimated as 4.2%/yr by Helmig et al. (2016) and 4.9%/yr by Franco et al. (2015), showing a ~17% difference. The choice of the turnover time of ethane trends is often somewhat subjective, lacking objective criteria. Due to these reasons, we don't think separating trends into different time periods will significantly improve the robustness of linear trend analysis. There are also some studies using only one linear fit across all time, e.g. Simpson et al. (2012) (published in *Nature*) applied one linear fit for ethane trend over 1985-2010. Although their study didn't cover the ethane turnover time in 2014, we can still see by eye from Fig.4 of their paper that the ethane trends during 1985-2000 vs 2000-2010 are very different. As the scope of this journal is to present an overview of the datasets, instead of data interpretation, we therefore applied linear trend analysis across all time to provide a general overview of our datasets. The trend analysis results in Figs.3 and 6 show non-linear trends with higher temporal variability and are less influenced by the choice of starting and end time. Our linear and non-linear trend analysis helps readers to understand our dataset from different aspects.

The second problem I see, and this is a big one, is that to the best of my understanding and evaluation, it seems that the data that are used for attempting these trend analyses are ways to variable in its source and representation of a particular altitude (troposphere versus stratosphere) as well as latitude for justifying this analysis. The authors conducted filtering of their data based on polar vorticity (PV) and continental areas. I don't think this worked. The data that are displayed in Figures 2 and the derived trend results in Figure 3 and 5 defy the atmospheric behavior of methane, ethane, and propane that has been well established in other literature. Ethane is a relatively long-lived gas in the atmosphere, with its lifetime being on the scale of or longer than the mixing time scale in the Northern Hemisphere troposphere.

Therefore, it does not seem plausible that upper troposphere ethane would behave so completely different than ethane in mid to lower tropospheric records. I have pulled out some data examples in the figures below, i.e. from Mauna Loa Observatory, HI, USA, Jungfraujoch, Switzerland, and Summit, Greenland. There are many more that can be easily reproduced from records in the World Data Center for Reactive Gases (EBAS/Nilu). I have intentionally chosen examples from high altitude monitoring stations as those are deemed to best describe the behavior in the lower free troposphere. The important take home messages are that seasonal cycles of ethane are very regular. Year-to-year changes are relatively small, just a few percent, if at all. These are all deemed high quality observations, reflecting the seasonal and year-to-year changes at a fixed site. However, there is a very substantial latitudinal gradient, with ethane dropping steeply from mid-latitudes to the tropics (see Figure 4).

The data presented in this paper look very different, showing a much higher variability and much higher year-to-year changes, which the authors interpret as atmospheric concentration trends and attempt to relate to surface emissions changes. I don't think this is appropriate and correct. I think it is much more likely that the variability seen in these data to a large extend reflects differences in the origin of the air masses that were sampled, both in altitude (tropospheric versus stratospheric contribution) and with latitude. What the authors interpret as trends likely are year-to-year changes in the fraction of collected samples having one sort of these influences versus another.

We would like to emphasize again the difference between surface and lower stratosphere. Many surface observation sites are remote from any known anthropogenic or natural sources to represent the background conditions. At these sites, the ethane, methane or propane trends are largely influenced by the variation of their sink, i.e. OH radicals. As OH radicals are high in summer and low in winter, the trends of ethane, methane and propane observed at ground are low in summer and high in winter. This explains why surface seasonal cycles of ethane are regular and their yearly variation are small. Whereas it is very different for the upper troposphere/lower stratosphere (UT/LS). The air at UT/LS region has a broad range of age (often 0-3 years) (Bönisch et al., 2009), which means the air represents a mixture of air parcels emitted 0-3 years ago. The transport time from surface to UT/LS is about 3 months which is similar to ethane's atmospheric lifetime. Orbe et al. (2015) has shown that compounds within the lower stratosphere of the northern hemisphere are predominately influenced by the Brewer-Dobson circulation, not the northern mid-latitude emissions (~ 10%). As we have stated clearly in the manuscript, region designated does not correspond to the source region, only the geographical location of the data points. Due to these reasons, we don't think surface trends are comparable with UT/LS trends.

As for the reviewer's comment on the use of PV filter, it is a commonly used method in the community and has been examined many times in the literature, e.g. Sprenger et al. (2007); Sprenger et al. (2003). Air samples with PV larger than 2 are commonly defined as stratospheric samples. Creating a new filter for stratospheric samples is out of our scope and will make our results difficult to be compared with other studies.

I don't think the paper is publishable as it is. Since this journal focuses on presentation of data, a way to proceed may be to just publish the data by themselves without the trends analyses.

Another, totally different approach might be to focus instead on the analysis of ratios

of the hydrocarbon over methane. One could also utilize the propane/ethane ratio as an indicator of photochemical age and possibly derive a fractional contribution of stratospheric air on the samples and then compare that with the PV rating. Or maybe one could attempt filtering the data based on these ratios and attempt a trend analysis on the ratio-filtered data. But all of that may be well beyond the scope of an ESSD publication and be better suited for a regular Atmospheric Science journal.

We think trend analysis provides important information about our data, and is within the scope of the journal (e.g. https://essd.copernicus.org/articles/14/1917/2022/). We present the ethane, methane and propane trends from different analysis approaches (seasonal, linear, and non-linear trends) which provide the readers a broad spectrum of understanding our dataset and therefore the possibility of using our data for their work in the future. The discussion about differences with trends and variability derived from surface data is very interesting and indispensable, but will need extensive interpretation supported by atmospheric modelling.

We thank the reviewer for providing ideas on data analysis and interpretation. These are important points that we will consider for our future work. Indeed, as the reviewer wrote, they are beyond the scope of this journal.

References:

Bönisch, H., Engel, A., Curtius, J., Birner, T., and Hoor, P.: Quantifying transport into the lowermost stratosphere using simultaneous in-situ measurements of $SF_6$ and $CO_2$, Atmos. Chem. Phys., 9, 5905-5919, 10.5194/acp-9-5905-2009, 2009.

Franco, B., Bader, W., Toon, G. C., Bray, C., Perrin, A., Fischer, E. V., Sudo, K., Boone, C. D., Bovy, B., Lejeune, B., Servais, C., and Mahieu, E.: Retrieval of ethane from ground-based FTIR solar spectra using improved spectroscopy: Recent burden increase above Jungfraujoch, Journal of Quantitative Spectroscopy and Radiative Transfer, 160, 36-49, 10.1016/j.jqsrt.2015.03.017, 2015.

Helmig, D., Rossabi, S., Hueber, J., Tans, P., Montzka, S. A., Masarie, K., Thoning, K., Plass-Duelmer, C., Claude, A., Carpenter, L. J., Lewis, A. C., Punjabi, S., Reimann, S., Vollmer, M. K., Steinbrecher, R., Hannigan, J. W., Emmons, L. K., Mahieu, E., Franco, B., Smale, D., and Pozzer, A.: Reversal of global atmospheric ethane and propane trends largely due to US oil and natural gas production, Nature Geoscience, 9, 490-495, 10.1038/ngeo2721, 2016.

Orbe, C., Newman, P. A., Waugh, D. W., Holzer, M., Oman, L. D., Li, F., and Polvani, L. M.: Airmass origin in the Arctic. Part I: Seasonality, Journal of Climate, 28, 4997-5014, 2015.

Simpson, I. J., Sulbaek Andersen, M. P., Meinardi, S., Bruhwiler, L., Blake, N. J., Helmig, D., Rowland, F. S., and Blake, D. R.: Long-term decline of global atmospheric ethane concentrations and implications for

methane, Nature, 488, 490-494, 10.1038/nature11342, 2012.

Sprenger, M., Croci Maspoli, M., and Wernli, H.: Tropopause folds and cross-tropopause exchange: A global investigation based upon ECMWF analyses for the time period March 2000 to February 2001, Journal of Geophysical Research: Atmospheres, 108, 2003.

Sprenger, M., Wernli, H., and Bourqui, M.: Stratosphere–troposphere exchange and its relation to potential vorticity streamers and cutoffs near the extratropical tropopause, Journal of the atmospheric sciences, 64, 1587-1602, 2007.

[Figure]

 Figure 1: Ethane atmospheric mole fractions (red crosses) at Mauna Loa Observatory (3397 m asl, data and graph retrieved from https://www.nilu.no/).

[Figure]

Figure 2: Ethane atmospheric mole fractions (red crosses) at Jungfraujoch (3463 m asl, data and graph retrieved from https://www.nilu.no/).

[Figure]

Figure 3: Ethane atmospheric mole fractions from flask sampling (red dots) and in-situ monitoring (black dots) at Summit, Greenland (3200 m asl, graph reproduced from WMO Reactive Gases Bulletin (https://library.wmo.int/index.php?lvl=notice_display&id=19877#.Yr9JIXbMKUk ).

[Figure]

Figure 4: Latitudinal gradient and seasonal cycle of atmospheric ethane (graph reproduced from WMO Reactive Gases Bulletin (https://library.wmo.int/index.php?lvl=notice_display&id=19877#.Yr9JIXbMKUk )).

---

## Author Response (AR4)

**Editor's comments:**

Dear authors,

Thank you for providing your responses to the referees' comments and submitting your revised manuscript. I believe that the IAGOS-CARIBIC measurements and your dataset are valuable to the scientific community, therefore I am recommending that your manuscript be published after minor revisions.

Specifically, please address in your manuscript the concerns brought up by referee 2 regarding,

1) The selection of time period used for calculating the trends: In particular, how would the linear trend be affected if you selected a different time period (i.e. before or after the 2009/2010 minimum).

2) Expand section 3.4 on Limitations to include a discussion on the selection of regions. Why were these regions selected (i.e. are they considered homogeneous in terms of latitudinal variability?) How might selecting different regions (e.g. smaller areas) affect the UTLS trends?

Please don't hesitate to contact me directly if you have any questions or would like me to provide feedback on a proposed draft before you submit the revised version.

Best regards,

Nellie

**Authors' response:**

Dear Editor,

We really appreciate your time and effort in editing our manuscript and providing helpful comments. We have carefully addressed the concerns from referee 2 and our replies are shown below.

For 1), we now calculate the trends of all regions listed in Table 2 without the data in 2009 and 2010, and the values are listed in Table S2. The difference between trends with all data and trends without 2009/2010 for all the regions listed in Table 2 are 10.7% (ethane), 3.1% (methane), and 24.7% (propane) (median). The difference reflects the atmospheric variability (propane highest, methane lowest). We now add the following text in the limitation section to address this point:

"(d) Growth rates are different when choosing different time periods. Excluding data collected in 2009 and 2010 when trend anomalies were seen in some regions shows 10.7% (ethane), 3.1% (methane), and 24.7% (propane) difference (median) (Table S2) compared with the growth rates calculated with all 2006-2016 data (Table 2). The difference is associated with the atmospheric variability of trace gases, but not the quality of data."

For 2), we now add the following text in the limitation section:

"(e) Selection of regions. Regions of interest are selected at continental scale to ensure enough number of observations (>95) in each region. The spatial variability within each region is considered homogeneous. This might introduce uncertainty but its quantification requires more observations or model simulations. The typical transport time from surface to tropopause is about 1-3 months, assuming a wind speed of 1m/s, air travels 2,592-7,776 km within 1-3 months which is larger than continental coverage. Thus the assumption of homogeneous spatial variability at continental scale may not have large uncertainty."

Best regards,

Mengze on behalf of team